# FLEA: Provably Robust Fair Multisource Learning from Unreliable Training Data

**Eugenia Iofinova**[*]
*eugenia.iofinova@ist.ac.at*
*Institute of Science and Technology Austria (ISTA)*

**Nikola Konstantinov**[*]
*nikolahristov.konstantinov@inf.ethz.ch*
*ETH AI Center and ETH Department of Computer Science*

**Christoph H. Lampert**
*chl@ist.ac.at*
*Institute of Science and Technology Austria (ISTA)*

**Reviewed on OpenReview:** *https://openreview.net/forum?id=XsPopigZXV*

## Abstract

Fairness-aware learning aims at constructing classifiers that not only make accurate predictions, but also do not discriminate against specific groups. It is a fast-growing area of machine learning with far-reaching societal impact. However, existing fair learning methods are vulnerable to accidental or malicious artifacts in the training data, which can cause them to unknowingly produce unfair classifiers. In this work we address the problem of fair learning from unreliable training data in the robust multisource setting, where the available training data comes from multiple sources, a fraction of which might not be representative of the true data distribution. We introduce FLEA, a filtering-based algorithm that identifies and suppresses those data sources that would have a negative impact on fairness or accuracy if they were used for training. As such, FLEA is not a replacement of prior fairness-aware learning methods but rather an augmentation that makes any of them robust against unreliable training data. We show the effectiveness of our approach by a diverse range of experiments on multiple datasets. Additionally, we prove formally that –given enough data– FLEA protects the learner against corruptions as long as the fraction of affected data sources is less than half. Our source code and documentation are available at `https://github.com/ISTAustria-CVML/FLEA`.

## 1 Introduction

Machine learning systems have started to permeate many aspects of our everyday life, such as finance (e.g. credit scoring), employment (e.g. judging job applications) or even judiciary (e.g. recidivism prediction). In the wake of this trend, other aspects besides prediction accuracy become important to consider. One crucial aspect is *(group) fairness*, which aims at preventing learned classifiers from acting in a discriminatory way. To achieve this goal, fairness-aware learning methods adjust the classifier parameters in order to fulfill an appropriate measure of fairness. This strategy is highly successful, but only under idealized conditions of clean i.i.d.-sampled data. Unfortunately, *fairness-aware learning methods are not robust against unintentional errors or intentional manipulations of the training data.*

---

[*]These authors contributed equally.

This manuscript differs from the accepted camera-ready version: the definition of *demographic parity violation* has been changed to allow for two-sided bounds in Theorem 1. The proof in Appendix *F* has been adjusted accordingly.

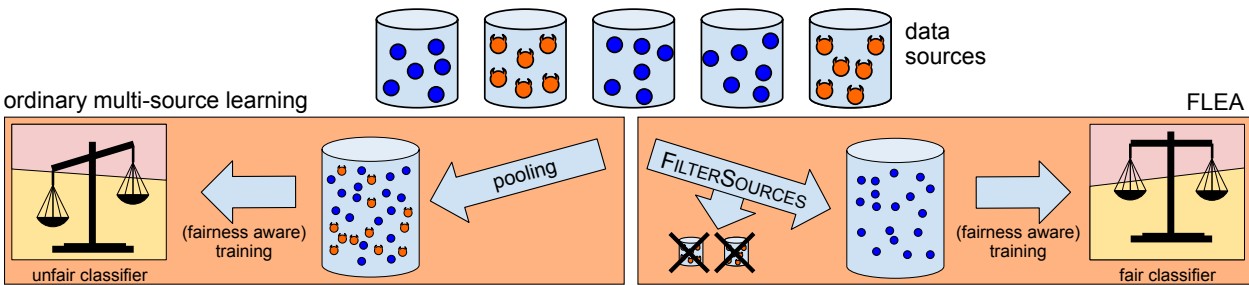

Figure 1: Illustration of robust multisource learning with FLEA: (top) We are given multiple sources, some of which might contain noisy or manipulated data. (left) Ordinary multisource learning pools the data from all sources, which can cause the resulting classifier to be inaccurate and/or unfair, even if fairness-aware training is employed. (right) FLEA filters the data before pooling, thereby suppressing likely corrupted sources. This allows fairness-aware training to succeed.

In this work, we address this problem in a setting where the training data is not one monolithic block, but rather a centralized collection of data obtained from multiple sources. This is, in fact, a common scenario. For instance, organizations that specialize in large-scale data mining, such as large hospital chains or political analysis firms, may receive data that is collected separately from multiple physical locations or data vendors. In such cases, it may be that not all of these data sources are completely trustworthy, and so robustness concerns arise.

In order to achieve robustness to unreliable data in such contexts, we propose a new algorithm, FLEA (Fair LEarning against Adversaries). FLEA adds a filtering step on top of any standard fairness-aware learning algorithm and effectively identifies and suppresses data sources that could have a negative impact on the classifier fairness or accuracy. Thereby, FLEA acts as a procedure that guarantees robustness in the context of fair learning.

To accomplish this, we introduce a new dissimilarity measure, *disparity*, that measures the maximum achievable difference in classifier fairness between two data sources. We use this measure as a filtering criterion, since it has the property of flagging changes in the data distribution that can be potentially harmful for the end-classifier fairness. We combine this with the existing *discrepancy* measure, which plays an analogous role for the classifier accuracy, and the *disbalance*, which measures changes to the group composition of the training data. We show both empirically and theoretically that a combination of these three measures provides a sufficient criterion for detecting harmful data, as long as the fraction of harmful sources is less than half.[1]

While previous method for robust fairness-aware learning were only able to protect against specific data issues, such as random label flips, FLEA ensures that even a worst-case adversary is unable to negatively affect the training process: either the changes to the data are minor and will not hurt learning, or they are large enough so that the affected data sources are identified and removed. Our theoretical analysis provides finite sample guarantees and certifies the ability of FLEA to learn classifiers with optimal fairness and accuracy in the infinite sample size limit. Our extensive experimental evaluation demonstrates FLEA's practical usefulness in suppressing the effect of corrupted data when learning fair models, even in cases where previous robust methods fail.

## 2 Preliminaries and related work

### 2.1 Fair classification

Throughout this work, we adopt a standard classification setting in which the task is to predict a binary label $y \in \mathcal{Y} = \{0, 1\}$ for any $x \in \mathcal{X}$. For a fixed data distribution $p(x, y) \in \mathcal{P}(\mathcal{X} \times \mathcal{Y})$, the classic goal of learning

---

[1]The case where half or more sources are harmful is impossible to solve in general, see e.g. Charikar et al. (2017)

is to find a prediction function $f : \mathcal{X} \to \mathcal{Y}$ with high accuracy, i.e. small *risk*, $\mathcal{R}_p(f) = \mathbb{E}_{p(x,y)} \mathbb{1}\{y \neq f(x)\}$, where $\mathbb{1}\{P\} = 1$ if a predicate $P$ is true and $\mathbb{1}\{P\} = 0$ otherwise.

With the recent trend to consider not only the accuracy but also the *fairness* of a classifier, a number of statistical measures have been proposed to formalize this notion. In this work, we focus on the most common and simplest one, *demographic parity (DP)* (Calders et al., 2009). It postulates that the probability of a positive classifier decision should be equal for all subgroups of the population. Formally, we assume that each example $(x, y)$ also possesses a *protected attribute*, $a \in \mathcal{A}$, which indicates its membership in a specific subgroup of the population. For example, $a$ could indicate *race*, *gender* or a *disability*. For simplicity of exposition, we treat the protected attribute as binary-valued, but extensions to multi-valued attributes are straightforward by summing over all pairwise terms. Note that $a$ might be a component or a function of $x$, in which case it is available at prediction time, or it might be contextual information, in which case it would only be available for the learning algorithm at training time, but not for the resulting classifier at prediction time. We cover both aspects by treating $a$ as an additional random variable, and write the underlying joint data distribution as $p(x, y, a)$.

For a classifier $f : \mathcal{X} \to \{0, 1\}$, the *demographic parity violation*, $\Gamma_p$, and the empirical counterpart, $\Gamma_S$ for a dataset $S \subset \mathcal{X} \times \mathcal{Y} \times \mathcal{A}$ are defined as (Calders et al., 2009; Dwork et al., 2012),

$$\Gamma_p(f) = \mathbb{E}_{p(x|a=0)} f(x) - \mathbb{E}_{p(x|a=1)} f(x), \qquad \Gamma_S(f) = \frac{1}{n^{a=0}} \sum_{x \in S^{a=0}} f(x) - \frac{1}{n^{a=1}} \sum_{x \in S^{a=1}} f(x) \qquad (1)$$

Negative values of the demographic parity violation indicate unfairness against group 0, while positive values indicate unfairness against group 1. Analogous quantities can be defined for related fairness measures, such as *equality of opportunity* or *equalized odds* (Hardt et al., 2016; Zafar et al., 2017a). A detailed description of these and many others choices can be found in Barocas et al. (2019).

**Fairness-aware learning** In the last years, a plethora of algorithms have been developed that are able to learn classifiers that are not only accurate but also fair, see, for example Mehrabi et al. (2021a) for an overview. They mostly rely on one or multiple of four core mechanisms. *Preprocessing methods* (Kamiran & Calders, 2012; Calmon et al., 2017; Wang et al., 2019; Celis et al., 2020) change the training data to remove a potential bias. This is often simple and effective, but comes with the danger of reduced accuracy, since the data distribution at training time will not reflect the distribution at prediction time anymore. *Postprocessing methods* (Hardt et al., 2016; Woodworth et al., 2017; Chzhen et al., 2020) adjust the acceptance thresholds of a previously trained classifier for each protected group, so that the desired fairness criterion is met. This is a simple, reliable and often effective method, but it requires the protected attribute to be available at prediction time. *Penalty-based methods* (Kamishima et al., 2012; Zemel et al., 2013; Zafar et al., 2017b; Donini et al., 2018; Mandal et al., 2020; Chuang & Mroueh, 2021) add a regularizer or constraints to the learning objective that penalize or prevent parameter choices that lead to unfair decisions. *Adversarial methods* (Beutel et al., 2017; Wadsworth et al., 2018; Zhang et al., 2018; Lahoti et al., 2020) train an adversary in parallel to the classifier that tries to predict the protected attribute from the model outputs; if this cannot be done better than chance level, fairness is achieved.

Many other methods have been proposed, e.g. based on distributionally robust optimization (Rezaei et al., 2020) or tailored to a specific family of classifiers or optimization procedures (Cho et al., 2020; Tan et al., 2020). They all share, however, the property that accurate information about the data distribution and the protected attribute is needed at training time.

If the training data is not representative of the actual data distribution, e.g. it is noisy, biased, or has been manipulated, then fairness-enforcing mechanisms fall short (Kallus et al., 2020; Mehrabi et al., 2021b). Partial solutions have been proposed, e.g., when only the protected attribute or only the label is noisy (Lamy et al., 2019; Wang et al., 2020; Celis et al., 2021b;a; Mehrotra & Celis, 2021; Roh et al., 2021). However, as shown in Konstantinov & Lampert (2022), full protection against malicious manipulations of the training data is provably impossible when learning from a single dataset.

## 2.2 Robust multisource learning

**Learning from multiple sources** The multisource learning setting formalizes the increasingly frequent situation in which the training data is not collected as a single batch, but from multiple data sources (Ting & Low, 1997; Russakovsky et al., 2015). For fairness-aware learning, this means we are given $N$ datasets, $S_1, \ldots, S_N \subset \mathcal{X} \times \mathcal{Y} \times \mathcal{A}$. Each $S_i = \{(x_1^{(i)}, y_1^{(i)}, a_1^{(i)}), \ldots, (x_{n_i}^{(i)}, y_{n_i}^{(i)}, a_{n_i}^{(i)})\}$ contains i.i.d. samples from a data distribution $p_i(x, y, a)$. Given these datasets, the learning algorithm has the goal of selecting a prediction function $f$ from a hypothesis class $\mathcal{H}$ that has as-small-as-possible risk (expected prediction error) and unfairness (e.g. demographic parity violation) with respect to the unknown distribution at prediction time, $p(x, y, a)$, (also called *target distribution*). The classical setting of $p_1 = p_2 = \cdots = p_N = p$, we call *homogeneous multisource learning*. Otherwise, we call the setting *heterogeneous*.

In a *clean data scenario*, when all data distribution are the same or very similar to each other, then there is no drawback to simply merging all sources and training on the resulting large dataset. However, merging all data is not the best strategy when some of the data sources are *unrepresentative*, i.e. their data distribution differs a lot from the target one. Such data can occur accidentally, for instance due to biases in the data collection or annotation process. In some cases, such issues can be overcome by domain adaptation techniques (Ben-David et al., 2010; Crammer et al., 2008; Natarajan et al., 2013). Unrepresentative sources can also be the result of intentional manipulations, which are typically harder to detect and compensate for (Feng et al., 2019; Fowl et al., 2021). In fact, the datasets might not be samples from any probability distribution in that case, but adversarially constructed.

**Robust multisource learning** In this work we aim to cover as wide a range of possible problems with some of the data sources as possible. Therefore, we study the multisource learning problem in the presence of an *adversary*[2]. In this setting, the adversary observes an original collection of $N$ datasets, $\tilde{S}_1, \ldots, \tilde{S}_N$, where each $\tilde{S}_i$ contains i.i.d. samples from a data distribution $p_i(x, y, a)$. Next, the adversary manipulates the data in an arbitrary (deterministic or randomized) way with the only restriction that for a fixed subset of indices, $G \subset \{1, \ldots, N\}$, the data source remains unaffected. That is, $S_i = \tilde{S}_i$ for all $i \in G$, and $S_i$ is arbitrary for $i \notin G$. The subset $G$ is unknown to the learning algorithm, of course. The adversary model places no restrictions on the corruptions, and thus subsumes many scenarios that have to otherwise be studied in isolation. In particular, both data-quality issues, such as sampling bias, data entry errors or label noise, as well as malicious manipulations, such as class erasure or data poisoning, are covered as special cases.

Multisource learning with protection against potential manipulations is known as *robust multisource learning* (Erfani et al., 2017). In order to detect harmful sources, a natural approach is to compare all pairs of datasets with an appropriate distance measure and then use the pairwise distances to filter out sources that are far from the others. Key to the success of such an approach is using the right definition of distance. On the one hand one must be able to estimate the measure from finite sample sets in a statistically efficient way. Many common information-theoretic measures, such as *Kullback-Leibler divergence* (Kullback & Leibler, 1951), *total variation* (Tsybakov, 2009) or *Wasserstein distance* (Villani, 2009), do not fulfill this criterion. On the other hand, the measure must be sensitive enough such that if two sources appear similar then training on either of them must yield similar classifiers. Classical two-sample tests, such as Student's $t$-test (Student, 1908) or MMD (Gretton et al., 2012), fail to guarantee this.

In the context of multisource learning a measure that combines both useful properties is the *(empirical) discrepancy distance* (Kifer et al., 2004; Mohri & Medina, 2012). For two datasets, $S_1, S_2$, and a hypothesis set $\mathcal{H} \subset \{h : \mathcal{X} \to \mathcal{Y}\}$, it measures the maximal amount by which their estimates of the classification accuracy can differ:

$$\text{disc}(S_1, S_2) = \sup_{h \in \mathcal{H}} \left| \mathcal{R}_{S_1}(h) - \mathcal{R}_{S_2}(h) \right|, \tag{2}$$

where $\mathcal{R}_S(h) = \frac{1}{|S|} \sum_{(x,y) \in S} \mathbb{1}\{y \neq h(x)\}$ is the *empirical risk* of $h$ on $S$. In Konstantinov et al. (2020) the discrepancy is used as a distance measure to identify and suppress data sources that might harm the classifier's accuracy. However, the associated algorithm is mostly of theoretical interest: it only suppresses

---

[2]*Adversary* is the common computer science term for a process whose aim it is to prevent a system from operating as intended. Our adversaries manipulate the training data and should not be confused with adversaries in *adversarial machine learning*, such as *adversarial examples* (Goodfellow et al., 2015), or *generative adversarial networks* (Goodfellow et al., 2014).

those sources of which it is certain that they have been manipulated using thresholds that are derived from its generalization bound. As a consequence, it requires training sets that are too large to be practical. Similarly, Jain & Orlitsky (2020b) provide an analysis of the learning-theoretic limitations of robust multisource learning. Konstantinov & Lampert (2019) also use the discrepancy measure for detecting harmful data sources, but the proposed algorithm requires access to a reference set that is guaranteed to be free of data manipulations. In Qiao & Valiant (2018); Chen et al. (2019); Jain & Orlitsky (2020a) robust multisource learning is addressed using tools from robust statistics, but only in the context of discrete density estimation. The problem of achieving robustness to noisy data annotators is also related (Awasthi et al. (2017); Khetan et al. (2018)), but more restricted, as in our context we allow for arbitrary changes of the inputs and protected attributes, in addition to the labels.

All of the above works are tailored to the task of ensuring high accuracy of the learned classifiers or estimators, but they are not sensitive to issues of fairness. To our awareness, the only prior work that considers achieving fairness in a multisource learning setting and in the presence of data corruption is the one of Li et al. (2021b). However, that paper focuses on *personalized federated learning* and on a fairness objective tailored to federated learning, which postulates that models' performances should be relatively similar across edge devices. In contrast, we study a centralized setup, where privacy and communication issues are not present and where a *single global model is trained*. In addition, we aim to ensure that this model does not act discriminatory against members of protected subgroups, aligned with the classic notions of group fairness in supervised learning.

## 3 Fair multisource learning

The goal of this work is to develop a method that allows fairness-aware learning, even if some of the available data sources are unrepresentative of the true training distribution. For this, we introduce FLEA, a filtering-based algorithm that identifies and suppresses those data sources that would negatively impact the fairness of the trained classifier. Its main innovation is the *disparity measure* for comparing datasets in terms of their fairness estimates.

**Definition 1** (Empirical Disparity). For two datasets $S_1, S_2 \subset \mathcal{X} \times \mathcal{Y} \times \mathcal{A}$, their *empirical disparity* with respect to a hypothesis class $\mathcal{H}$ is

$$\mathrm{disp}(S_1, S_2) = \sup_{h \in \mathcal{H}} \left| \Gamma_{S_1}(h) - \Gamma_{S_2}(h) \right|. \tag{3}$$

where $\Gamma_S : \mathcal{H} \to \mathbb{R}$ is an empirical (un)fairness measure, such as the *demographic parity violation* (1).

The *disparity* measures the maximal amount by which the estimated fairness of a classifier in $\mathcal{H}$ can differ between using $S_1$ or $S_2$ as the basis of the estimate. A small disparity value implies that if we construct a classifier that is fair with respect to $S_1$, then it will also be fair with respect to $S_2$.

Definition 1 is inspired by the *empirical discrepancy* (2). Just as low discrepancy implies that a classifier learned on one dataset will have comparable accuracy as one learned on the other, low disparity implies that the two classifiers will have comparable fairness. FLEA makes use of the discrepancy as well as the disparity, because ensuring fairness alone does not suffice (e.g. a constant classifier is perfectly fair). As a third relevant quantity we introduce the *(empirical) disbalance*.

$$\mathrm{disb}(S_1, S_2) = \left| \frac{|S_1^{a=1}|}{|S_1|} - \frac{|S_2^{a=1}|}{|S_2|} \right|. \tag{4}$$

The disbalance compares the relative sizes of the protected groups of two datasets. Its inclusion is a technical requirement to be able to also formally prove that demographic parity fairness remains unaffected by corruptions.

In combination, disparity, discrepancy, and disbalance form an effective criterion for detecting dataset manipulations. This is most apparent in the homogeneous setting: if two datasets of sufficient size are sampled i.i.d. from distributions close to the target one, then by the law of large numbers we can expect all

three measures to be small. If one of the datasets is sampled like this (called *clean* from now on) but the other is manipulated, then there are two possibilities. It is still possible that all three values are small. In this case, equations (2)–(4) ensure that neither accuracy nor fairness would be negatively affected, and we call such manipulations *benign*. If at least one of the values is large, training on such a manipulated datasource could have undesirable consequences. Such manipulations we will call *malignant*. Finally, when comparing two manipulated datasets, discrepancy, disparity, and disbalance can each have arbitrary values.

In the heterogeneous setting, a path of similar reasoning applies, though the measures for clean sources will not approach exactly zero due to the difference in their data distributions.

### 3.1 FLEA: Fair learning against adversaries

We now introduce the FLEA algorithm, which is able to learn fair classifiers even if up to half of the datasets are noisy, biased or have been manipulated. Similar to classic outlier rejection techniques (Barnett & Lewis, 1984) and statistical two-sample tests (Corder & Foreman, 2014), the main algorithm (Algorithm 1) takes a filtering approach. Given the available data sources and additional parameters, it calls a subroutine that identifies a subset of clean or benign sources, merges the training data from these, and trains a (presumably fairness-aware) learning algorithm on the resulting dataset.

FLEA's crucial component is the filtering subroutine. This estimates the pairwise disparity, discrepancy and disbalance between all pairs of data sources and combines them, by summing,[3] into a matrix of dissimilarity scores (short: $D$-scores). As discussed above, large values indicate that at least one of the two compared sources must be *malignant*. It is not a priori clear, though, how to use this information. On the one hand, we do not know which of the two datasets is malignant or if both are.

On the other hand, malignant sources can also occur in pairs with small $D$-score, when both datasets were manipulated in similar ways. Finally, even the $D$-scores between two clean or benign sources will have non-zero values, which depend on a number of factors, in particular the data distributions and the hypothesis class.

FLEA overcomes this problem by using tools from robust statistics. For any dataset $S_i$, it computes a value $q_i$ (called $q$-value) as the $\beta$-quantile of the $D$-scores to all other datasets, where $\beta$ is a hyperparameter we discuss below. It then computes the $\beta$-quantile of all such values and selects those datasets with $q$-values up to this threshold.

---

**Algorithm 1** FLEA

**Input:** datasets $S_1, \ldots, S_N$
**Input:** quantile parameter $\beta$
**Input:** (fairness-aware) learning algorithm $\mathcal{L}$
1: $I \leftarrow \text{FILTERSOURCES}(S_1, \ldots, S_N; \beta)$
2: $S \leftarrow \bigcup_{i \in I} S_i$
3: $f \leftarrow \mathcal{L}(S)$
**Output:** trained model $f : \mathcal{X} \to \mathcal{Y}$

---

**Subroutine** FILTERSOURCES

**Input:** $S_1, \ldots, S_N; \beta$
1: **for** $i = 1, \ldots, N$ **do**
2:     **for** $j = 1, \ldots, N$ **do**
3:         $D_{i,j} \leftarrow \text{disc}(S_i, S_j) + \text{disp}(S_i, S_j) + \text{disb}(S_i, S_j)$
4:     **end for**
5:     $q_i \leftarrow \beta\text{-quantile}(D_{i,1}, \ldots, D_{i,N})$
6: **end for**
7: $I \leftarrow \{i : q_i \leq \beta\text{-quantile}(q_1, \ldots, q_N)\}$
**Output:** index set $I$

---

To see that this procedure has the desired effect of filtering out malignant datasets, we first look at the case in which the sources are homogeneous and $\beta = \frac{K}{N}$, where $K = |G| > \frac{N}{2}$ is the number of clean data sources.

For any clean dataset $S_i$, by assumption there are at least $K - 1$ other clean sources with which it is compared. We can expect the $D$-scores of these pairs are small, and, of course, that $D_{ii} = 0$. Because $\beta = \frac{K}{N}$, the $\beta$-quantile, $q_i$, is simply the $K$th-smallest of $S_i$'s $D$-scores. Consequently, $q_i$ will be at least as small as the result of comparing two clean sources. For benign sources, the same reasoning applies, since their $D$-scores are indistinguishable from clean ones. For a malignant $S_i$, at least $K$ of the $D$-scores will be large, namely the ones where $S_i$ is compared to a clean source. Hence, there can be at most $N - K$ small $D$-scores for $S_i$.

---

[3]Other aggregation methods would be possible, as long as they ensure to preserve large values, such as the maximum. This would yield similar theoretical guarantees, but we did not find it to perform better in practice.

Because $\beta N = K$ and $K > N - K$, the $\beta$-quantile $q_i$ will be at least as large as comparing a clean dataset to a malignant one.

Choosing those sources that fall into the $\beta$-quantile of the $q_i$ values means selecting the $K$ sources of smallest $q_i$ value. By the above argument, these will either be not manipulated at all, or only in a way that does not have a negative effect on either the fairness or the accuracy of the training process. In practice, the regimes of *large* and *small D*-scores can overlap due to noise in the sampling process, and the perfect filtering property will only hold approximately. We later discuss a generalization bound that makes this reasoning rigorous.

Revisiting the above arguments one sees that the guarantees on the $q_i$ follow also for any $\beta > \frac{N-K}{N}$, so in particular for $\beta \geq \frac{1}{2}$. To obtain the guarantee on the selected sources, $\beta \leq \frac{K}{N}$ suffices. Therefore, even if the exact value of $K$ is unknown in practice, setting $\beta = \frac{1}{2} + \frac{1}{N}$ for even $N$ and $\beta = \frac{1}{2} + \frac{1}{2N}$ for odd $N$ will always be working choices. These are also the values we use in our experiments.

In the heterogeneous situation, the $D$-scores between clean sources might not tend to zero for large $n$ anymore. However, they will approach the true discrepancy, disparity and disbalance values between the sources' distributions. From this, one can obtain a guarantee that the selected sources are not more dissimilar from each other than the clean sources are, which is the best one can hope for in the heterogeneous setting.

## 3.2 Implementation

FLEA is straightforward to implement, with only the discrepancy and disparity estimates in the FILTER-SOURCES routine requiring some consideration. Naively, these would require optimizing combinatorial functions (the differences of fraction of errors or positive decisions) over all functions in the hypothesis class. This task is at least as hard as the problem of separating two point sets by a hyperplane, which is known to be NP-hard (Marcotte & Savard, 1992) and even difficult to approximate under any real-world conditions. Instead, we exploit the structure of the optimization problems to derive tractable approximations.

We describe the procedure here and provide pseudocode in Appendix B. For the discrepancy (2) such a method was originally proposed in the domain adaptation literature (Ben-David et al., 2010): finding the hypothesis with maximal accuracy difference between two datasets is equivalent to training a binary classifier on their union with the labels of one of the datasets flipped.

For the disparity (3), we propose an analogous route. Intuitively, the optimization step requires finding a hypothesis that is as unfair as possible on $S_1$ (i.e. maximizes $\Gamma_{S_1}$) while being as unfair as possible in the opposite direction on $S_2$ (i.e. minimizes $\Gamma_{S_2}$), or vice versa. From Equation (1) one sees that a hypothesis $f$ is maximally positively unfair if it outputs $f(x) = 1$ on $S_1^{a=1}$ and $f(x) = 0$ on $S_1^{a=0}$, and maximally negatively unfair if it has the opposite outputs. Consequently, to estimate the disparity, we can use a classifier trained to predict $f(x) = a$ on $S_1$ and $f(x) = 1 - a$ on $S_2$. To give both protected groups equal importance, as the definition requires, we use per-sample weights that are inversely proportional to the group sizes.

## 3.3 Theoretical analysis

The informal justification of FLEA can be made precise in the form of a generalization bound. In this section we present our theoretical guarantees for FLEA. We begin by stating formally the assumptions we make on the data generating process, both for the heterogeneous and the homogeneous cases discussed above. We then state our main theoretical result, which certifies the performance of FLEA in the both the homogeneous and the more general heterogeneous case. Finally, we briefly outline the main proof steps. The full proofs can be found in Appendix F.

### 3.3.1 Assumptions and formal adversarial model

First we present our formal set of assumptions, directly in the general setting of *heterogeneous* data sources. A crucial parameter here setup is $\eta$, which denotes the amount of variability between the clean sources' distributions. The case of $\eta = 0$ recovers the *homogeneous* setup.

We assume the following data generation model, similar to the one of Qiao & Valiant (2018). By $p(x, y, a)$ we denote the target distribution. It is unknown to the learning algorithm, though potentially known to the adversary. Initially, there are $N$ datasets $\tilde{S}_1, \ldots, \tilde{S}_N$, with the $i$-th set of samples being drawn i.i.d. from a distribution $p_i(x, y, a)$. These distributions might differ from the target distribution $p$ by at most $\eta$ in terms of total variation both with respect to the overall distributions as well as the conditional distributions with respect to $a$. Formally, we assume the following conditions for $i = 1, \ldots, N$:

$$\mathrm{TV}(p_i(x, y, a), p(x, y, a)) \le \eta, \quad \text{and} \quad \max_{z \in \mathcal{A}} \left\{ \mathrm{TV}(p_i(x, y | a = z), p(x, y | a = z)) \right\} \le \eta, \tag{5}$$

where TV is the *total variation distance* between probability distributions Halmos (2013).

Once the clean datasets $\tilde{S}_1, \ldots, \tilde{S}_N$ are sampled, an *adversary* operates on them. This results in new datasets, $S_1, \ldots, S_N$, which the learning algorithm receives as input. The adversary is an arbitrary (deterministic or randomized) function $\mathcal{F} : \prod_{i=1}^{N} (\mathcal{X} \times \mathcal{Y} \times \mathcal{A})^{n_i} \to \prod_{i=1}^{N} (\mathcal{X} \times \mathcal{Y} \times \mathcal{A})^{n_i}$, with the only restriction that for a fixed subset of indices, $G \subset \{1, \ldots, N\}$, the data remains unchanged. That is, $S_i = \tilde{S}_i$ for all $i \in G$, and $S_i$ is arbitrary for $i \notin G$. For simplicity, we refer to a dataset $S_i$ or a source $i \in [N]$ as clean if $i \in G$.

### 3.3.2 Theoretical guarantees on FLEA

We are now ready to state our theoretical guarantee on FLEA. For simplicity of notation, we present the case where all sources have the same number of samples. Results for general sample sizes can be obtain in an analogous way. We first state the guarantees for the homogeneous situations, which we obtain in fact as a corollary for $\eta = 0$ of the general theorem later in this section.

**Theorem 1** (Homogeneous setting)**.** *Assume that $\mathcal{H}$ has a finite VC-dimension $d \ge 1$. Let $p$ be an arbitrary target data distribution and without loss of generality let $\tau = p(a = 0) \in (0, 0.5]$. Let $S_1, \ldots, S_N$ be $N$ datasets, each consisting of $n$ samples, out of which $K > \frac{N}{2}$ are sampled i.i.d. from the distribution $p$. For $\frac{1}{2} < \beta \le \frac{K}{N}$ and $I = \textsc{FilterSources}(S_1, \ldots, S_N; \beta)$ set $S = \bigcup_{i \in I} S_i$. Let $\delta > 0$. Then there exists a constant $C = C(\delta, \tau, d, N, \eta)$, such that for any $n \ge C$, the following inequalities hold with probability at least $1 - \delta$ uniformly over all $f \in \mathcal{H}$ and against any adversary:*

$$\left| \Gamma_S(f) - \Gamma_p(f) \right| \le \widetilde{\mathcal{O}} \left( \sqrt{\frac{1}{n}} \right), \qquad \left| \mathcal{R}_S(f) - \mathcal{R}_p(f) \right| \le \widetilde{\mathcal{O}} \left( \sqrt{\frac{1}{n}} \right), \tag{6}$$

*where $\widetilde{\mathcal{O}}$ indicates Landau's big-O notation for function growth up to logarithmic factors (Cormen et al., 2009).*

**Discussion**   To analyze the statement, we observe that Equation (6) ensures that for large enough training sets the filtered training data $S$ becomes an arbitrarily good representative of the true underlying data distribution with respect to the classification accuracy as well as the fairness. Moreover, the approximation holds uniformly across all hypotheses in the class. We note that a similar generalization bound for accuracy in the homogeneous setting is given in Konstantinov et al. (2020).

This uniform convergence property is similar to the classic concentration results from learning theory for learning with clean data (Shalev-Shwartz & Ben-David, 2014; Woodworth et al., 2017) and essentially ensures that using the data $S$ is *safe* for the purposes of fairness-aware learning. Indeed, since the empirical risk and fairness deviation on the filtered data $S$ are good estimates of the true population measures, any algorithm that uses the data $S$ to learn a hypothesis with good empirical fairness and accuracy will also perform well at prediction time.

Note that despite the intuitive conclusion, the result from Theorem 1 is highly non-trivial, due to the presence of data corruption. For example, in the case of learning from a single datasource in which a constant fraction of the data can be manipulated, an analogous theorem is provably impossible (Kearns & Li, 1993; Konstantinov & Lampert, 2022). This observation also implies that no learning algorithm can guarantee accurate and fair learning if it is given access to the training data only after all sources have been merged.

For the general situation ($\eta \ge 0$), we obtain the following guarantees:

**Theorem 1** (Heterogeneous setting). *Assume that $\mathcal{H}$ has a finite VC-dimension $d \geq 1$. Let $p$ be an arbitrary target data distribution and without loss of generality let $\tau = p(a = 0) \in (0, 0.5]$. Let $S_1, \ldots, S_N$ be $N$ datasets, each consisting of $n$ samples, out of which $K > \frac{N}{2}$ are sampled i.i.d. from data distributions $p_i$ that are $\eta$-close to the distribution $p$ in the sense of Section 3.3.1. Assume that $18\eta < \tau$. For $\frac{1}{2} < \beta \leq \frac{K}{N}$ and $I = \text{FILTERSOURCES}(S_1, \ldots, S_N; \beta)$ set $S = \bigcup_{i \in I} S_i$. Let $\delta > 0$. Then there exists a constant $C = C(\delta, \tau, d, N, \eta)$, such that for any $n \geq C$, the following inequalities hold with probability at least $1 - \delta$ uniformly over all $f \in \mathcal{H}$ and against any adversary:*

$$\left| \Gamma_S(f) - \Gamma_p(f) \right| \leq \mathcal{O}(\eta) + \tilde{\mathcal{O}}\left( \sqrt{\frac{1}{n}} \right), \qquad \left| \mathcal{R}_S(f) - \mathcal{R}_p(f) \right| \leq \mathcal{O}(\eta) + \tilde{\mathcal{O}}\left( \sqrt{\frac{1}{n}} \right). \tag{7}$$

**Discussion** In contrast to the homogeneous situation, an additional factor linear in $\eta$ enters the right hand side of the bound. As discussed in Section 3.1, we believe that such a factor will be unavoidable in the heterogeneous case: $\eta$ is a measure of the dissimilarity between the clean sources and the target distribution. Therefore, even without data corruptions, the accuracy of a learned classifier for the unknown target distribution $p$ will be limited by how close that is to the training distributions (Bartlett, 1992; Hanneke & Kpotufe, 2020). That the increase in risk is of order $\eta$ can be seen from a simple binary classification example: let $p(x, y) = \mathbb{1}\{x \geq 0.5\}$ and $p_1(x, y) = \mathbb{1}\{x \geq 0.5 + \eta\}$, such that $\text{TV}(p, p_1) = \eta$. Then, the optimal classifier learned with respect to $p_1$ has expected error $\eta$ with respect to $p$.

In cases when $\eta$ is small, the stated result still certifies that the empirical risk and fairness deviation on the data $S$ are good estimates of the underlying population values. Therefore, the discussion from the homogeneous case applies here as well, meaning that the data $S$ is "safe" to train on for the purposes of fairness-aware learning.

**Proof sketch** The proof consists of three steps. First, we characterize a set of values into which the empirical risks and empirical deviation measures of the clean data sources fall with probability at least $1 - \delta$. Then we show that because the clean datasets cluster in such a way, any individual dataset that is accepted by the FILTERSOURCES algorithm provides good empirical estimates of the true risk and the true unfairness measure. Finally, we show that the same holds for the union of these sets, $S$, which implies the inequalities in the theorem. For the risk, the last step is a straightforward consequence of the second. For the fairness, which is not simply an expectation or average over per-sample contributions, a more careful derivation is needed that crucially uses the disbalance measure as well. For details of the steps, please see Appendix F.

### 3.4 Computational complexity of FLEA

In order to apply FLEA, we must train two classifiers (one to estimate disc and one to estimate disp) for every pair of sources in the dataset. Assuming that the maximum number of points in every data source (after adversarial perturbation) is $n$, the complexity of training all of these classifiers is therefore bounded above by $\mathcal{O}(N^2 F(2n))$, where $N$ is the number of data sources, and $F(t)$ is the computational complexity of running the chosen method of learning a classifier on a data set of size $t$. Then, all but $K$ sources are filtered out and the data in the rest is combined, resulting in a total computational complexity of $\mathcal{O}(N^2 F(2n) + F(Kn))$. Since $\frac{N}{2} < K \leq N$, which term dominates depends on the complexity of the learning algorithm. In the case that $F$ is subquadratic, the total complexity is dominated by the first term; if $F$ is quadratic, by neither term, and if $F$ is of higher complexity then the combined training dominates. Please see Appendix sections A.2 and A.4 for specific details and running time of our experimental setup.

## 4 Experiments

FLEA's claim is that it allows learning classifiers that are fair even in the presence of perturbations in the training data. Due to its filtering approach it can be used in combination with any existing learning method. For our experiments, we run it in combination with four fairness-aware learning methods as well as one fairness-unaware one against a variety of adversaries on five established fair classification datasets. We benchmark our method against the corresponding base learning algorithms without pre-filtering, as well as against four robust learning baselines.

### 4.1 Experimental setup

We report experiments in two setups: for *homogeneous* and *heterogeneous* data sources.

**Datasets**    For the homogeneous setup we use four standard benchmark datasets from the fair classification literature: `COMPAS` (Aingwin et al., 2016) (6171 examples), `adult` (48841), `germancredit`(1000) and `drugs` (1885) (Dua & Graff, 2017). To obtain multiple identically distributed sources, we randomly split each training set into $N \in \{3, 5, 7, 9, 11\}$ equal-sized parts, out of which the adversary can manipulate $\lfloor \frac{N-1}{2} \rfloor$. For the heterogeneous case we use the 2018 US census data of the `folktables` dataset(Ding et al., 2021). We form 51 similarly but not identically distributed data sources by using up to 10000 examples from each of the US-states. Out of these 5, 10, 15, 20 or 25 can be manipulated. Details about the data preprocessing and feature extraction steps can be found in the supplemental material.

In all cases, we use *gender* as the exemplary protected attribute, because it is present in all feature sets. We train linear classifiers by logistic regression without regularization, using 80% of the data for training and the remaining 20% for evaluation. All experiments are repeated ten times with different train-test splits and random seeds. We measure the mean and standard deviation of the accuracy and the fairness of the learned classifiers, where we compute fairness as $1 - \Gamma_S$, where $\Gamma_S$ is the demographic parity violation on the test set.

**Fairness-Aware Learners**    We use FLEA in combination with four fairness-aware learning methods that have found wide adoption in research and practice. In all cases, we use *logistic regression* as the underlying classification model.

- *Fairness regularization* (Kamishima et al., 2012) learns a fair classifier by minimizing a linear combination of the classification loss and the empirical unfairness measure $\Gamma_S$, where for numeric stability, in the latter the binary-valued classifier decisions $f(x)$ are replaced by the real-valued confidences $p(f(x) = 1|x)$.

- *Data preprocessing* (Kamiran & Calders, 2012) modifies the training data to remove potential biases. Specifically, it creates a new dataset by *uniform resampling* (with repetition) from the original dataset, such that the the fractions of positive and negative labels are the same for each protected group. On the resulting unbiased dataset it trains an ordinary fairness-unaware classifier.

- *Score postprocessing* (Hardt et al., 2016) first learns an ordinary (fairness-unaware) classifier on the available data. Afterwards, it determines which decision thresholds for each protected groups achieve (approximate) demographic parity on the training set, finally picking the fair thresholds with highest training accuracy.

- *Adversarial fairness* (Wadsworth et al., 2018) learns by minimizing a weighted difference between two terms. One is the loss of the actual classifier; the other is the loss of a classifier that tries to predict the protected attribute from the real-valued outputs of the main classifier.

For completeness, we also include plain logistic regression as a *fairness-unaware learner*. The supplemental material details the learners' implementations and parameters.

**Adversaries**    In a real-world setting, one does not know what kind of data quality issues will occur. Therefore, we test the baselines and FLEA for a range of adversaries that reflect potentially unintentional errors as well as intentional manipulations.

- *flip protected (FP), flip label (FL), flip both (FB)*: the adversary flips the value of protected attribute, of the label, or both, in all sources it can manipulate.

- *shuffle protected (SP)*: the adversary shuffles the protected attribute entry in each effected batch.

- *overwrite protected (OP), overwrite label (OL)*: the adversary overwrites the protected attribute of each sample in the affected batch by its label, or vice versa.

- *resample protected (RP)*: the adversary samples new batches of data in the following ways: all original samples of protected group $a = 0$ with labels $y = 1$ are replaced by data samples from other sources which also have $a = 0$ but $y = 0$. Analogously, all samples of group $a = 1$ with labels $y = 0$ are replaced by data samples from other sources with $a = 1$ and $y = 1$.

Table 1: Result of FLEA and baselines for robust fairness-aware multisource learning with homogeneous and heterogeneous data sources. Reported accuracy and fairness values are the minimal (worst-case) ones across all tested data manipulations in the respective settings. See main text for an explanation of the methods and details of the experimental setup.

(a) homogeneous: `adult`, `COMPAS`, `drugs` and `germancredit` datasets with 5 sources of which 2 are unreliable.

| method | adult accuracy | adult fairness | COMPAS accuracy | COMPAS fairness | drugs accuracy | drugs fairness | germancredit accuracy | germancredit fairness |
|---|---|---|---|---|---|---|---|---|
| naive | $66.2_{\pm1.1}$ | $77.6_{\pm1.2}$ | $63.1_{\pm1.8}$ | $78.9_{\pm2.3}$ | $60.0_{\pm2.5}$ | $72.3_{\pm3.1}$ | $58.0_{\pm4.0}$ | $78.7_{\pm5.3}$ |
| robust ensemble | $69.9_{\pm0.4}$ | $90.9_{\pm1.6}$ | $64.9_{\pm1.1}$ | $87.1_{\pm2.6}$ | $61.0_{\pm2.1}$ | $66.8_{\pm4.5}$ | $61.9_{\pm2.9}$ | $62.3_{\pm7.6}$ |
| DRO (Wang et al., 2020) | $52.8_{\pm0.3}$ | $15.4_{\pm1.3}$ | $53.7_{\pm1.4}$ | $57.1_{\pm23.9}$ | $55.2_{\pm2.5}$ | $48.5_{\pm27.0}$ | $34.4_{\pm6.0}$ | $81.1_{\pm12.5}$ |
| hTERM (Li et al., 2021a) | $66.8_{\pm0.9}$ | $50.7_{\pm1.8}$ | $52.9_{\pm2.5}$ | $29.3_{\pm12.9}$ | $54.6_{\pm3.0}$ | $40.7_{\pm9.1}$ | $41.2_{\pm4.0}$ | $25.2_{\pm9.4}$ |
| (Konstantinov et al., 2020) | $69.3_{\pm0.4}$ | $77.6_{\pm1.2}$ | $63.1_{\pm1.8}$ | $78.9_{\pm2.3}$ | $60.0_{\pm2.5}$ | $72.3_{\pm3.1}$ | $58.0_{\pm4.0}$ | $78.7_{\pm5.3}$ |
| FLEA (proposed) | $70.2_{\pm0.4}$ | $97.9_{\pm1.1}$ | $65.9_{\pm1.0}$ | $94.5_{\pm3.0}$ | $64.3_{\pm1.4}$ | $92.6_{\pm4.2}$ | $65.9_{\pm3.0}$ | $93.4_{\pm3.9}$ |
| oracle | $70.3_{\pm0.4}$ | $98.2_{\pm1.0}$ | $66.2_{\pm1.1}$ | $96.2_{\pm1.3}$ | $64.4_{\pm1.5}$ | $93.6_{\pm3.3}$ | $67.3_{\pm3.0}$ | $94.4_{\pm4.0}$ |

(b) heterogeneous: `folktables` dataset with $N = 51$ sources of which $N - K \in \{5, 10, 15, 20, 25\}$ are unreliable.

| method | $N-K=5$ accuracy | fairness | $N-K=10$ accuracy | fairness | $N-K=15$ accuracy | fairness | $N-K=20$ accuracy | fairness | $N-K=25$ accuracy | fairness |
|---|---|---|---|---|---|---|---|---|---|---|
| naive | $74.4_{\pm0.2}$ | $93.4_{\pm0.8}$ | $73.7_{\pm0.2}$ | $87.0_{\pm0.8}$ | $72.9_{\pm0.5}$ | $80.1_{\pm0.9}$ | $71.2_{\pm0.8}$ | $73.4_{\pm0.6}$ | $58.2_{\pm6.2}$ | $73.9_{\pm1.0}$ |
| robust ensemble | $74.9_{\pm0.2}$ | $97.1_{\pm0.3}$ | $74.3_{\pm0.2}$ | $93.8_{\pm0.4}$ | $73.5_{\pm0.3}$ | $89.1_{\pm0.5}$ | $71.9_{\pm0.3}$ | $81.7_{\pm0.7}$ | $65.8_{\pm1.1}$ | $60.4_{\pm2.2}$ |
| DRO (Wang et al., 2020) | $65.2_{\pm0.8}$ | $96.0_{\pm0.7}$ | $68.1_{\pm1.5}$ | $95.2_{\pm0.7}$ | $66.2_{\pm0.9}$ | $85.8_{\pm2.6}$ | $66.1_{\pm1.3}$ | $77.4_{\pm12.2}$ | $58.1_{\pm5.6}$ | $6.7_{\pm8.5}$ |
| hTERM (Li et al., 2021a) | $76.3_{\pm0.3}$ | $73.9_{\pm2.0}$ | $74.3_{\pm0.6}$ | $63.4_{\pm1.3}$ | $71.0_{\pm0.7}$ | $52.2_{\pm1.8}$ | $65.3_{\pm1.1}$ | $45.9_{\pm1.1}$ | $64.7_{\pm0.4}$ | $39.6_{\pm1.4}$ |
| (Konstantinov et al., 2020) | $74.3_{\pm0.2}$ | $93.4_{\pm0.8}$ | $73.7_{\pm0.2}$ | $87.0_{\pm0.8}$ | $72.9_{\pm0.5}$ | $80.1_{\pm0.9}$ | $71.2_{\pm0.8}$ | $73.4_{\pm0.6}$ | $58.2_{\pm6.2}$ | $73.9_{\pm1.0}$ |
| FLEA (proposed) | $75.4_{\pm0.2}$ | $99.4_{\pm0.2}$ | $75.4_{\pm0.2}$ | $99.5_{\pm0.2}$ | $75.4_{\pm0.2}$ | $99.5_{\pm0.2}$ | $75.3_{\pm0.2}$ | $99.4_{\pm0.2}$ | $74.0_{\pm1.4}$ | $94.2_{\pm1.5}$ |
| oracle | $75.2_{\pm0.2}$ | $99.5_{\pm0.3}$ | $75.2_{\pm0.2}$ | $99.6_{\pm0.2}$ | $75.3_{\pm0.2}$ | $99.7_{\pm0.2}$ | $75.3_{\pm0.2}$ | $99.7_{\pm0.2}$ | $75.1_{\pm0.3}$ | $99.6_{\pm0.4}$ |

- *random anchor (RA0/RA1)*: these adversaries follow the protocol introduced in Mehrabi et al. (2021b). After picking *anchor* points from each protected group they create poisoned datasets consisting of examples that lie close to the anchors but have opposite label to them. The difference between RA0 and RA1 lies in which combinations of label and protected attribute are encouraged or discouraged.

- *random (RND)*: the adversary randomly picks one of the strategies above for each source.

- *identity (ID)*: the adversary makes no changes to the data.

We include ID to certify that FLEA does not unnecessarily damage the learning process in the case when the training data is actually clean. The other adversaries either weaken the correlations between the protected attribute and the target data, thereby masking a potential existing bias in the data, or they strengthen the correlation between the protected attribute and the target label, thereby increasing the chance that the learned classifier will use the protected attribute as a basis for its decisions. In both cases, the dataset statistics at training time will differ from the situation at test time, and the efficacy of a potential mechanisms to ensure fairness at training time can be expected to suffer. For a more detailed discussion of the adversaries' effects, please see the supplemental material.

**Baselines** To the best of our knowledge, FLEA is the only existing method to tackle fair learning under arbitrary data manipulations. To nevertheless put our results into context, we compare it to four baselines: 1) a *robust ensemble* (similar to Smith & Martinez (2018)), which learns separate classifiers on each datasource and then combines their decisions by a majority vote. 2) A *distributionally robust optimization (DRO)* approach as proposed in Wang et al. (2020) to address noisy protected attributes. 3) Hierarchical *tilted empirical risk minimization* (hTERM) Li et al. (2021a), which aims at enforcing *robustness* by a *softmin* across per-sources losses, which themselves express a form of *fairness* by a *softmax*-loss across protected groups. 4) The *filtering approach* of Konstantinov et al. (2020) which uses discrepancy to identify manipulated sources but does not specifically aim to preserve fairness. More details on these can be found in the supplemental material. Further candidates could be Roh et al. (2020); Konstantinov & Lampert (2019), but these are not applicable in our setting, as they require access to guaranteed clean validation data.

## 4.2 Results

The results of our experiments show a very consistent picture across different datasets, base learners and adversaries. For the sake of conciseness, for FLEA we only present the results using a regularization-based fairness-aware learner in the main manuscript. Results for other learners are qualitatively the same and can be found, together with more detailed results and ablation studies, in the supplemental material.

In Table 1 we report results for six learning methods: an ordinary learner that is fairness-aware but not protected against data manipulations (naive), the proposed FLEA, and the four baseline methods: the robust ensemble, DRO (adapted from Wang et al. (2020)), hTERM (following Li et al. (2021a)), and discrepancy-based filtering Konstantinov et al. (2020). In addition, we report the value of a hypothetical oracle-based learner that knows which of the sources are actually clean and learns only on their data.

Each entry in the table is the *minimum* accuracy and fairness in the respective setting across all eleven tested adversaries. We choose this worst-case measure because it allows a compact representation and reflects the fact that a real-world system should be robust against all possible data errors or manipulation simultaneously. Results broken down by individual adversaries are provided in the supplemental material.

Examining the results, a comparison of the *naive* results with the *oracle* confirms that the need for robust learning method is real: naive fairness-aware learning is not sufficient to ensure fair (or accurate) classifiers in the presence of unreliable data.

An ideal robust method should achieve results approximately as good as the *oracle* result, as this would indicate that the adversary was indeed not able to negatively affect the learning process beyond the unavoidable loss of some training data. The results show that FLEA comes close to this behavior, but none of the other methods does. In the homogeneous setting (Table 1a), for the largest dataset, `adult`, FLEA reliably suppresses the effects of all tested adversaries. It learns classifiers with accuracy and fairness almost exactly those of a fair classifier trained only on the clean data sources. For the other datasets, `COMPAS`, `drugs` and `germancredit`, FLEA increases the accuracy and fairness to levels only slightly below the oracle. In all cases, FLEA's results are as good as or better than the baselines; the robust ensemble is also able to improve fairness to some extent, but it does not reach the oracle results.

The DRO-based and hTERM approaches show highly volatile behavior. For some adversaries they improve fairness or accuracy, but for some adversaries they fail severely. Consequently, their min-aggregated values in the table are often even lower than for the naive method. Note that these results should be seen in context though: Wang et al. (2020) is designed for a different and less challenging data manipulation model. Li et al. (2021a)'s notion of robustness and fairness differ from the ones we employ in this work.

The approach from Konstantinov et al. (2020) has almost no effect. Only for the largest dataset, `adult`, it yields a slight accuracy improvement. This can be explained by the fact that the method only removes sources that it can confidently identify as manipulated. The theory-derived thresholds for this are quite strict, so the method is ineffective unless a lot of data is available. The observed characteristics of the different methods hold also for the other base learners, see the supplemental material.

In the heterogeneous setting (Table 1b) the results show similar trends: for $N - K \in \{5, 10, 15, 20\}$, FLEA manages reliably to filter out the malignant sources, such that the accuracy and fairness of the learned classifiers matches the one of the oracle method almost perfectly. The robust ensemble has a positive effect, but less so than FLEA. For this data, DRO somewhat improves fairness, but this comes at a loss of accuracy. The method from Konstantinov et al. (2020) has no noticeable effect. For $N - K = 25$, FLEA still performs best, although a bit worse than the hypothetical best oracle. Presumably, this is because the combined effect of distribution differences between the sources and the uncertainty due to finite sampling when estimating disc, disp and disb are too large to perfectly allow a decision which 26 sources to keep and which 25 to exclude.

## 5    Conclusion

We studied the task of fairness-aware classification in the setting when data from multiple sources is available, but some of them might by noisy, contain unintentional errors, or even have been maliciously manipulated. Ordinary fairness-aware learning methods are not robust against such problems and often fail to produce fair classifiers. We proposed a filtering-based algorithm, FLEA, that is able to identify and suppress those data sources that would negatively affect the training process, thereby restoring the property that fairness-aware learning methods actually produce fair classifiers. We showed the effectiveness of FLEA experimentally, and we also presented a theorem that provides formal guarantees of FLEA's efficacy.

Despite our promising results, we consider FLEA just a first step on the path toward making fairness-aware learning more robust. One potential future step is to include other notions of fairness besides *demographic parity*. So far, FLEA can already be used as it is with classifiers that enforce other fairness criteria. However, our theoretical guarantees do not holds for these, as the *disparity* measure that enters our filtering step is not tailored to them. We do not see fundamental problems in deriving filtering steps for other fairness notions that are also defined in terms of properties of the joint distribution of inputs, outputs, and protected attributes, such as *equality of opportunity* or *equalized odds*. However, the theoretical analysis and the practical implementation could get more involved.

On the algorithmic side, FLEA as we formulated it, requires computing all pairwise similarities between the sources. This could render it inefficient when the number of sources is very large (e.g. thousands). We expect that it will be possible to overcome this, for example by randomization of the sources, but we leave this step to future work.

## 6    Acknowledgements

The authors would like to thank Bernd Prach, Elias Frantar, Alexandra Peste, Mahdi Nikdan, and Peter Súkeník for their helpful feedback. This research was supported by the Scientific Service Units (SSU) of IST Austria through resources provided by Scientific Computing (SciComp). This publication was made possible by an ETH AI Center postdoctoral fellowship granted to Nikola Konstantinov. Eugenia Iofinova was supported in part by the FWF DK VGSCO, grant agreement number W1260-N35.

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

# Appendix

**Table of Contents**

## A    Experimental setup

### A.1    Dataset preparation

The datasets we use are publicly available and frequently used to evaluate fair classification methods.

The `COMPAS` dataset was introduced by ProPublica. It contains data from the US criminal justice system and was obtained by a public records request. The dataset contains personal information. To mitigate negative side effects, we delete the *name*, *first*, *last* and *dob* (date of birth) entries from the dataset before processing it further. We then exclude entries that do not fit the problem setting of predicting two year recidivism, following the steps of the original analysis.[4] Specifically, this means keeping only cases from Broward county, Florida, for which data has been entered within 30 days of the arrest. Traffic offenses and cases with insufficient information are also excluded. This steps leave 6171 examples out of the original 7214 cases. The categorical features and numerical features that we extract from the data are provided in Table 3a.

`adult`, `germancredit`, and `drugs` are available in the UCI data repository as well as multiple other online sources.[5] We use them in unmodified form, except for binning some of the feature values; see Tables 2 and 3.

Table 2: Dataset information

(a) `adult`

| | | |
|---|---|---|
| dataset size | 48842 | |
| categorical features | *workclass* | federal-gov, local-gov, never-worked, private, self-emp-inc, self-emp-not-inc, state-gov, without-pay, unknown |
| | *education* | 1st-4th, 5th-6th, 7th-8th, 9th, 10th, 11th, 12th, Assoc-acdm, Assoc-voc, Bachelors, Doctorate, HS-grad, Masters, Preschool, Prof-school, Some-college |
| | *hours-per-week* | $\leq 19$, 20–29, 30–39, $\geq 40$ |
| | *age* | $\leq 24$, 25–34, 35–44, 45–54, 55–64, $\geq 65$ |
| | *native-country* | United States, other |
| | *race* | Amer-Indian-Eskimo, Asian-Pac-Islander, Black, White, other |
| numerical features | — | |
| protected attribute | *gender* | values: female (33.2%), male (66.8%) |
| target variable | *income* | $\leq 50K$ (76.1%), $> 50K$ (33.9%) |

---

[4]`https://github.com/propublica/compas-analysis`

[5]`adult:` `https://archive.ics.uci.edu/ml/datasets/adult`,
`germancredit:``https://github.com/praisan/hello-world/blob/master/german_credit_data.csv`,
`drugs:` `https://raw.githubusercontent.com/deepak525/Drug-Consumption/master/drug_consumption.csv`

Table 3: Dataset information (continued)

(a) COMPAS

| dataset size | 6171 (7214 before filtering) | |
|---|---|---|
| categorical features | *c-charge-degree* | values: F (felony), M (misconduct) |
| | *age-cat* | values: <25, 25–45, >45 |
| | *race* | values: African-American, Caucasian, Hispanic, Other |
| numerical features | *priors-count* | |
| protected attribute | *sex* | Female (19.0%), Male (81.0%) |
| target variable | *two-year-recid* | 0 (54.9%), 1 (45.1%) |

(b) drugs

| dataset size | 1885 | |
|---|---|---|
| categorical features | — | |
| numerical features | *Age, Gender, Education, Country, Ethnicity, Nscore, Escore, Oscore, Ascore, Cscore, Impulsive, SS* | (precomputed numeric values in dataset) |
| protected attribute | *Gender* | female (31.0%), male (69.0%) |
| target variable | *Coke* | never used (55.1%), used (44.9%) |

(c) germancredit

| dataset size | 1000 | |
|---|---|---|
| categorical features | *Age* | values: $\leq 24$, 25–34, 35–44, 45–54, 55–64, $\geq 65$ |
| | *Saving accounts* | little, moderate, quite rich, rich |
| | *Checking account* | little, moderate, rich |
| numerical features | *Duration, Credit amount* | |
| protected attribute | *Sex* | female (31.0%), male (69.0%) |
| target variable | *Risk* | bad (30%), good (70%) |

(d) folktables

| dataset size | 255078 | |
|---|---|---|
| categorical features | *AGE* (age; binned) | values: $\leq 14$, 15–24, 25–34, 35–44, 45–54, 55–64, $\geq 65$ |
| | *COW* (class of worker) | values: $1, \ldots, 9$ |
| | *SCHL* (education) | values: $1, \ldots, 24$ |
| | *MAR* (marital status) | values: married, widowed, divorced, separated, never married |
| | *OCCP* (occupation code) | values: $0, 1, \ldots, 9$ |
| | *POBP* (place of birth) | values: USA, other |
| | *RELP* (relationship in household) | values: $0, 1, \ldots, 17$ |
| | *WKHP* (weekly working hours; binned) | values: $\leq 19$, 20-29, 30-39, $\geq 40$ |
| | *RAC1P* (race code) | values: $1, \ldots, 9$ |
| numerical features | — | |
| protected attribute | *SEX* | female (52.1%), male (47.9%) |
| target variable | *income* | $\leq 50K$ (64.8%), $> 50K$ (35.2%) |

For details of the numeric codes, see https://www2.census.gov/programs-surveys/acs/tech_docs/pums/data_dict/PUMS_Data_Dictionary_2018.pdf

## A.2 Training objectives

All training objectives are derived from logistic regression classifiers. For data $S = \{(x_1, y_1), \ldots, (x_n, y_n)\} \subset \mathbb{R}^d \times \{\pm 1\}$ we learn a prediction function $g(x) = w^\top x + b$ by solving

$$\min_{w \in \mathbb{R}^d, b \in \mathbb{R}} \mathcal{L}_S(w, b) + \lambda \|w\|^2 \tag{8}$$

with

$$\mathcal{L}_S(w, b) = \frac{1}{|S|} \sum_{(x,y) \in S} y \log(1 + e^{-g(x)}) + (1 - y) \log(1 + e^{g(x)}) \tag{9}$$

We use the `LogisticRegression` routine of the `sklearn` package for this, which runs a LBFGS optimizer for up to 500 iterations. By default, we do not use a regularizer, i.e. $\lambda = 0$. From $g(x)$ we obtain classification decisions as $f(x) = \operatorname{sign} g(x)$ and probability estimates as $\sigma(x; w, b) = p(y = 1 | x) = \frac{1}{1 + e^{-g(x)}}$, where we clip the output of $g$ to the interval $[-20, 20]$ to avoid numeric issues.

To train with fairness regularization, we solve the optimization problem

$$\min_{w \in \mathbb{R}^d, b \in \mathbb{R}} \mathcal{L}_S(w, b) + \eta |\Gamma_S(w, b)|_\epsilon \tag{10}$$

with

$$\Gamma_S(w, b) = \frac{1}{|S^{a=0}|} \sum_{x \in S^{a=0}} \sigma(x; w, b) - \frac{1}{|S^{a=1}|} \sum_{x \in S^{a=1}} \sigma(x; w, b), \tag{11}$$

where for reasons of numeric stability, we use $|t|_\epsilon = \sqrt{\frac{t^2}{t^2 + \epsilon}}$ with $\epsilon = 10^{-8}$. To do so, we use the `scipy.minimize` routine with `bfgs` optimizer for up to 500 iterations. The necessary gradients are computed automatically using *jax*.[6] To initialize $(w, b)$, we use the result of training a (fairness-unaware) logistic regression with $\lambda = 1$, where the regularization is meant to ensure that the parameters do not take too extreme values. When estimating the disparity, we use the same objective, but with different datasets, $S_1, S_2$ for the two terms in (10), with the protected attributes as target labels for $S_1$, and the inverse of the protected attributes as target labels for $S_2$.

To train with adversarial regularization, we parameterize an adversary $g' : \mathbb{R} \to \mathbb{R}$ as $g'(x') = w'x' + b'$ and solve the optimization problem

$$\min_{w \in \mathbb{R}^d, b \in \mathbb{R}} \max_{w' \in \mathbb{R}, b' \in \mathbb{R}} \mathcal{L}_S(w, b) - \eta \mathcal{L}'_S(w', b') \tag{12}$$

with

$$\mathcal{L}'_S(w, b, w', b') = \frac{1}{|S|} \sum_{(x,a) \in S} a \log(1 + e^{-g'(g(x))}) + (1 - a) \log(1 + e^{g'(g(x))}) \tag{13}$$

To do so, we use the `optax` package with gradient updates by the Adam rule for up to 1000 steps. The learning rates for classifier and adversary are 0.001. The gradients are again computed using *jax*. We initialize $(w, b)$ the same way as for (10). $(w', b')$ we simply initialize with zeros.

To perform score postprocessing, we evaluate the linear prediction function on the training set and determine the thresholds that result in a fraction of $r \in \{0, 0.01, \ldots, 0.99, 1\}$ positive decision separately for each protected group. For each $r$ we then compute the overall accuracy of the classifier that results from using these group-specific thresholds and select the value for $r$ that leads to the highest accuracy. We then modify the classifier to use the corresponding thresholds for each group by adjusting the classifier weights of the protected attributes.

---

[6]`https://github.com/google/jax` (version 0.3.14)

### A.3 Baselines

In this section, we provide more details about the baselines.

**Robust ensemble**  For this baseline, we train $N$ classifiers, one per data source, using the respective base learner. For prediction, we compute the median value of the predicted probabilities and threshold it at 0.5 to obtain a binary label. Since in our experiments the number of sources is always odd, this is also equivalent to classifying using the majority vote rule.

**Filtering method from Konstantinov et al. (2020)**  The method proposed in Konstantinov et al. (2020) uses a filtering step to suppress unreliable sources, like we do, but that differs from FLEA's in two main aspects: it uses only the discrepancy score for its decisions, and its decision criterion is threshold-based, not quantile-based.

For its implementation, one first computes the pairwise discrepancy scores, $\text{disc}(S_i, S_j)$, between all sources. Then, one determines a threshold, $t = \sqrt{\frac{8d \log(2en/d) + 8 \log(8N/\delta)}{n}}$, where $d$ is the VC dimension of the hypothesis class (for us: the dimensionality of the feature vectors plus 1). $\delta$ is a freely choosable confidence parameter. In the limited data regime of our experiments, its value has little influence on the threshold, so we leave it at a default of $\delta = 0.1$. Finally, for each source, $S_i$, we check for how many other sources, $S_j$, their pairwise discrepancy to $S_i$ is less than $t$ (i.e. $\sum_{j \neq i} \mathbb{1}\{\text{disc}(S_i, S_j) < t\}$). If the number of such sources is at least $K - 1$, the source $S_i$ is made part of the overall training set, otherwise is it discarded.

One can check that in the setting of our experiments, only for the `adult` dataset one obtains values for $t$ substantially below 1. Therefore, only for this dataset, the filtering step can have a non-trivial effect.

**DRO method from Wang et al. (2020)**  The DRO method was proposed originally for the *equal opportunity* or *equalized odds* fairness measures. We adapt it to *demographic parity* by imposing constraints on the fraction of positive decisions instead of the true and false positive rates.

Our implementation follows the publicly available github repository,[7] which implements an approximate version of the method described in the publication. The main step is learning a classifier with fairness constraints. This is implemented by deriving a Lagrangian objective and performing simultaneous gradient descent on the classifier parameters and gradient ascent on the Lagrange multipliers. This construction has one hyperparameter, $\xi$, the permitted slack up to which the constraints have to be fulfilled. We set this adaptively, starting with a small value $\xi = 0.01$, but then doubling $\xi$ until the optimization results in a non-degenerate solution (i.e. not a constant classifier).

Additionally, the constraint term of the objective is optimized in a distributionally robust (DRO) way. For this, sample weights are introduced, and the Lagrangian term is maximized also with respect to these weights, subject to $L^1$-ball constraints around uniform weights, and $L^1$-simplex constraints to ensure that the weights encode a discrete probability distribution. Following the original code, we use a projected gradient algorithm for the ball constraint, while the simplex constraint is approximated by implicit renormalization. The DRO also has one hyperparameter, $s$, the radius of the $L^1$-ball. Following the derivation in the original work, we set this to twice the maximal total variation distance between the data distribution of the protected attribute in the original data and in the manipulated data, which in our case is $s = 2(1 - \alpha)$.

Additional hyperparameters are the learning rates for the classifier itself, for the Lagrangian multipliers, and for the sample weights. After some initial sanity checks we keep these at the values that worked best in the original publication, which is 0.01 in all three cases.

**hTERM method from Li et al. (2021a)**  *TERM (tilted empirical risk minimization)* learns a classifier by minimizing an exponentially weighted loss, $\frac{1}{t} \log \left( \frac{1}{|S|} \sum_{(x,y) \in S} e^{t\ell(y, f(x))} \right)$, instead of the standard uniform average of losses over all samples. For negative values of $t$, this expression acts as a *softmin*, thereby encouraging *robustness* in the sense that hard-to-classify outliers will be ignored. For positive values of $t$, the effect is of a *softmax*, which encourages *fairness* in the sense that all loss values should be comparably large.

---

[7]https://github.com/wenshuoguo/robust-fairness-code

For our experiments, we use TERM's hierarchical group-based extension (hTERM): an outer *softmin*-loss encourages robustness across sources, while an inner per-source *softmax*-loss enforces *fairness* across protected groups,

$$\mathcal{L}(f) = \frac{1}{t} \log \big( \frac{1}{N} \sum_{i=1}^{N} n_i e^{t R_i(f)} \big) \quad \text{with} \quad R_i(f) = \frac{1}{\tau} \log \big( \frac{1}{2} \sum_{z \in \{0,1\}} e^{\tau R_i^{a=z}(f)} \big), \tag{14}$$

where

$$R_i^{a=z}(f) = \frac{1}{|S_i^{a=z}|} \sum_{(x,y) \in S_i^{a=z}} \ell(y, f(x)). \tag{15}$$

Following the original manuscript, we use $t = -2$ and $\tau = 2$. To numerically solve the resulting optimization problem, we use the *binary cross-entropy* as the loss function, $\ell$, and we call `sklearn`'s `minimize` routine with LGFBS optimization.

### A.4 Computing resources

All experiments were run on CPU-only compute servers. For each train/test split of each dataset and each adversary, one experimental run across all baseline learning methods takes between 3 minutes and 3 hours on two CPU cores, depending on the number of sources, the size of the data sets, and the CPU architecture. The time needed for each row in the ablation study is similar, except for the `folktables` data, which each took 4-6 hours. The combined time for all reported experiments with linear classifiers (5 datasets, 12 adversaries, 10 train-test splits, 5 base learners) is approximately 1800 core hours. The experiments with nonlinear classifiers required approximately 500 times longer per setting, most of which is spent on cross-validation of the hyperparameters.

For the baselines we are able to reuse many already computed parts. If implemented individually, we'd estimate that the robust ensemble would be the fastest to train, but it is slower than the other methods at prediction time. hTERM would also be efficient to train, as it only requires learning one classifier on the combined training data. The training time for Konstantinov et al. (2020) and the DRO method would be comparable to FLEA's.

### A.5 Hyperparameters

We avoid hyperparameter tuning as far as possible. We do not use $L^2$-regularization (hyperparameter $\lambda$) except to create initializers, where we found the value used to hardly matter. For the fairness-regularizer and fairness-adversary we use fixed values of $\eta = \frac{1}{2}$. We found these to result in generally fair classifiers for unperturbed data without causing classifiers to degenerate (i.e. become constant). Hence we, did not tune these values on a case-by-case basis. When estimating the disparity, we use $\eta = 1$ to be consistent with the theory.

As learning rate for the adversarial fairness training, $\text{lr}_{\text{adv}} = 0.001$ was found by trial and error to ensure convergence at a reasonable speed. Once we identified a reliably working setting, we did not try to tune it further.

### A.6 Adversaries

In this section, we describe the adversaries and their motivation in more detail.

- *flip protected (FP)*: the adversary flips the value of protected attribute.

  This is a straightforward attack on fairness. FP inverts the correlation between the protected attribute and the rest of the data After the sources have been combined, the correlation is therefore weakened, which makes the training data look "less unfair". On the one hand, this can cause fairness-enforcing mechanisms as used, e.g., in postprocessing fairness, to erroneously believe that

little or no compensation for dataset unfairness is required. Consequently, the resulting classifier is actually unfair when applied to future unmanipulated data. On the other hand, it is possible that the training process actually learns to ignore the protected attribute during training, because it is uncorrelated with the target labels. This could make the classifier more fair, e.g. when used with fairness-unaware training.

Our detailed experimental results (Fig. 2 –10) show that both of these effect do, in fact, occur. FP typically increases unfairness when regularization-based or postprocessing-based base learners are used, but it has the opposite effect for the fairness-unaware base learner.

- *flip label (FL)*: the adversary flips the value of the label.

  This is a straightforward attack on accuracy. Following an analog reasoning as above, FL reduces the correlation between the target label and all other data, which makes it harder for the learner to identify a strong classifier.

  Indeed, the experiments shows that the FL adversary often succeeds in reducing the accuracy, while the fairness is relatively unaffected. The adverse effect is small for the large datasets (`adult`, `COMPAS`), and larger for the small ones (`drugs`, `germancredit`), presumably because having more data increases the robustness of the learners against mislabeled data.

- *flip both (FB)*: the adversary flips the value of the protected attribute and the label.

  This attack influences fairness and accuracy at the same time. It preserves the correlation between the protected attribute and the labels, but reduces the correlation between these two and all the other features. Consequently, the learned classifier might rely heavily on the protected attribute to predict the label, which would make it maximally unfair, but potentially also less accurate.

  Our experiments show that this is, indeed, often the observed effect, though the exact amount depends strongly on the dataset and the base learner.

- *shuffle protected (SP)*: the adversary shuffles the protected attribute entries of each batch it modifies, i.e. each example gets assigned the protected attribute of another example that has been chosen at random (without replacement).

  This adversary is similar to FP in that is reduces the overall correlation between the protected attribute and the other data. Its effect is weaker, since it does not explicitly introduce anti-correlation in the manipulated sources. However, its manipulations are less likely to be detected by automatic or manual inspection, since it does not change the marginal statistics of the data, i.e. even after the manipulation, the statistical distribution of each feature dimension, including the protected attribute, is the same as for clean sources.

  In experimental results, SP indeed performs similarly to FP for the fairness-aware base learners, and its effect are somewhat weaker for the fairness-unaware base learner.

- *overwrite protected (OP)*: the adversary overwrites the protected attribute of each sample in the affected batch by its label.

  This manipulation creates a strong artificial correlation between the protected attribute and the target label. In fact, the maximally unfair classifier that predicts the label directly from the protected attribute will have perfect accuracy on the manipulated data, and still a much higher accuracy than what would be correct on the overall training data. Consequently, the learned classifier might make strong use of the protected attribute, which leads to unfair and potentially incorrect decisions on clean data.

  Our experiments show that OP indeed often leads to large increases in unfairness. However, there are also cases where the unfairness is actually reduced, but then typically this is accompanied by loss of accuracy.

- *overwrite label (OL)*: the adversary overwrites the label of each sample in the affected batch by its protected attribute.

Like the OP adversary, this manipulation leads to a perfect correlation between the target labels and the protected attributes. However, it achieves this without changing the marginal distribution of the protected attribute, instead influencing the statistics of the labels. Depending on the specific situation, it might be easier or harder to detect from automatic or manual inspection. OL is also more likely to negatively affect the accuracy, since the classifier will try to predict incorrect labels.

The experiments show that OL indeed almost always reduces the accuracy, while at the same time often increasing unfairness.

- *resample protected (RP)*: the adversary samples new batches of data in the following ways: all original samples of protected group $a = 0$ with labels $y = 1$ are replaced by data samples from other sources which also have $a = 0$, but $y = 0$. Analogously, all samples of group $a = 1$ with labels $y = 0$ are replaced by data samples from other sources with $a = 1$ and $y = 1$.

  Like OL and OP, RP results in a perfect correlation between protected attributes and labels, thereby facilitating unfairness and reducing accuracy. It does so in a more subtle and harder-to-detect way, however, as it achieves the effect using original data samples.

  Indeed, in our experimental results RP influences fairness and accuracy in similar ways as the other two methods.

- *random anchor (RA0/RA1)*: these adversaries follow the protocol introduced in Mehrabi et al. (2021b). RA0 first picks a random *anchor* example $x_{\text{target}}^-$ of group $a = 1$ with label $y = 0$ from the target source. It then creates a group, $\mathcal{G}_+$, of poisoned data by constructed new examples within a feasible set that also have $a = 1$ and are close to $x_{\text{target}}^-$, but that have label $y = 1$. The number of samples in $\mathcal{G}_+$ matches the number of samples in the target source with $a = 1$. Subsequently, the adversary repeats the above procedure for group $a = 0$, but with the opposite label values, resulting in a second group of poisoned samples $\mathcal{G}_-$. Both poisoned sets are then merged to yield a manipulated source that is meant to influence the decision boundary near the anchor points in a maximally unfair way. The adversary RA1 performs the same construction as RA0, but with the roles of $a = 0$ and $a = 1$ exchanged.

  Given that our data sources mostly have categorical features, it is not possible to create realistic-looking new samples simply by small random perturbations. Instead, we define as feasible set the set of all samples that occur in any of the original training sources. As newly 'constructed' samples we then take those examples with smallest Euclidean distance to the anchors.

- *random (RND)*: the adversary randomly picks one of the strategies above (except ID) for each source.

  This adversary reflects the observation that different sources might be manipulated in different ways. One reason for this could be that in a real-world system, multiple adversaries exists who manipulate individual data sources without coordinating their actions. Alternatively, there might be just one adversary who manipulates all sources, but chooses to manipulate them in different ways, e.g. to avoid easy detection.

  The experimental results show that this strategy does, indeed, work to some extent, with RND often having an effect where some of the other methods do not, but the effect is weaker.

- *identity (ID)*: the adversary makes no changes to the data.

  The ID adversary serves as a useful check that FLEA does not damage the learning process in the case that all data is actually clean. It also reflects the fact that even though the adversary has the power to manipulate the data it does not have to. Ideally, the learning method will notice this and achieve even better results in presence of the ID adversary than for the *oracle*.

  In the experimental results, this is effect is only rarely visible for any method, though.

Note that even though we introduced the adversaries above as intentional manipulations, many of them could also occur accidentally when data from different sources is collected, e.g. as problems during data entering or numeric encoding.

## B    Detailed algorithm for estimating disc **and** disp

In this section we provide pseudocode for estimating the pairwise discrepancy and disparity between two data sources. In both cases we approximate a classifier $f$ that maximizes a continuous relaxation of the relevant metric, and then estimate the actual quantity of interest from it. The disc-maximizing classifier is trained by flipping the labels of one of the two sources, combining the sources into a single dataset, and then training a classifier to predict the (new) label. The disp-maximizing classifier is trained as a classifier that comes as close as possible to predicting the protected attribute $a$ on one data source, and $1 - a$ on the other, while balancing the loss from each subgroup.

---

EMPIRICAL DISCREPANCY ESTIMATION

**Input:** datasets $S_1, S_2$

1:   $f \leftarrow \min_f \left( \frac{1}{|S_1|} \sum\limits_{(x,y) \in S_1} \text{CROSSENTROPY}(f(x), y) + \frac{1}{|S_2|} \sum\limits_{(x,y) \in S_2} \text{CROSSENTROPY}(f(x), (1-y)) \right)$

2:   $\text{disc} \leftarrow \left| \frac{1}{|S_1|} \sum\limits_{x \in S_1} \mathbb{1}\{(f(x) \geq 0.5) \neq y\} - \frac{1}{|S_2|} \sum\limits_{x \in S_2} \mathbb{1}\{(f(x) \geq 0.5) = y\} \right|$

**Output:** Empirical discrepancy estimate disc $\in \mathbb{R}^+$

---

EMPIRICAL DISPARITY ESTIMATION (Demographic Parity)

**Input:** datasets $S_1, S_2$

1:   $f \leftarrow \min_f \left( \frac{1}{|S_1^{a=0}|} \sum\limits_{x \in S_1^{a=0}} \text{CROSSENTROPY}(f(x), 0) + \frac{1}{|S_1^{a=1}|} \sum\limits_{x \in S_1^{a=1}} \text{CROSSENTROPY}(f(x), 1) \right.$

$\left. + \frac{1}{|S_2^{a=0}|} \sum\limits_{x \in S_2^{a=0}} \text{CROSSENTROPY}(f(x), 1) + \frac{1}{|S_2^{a=1}|} \sum\limits_{x \in S_2^{a=1}} \text{CROSSENTROPY}(f(x), 0) \right)$

2:   $\text{disp} \leftarrow \left| \frac{1}{|S_1^{a=0}|} \sum\limits_{x \in S_1^{a=0}} \mathbb{1}\{(f(x) \geq 0.5) \neq y\} - \frac{1}{|S_1^{a=1}|} \sum\limits_{x \in S_1^{a=1}} \mathbb{1}\{(f(x) \geq 0.5) \neq y\} \right.$

$\left. - \frac{1}{|S_2^{a=0}|} \sum\limits_{x \in S_2^{a=0}} \mathbb{1}\{(f(x) \geq 0.5) \neq y\} + \frac{1}{|S_2^{a=1}|} \sum\limits_{x \in S_2^{a=1}} \mathbb{1}\{(f(x) \geq 0.5) \neq y\} \right|$

**Output:** Empirical discrepancy estimate disp $\in \mathbb{R}^+$

---

## C    Detailed experimental results

In addition to the experiments with a regularization-based base learner that were reported in the main manuscript, we also run experiments with postprocessing-based fairness, preprocessing-based fairness, adversarial fairness, and fairness-unaware learning. The results are depicted in Fig. 2–5 for the homogeneous setting and in Fig. 6–10 for the heterogeneous setting. Also included are results for two of the baselines, robust ensemble and Konstantinov et al. (2020). The DRO (Wang et al., 2020) and hTERM (Li et al., 2021a) require specific learning procedures and therefore cannot be combined with arbitrary base learners. We report them together with results for the regularization-based base learners.

The format of the figures is as follows: for each datasets and method, we report the accuracy and fairness results for different adversaries. Each panel contains 12 bars. The left-most one in each diagram shows the result of the hypothetical *oracle* setting, where the learning algorithm trains only on the clean data sources, i.e. the ones which the adversary cannot modify. The remaining bars correspond to the outcome when different adversaries have perturbed the data. An ideal robust method should achieve results approximately as good as the oracle result, as this would indicate that the adversary was indeed not able to negatively affect the learning process.

From the results, one can see that FLEA works almost perfectly in the homogeneous setting with a lot of data (`adult`) and still quite well when the amount of data is limited (`COMPAS`, `drugs` and `germancredit`data). In the latter cases, for some adversaries FLEA does not always exactly match the *oracle* results, but it still performs better than the baselines. In the heterogeneous case, FLEA works reliably in all settings, except for the fairness measure when $N - K = 25$, as we had already discussed in the main manuscript. The results also

show that different base learners achieve different accuracy/fairness trade-offs, but FLEA is effective with each of them. In a few cases, FLEA's results appear to even improve over the ones of the oracle. However, we do not believe this to be a systematic effect, but rather a case in which the adversarial perturbation were largely benign, and FLEA chooses a subset of sources that by random chance yields a better classifier than when using exactly the clean sources.

Figure 2: `adult` dataset, $N = 5, N - K = 2$

(a) regularization-based fairness

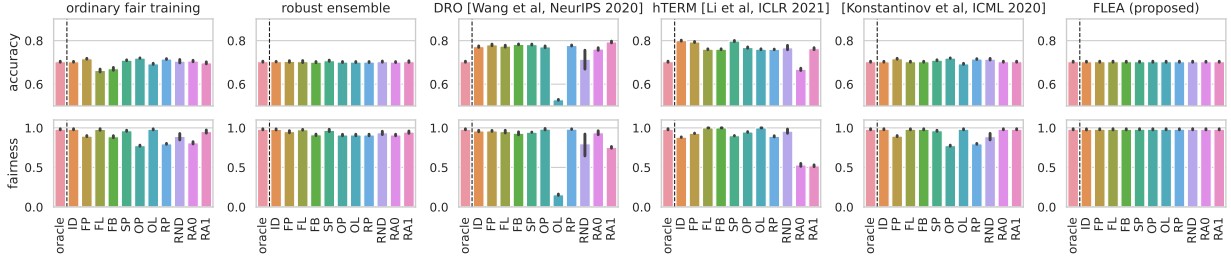

(b) preprocessing-based fairness

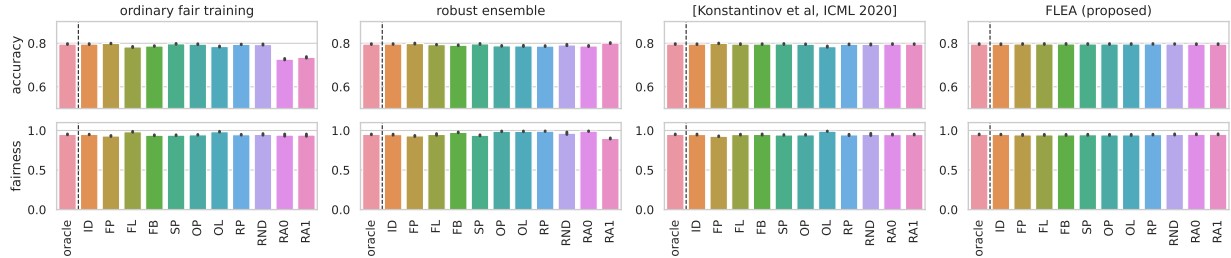

(c) postprocessing-based fairness

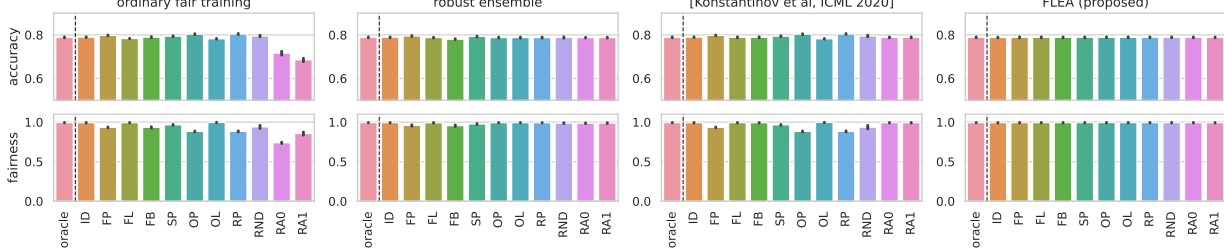

(d) adversarial fairness

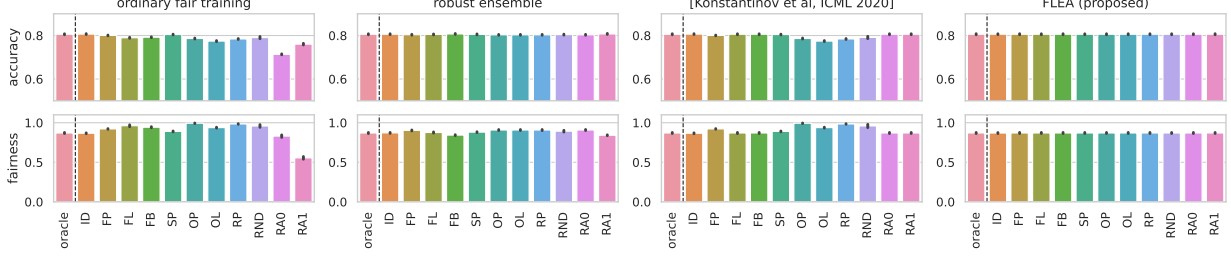

(e) fairness-unaware

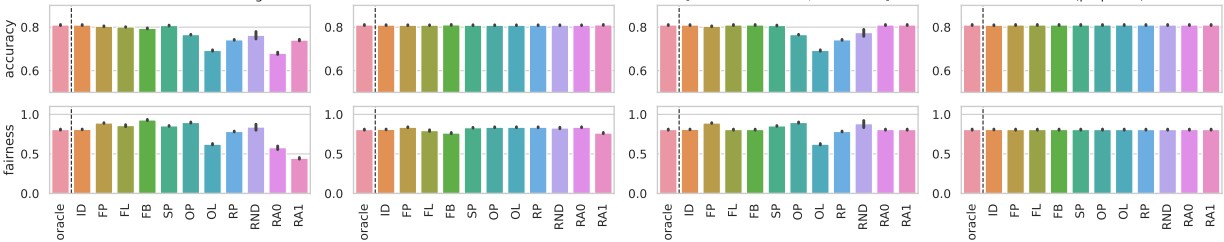

Figure 3: `COMPAS` dataset, $N = 5, N - K = 2$

(a) regularization-based fairness

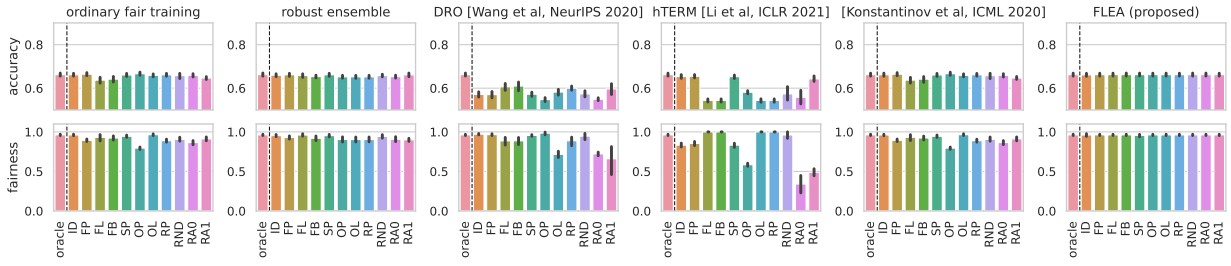

(b) preprocessing-based fairness

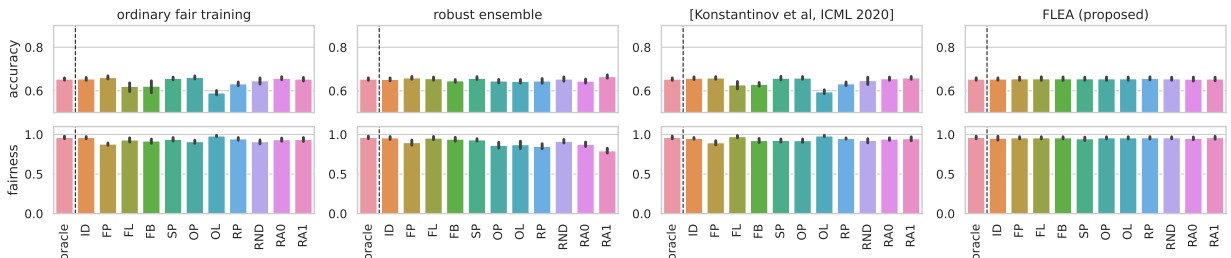

(c) postprocessing-based fairness

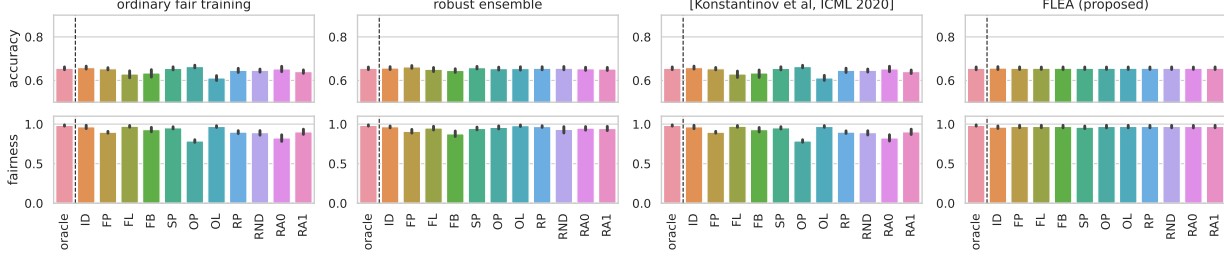

(d) adversarial fairness

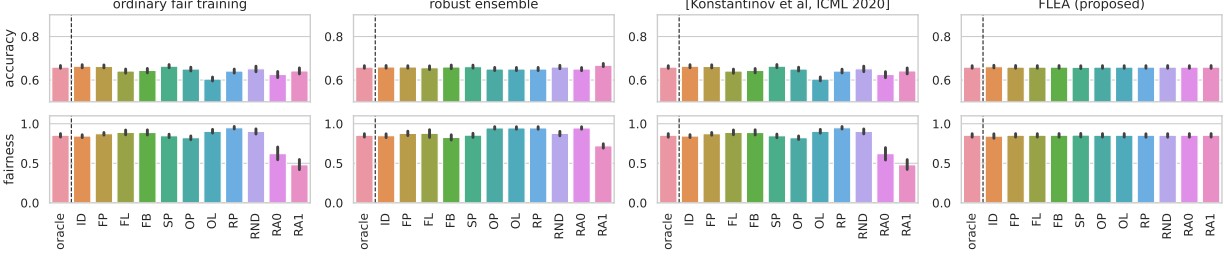

(e) fairness-unaware

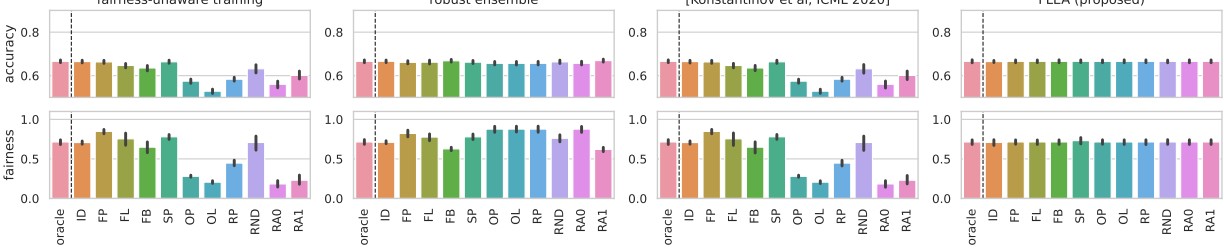

Figure 4: `drugs` dataset, $N = 5, N - K = 2$

(a) regularization-based fairness

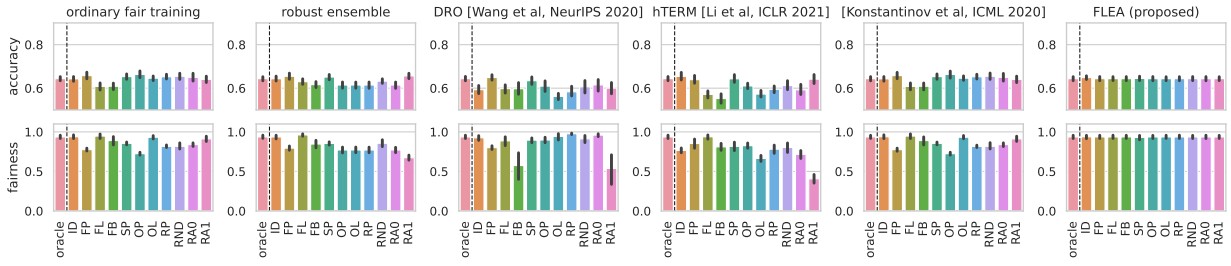

(b) preprocessing-based fairness

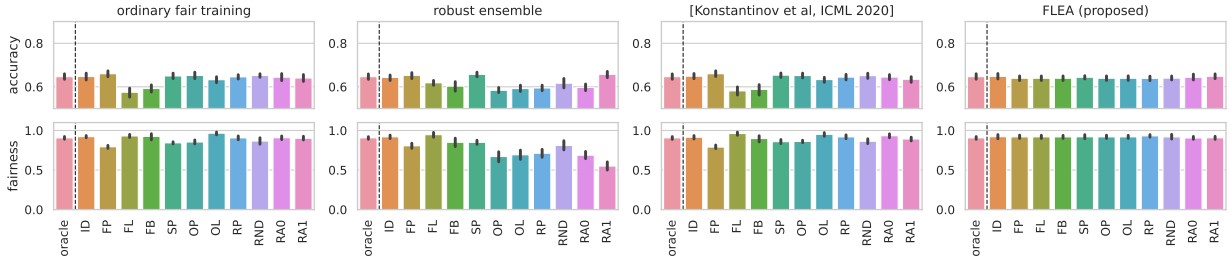

(c) postprocessing-based fairness

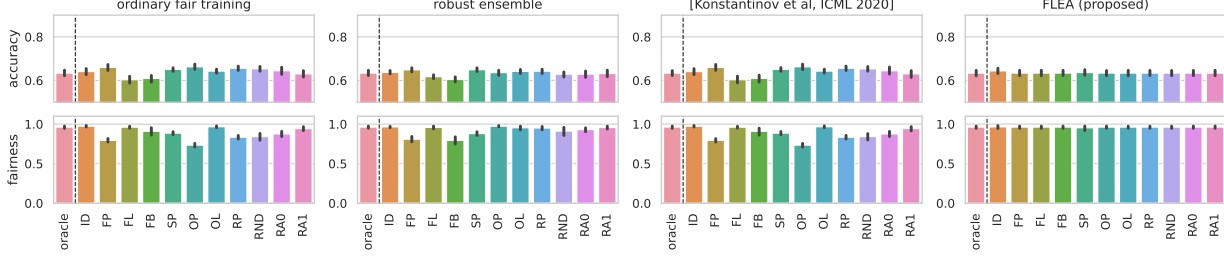

(d) adversarial fairness

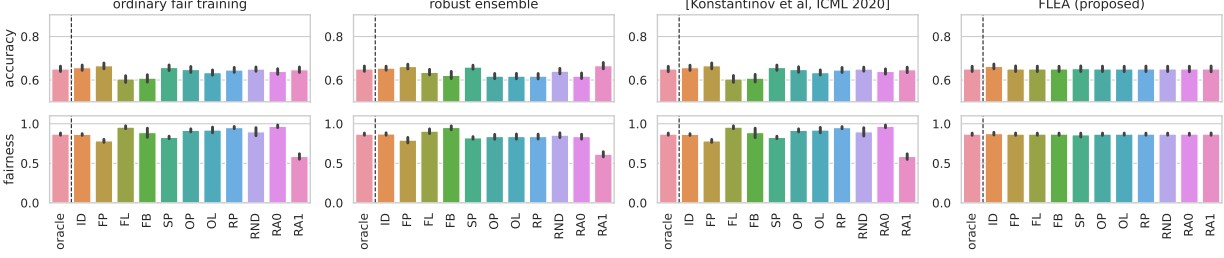

(e) fairness-unaware

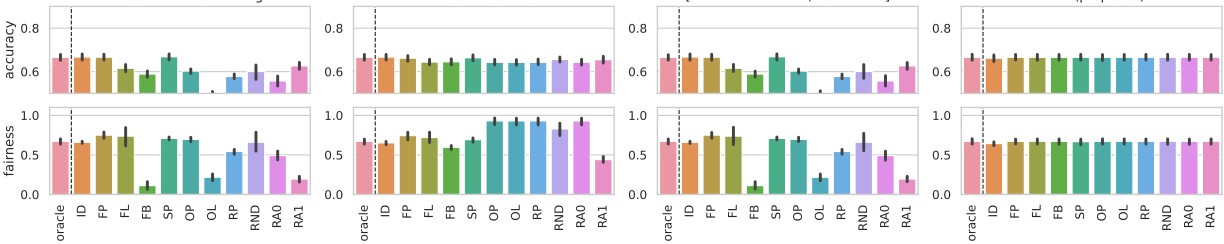

Figure 5: `germancredit` dataset, $N = 5, N - K = 2$

(a) regularization-based fairness

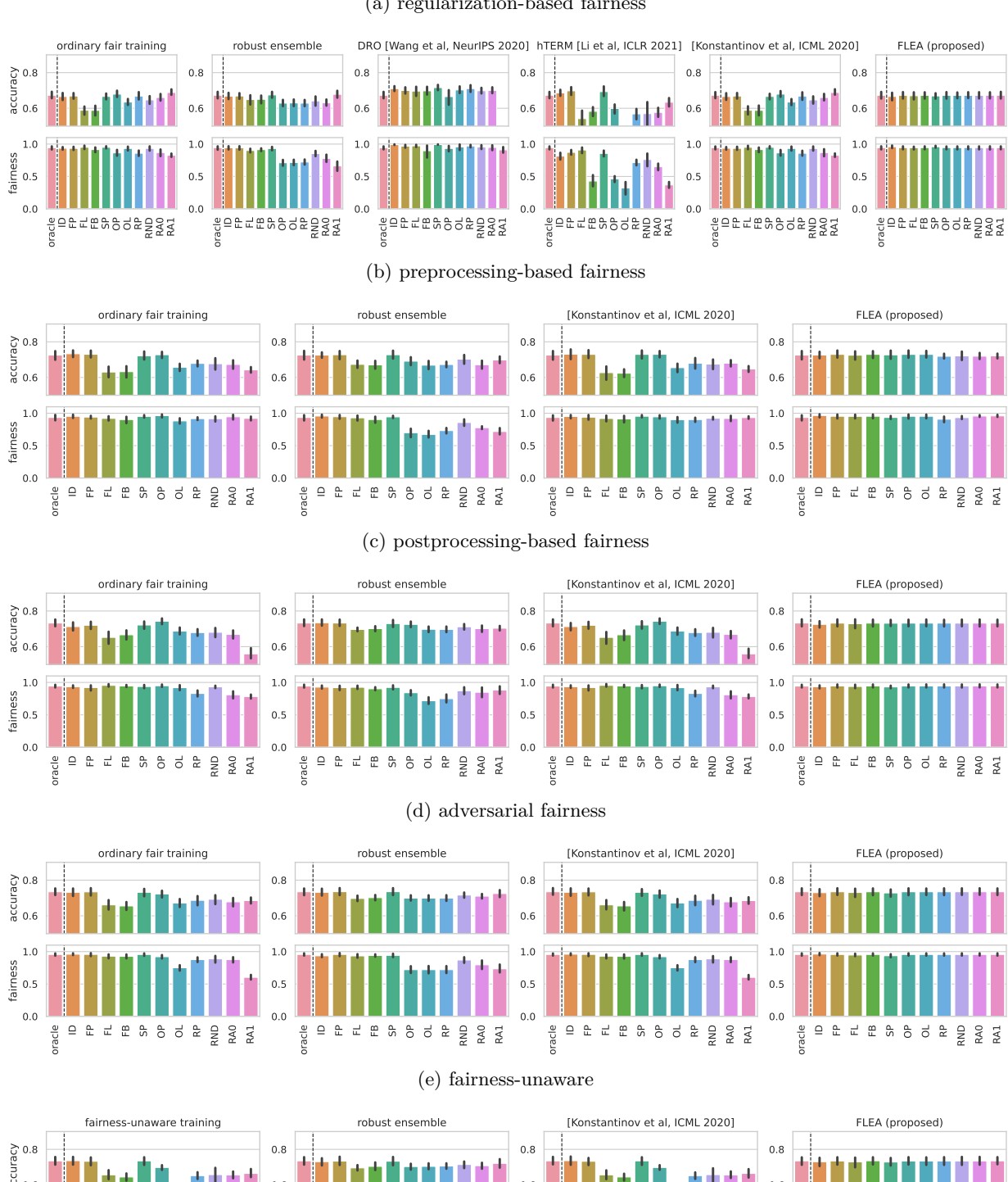

(b) preprocessing-based fairness

(c) postprocessing-based fairness

(d) adversarial fairness

(e) fairness-unaware

Figure 6: `folktables` dataset, regularization-based fairness

(a) $N = 51, N - K = 5$

(b) $N = 51, N - K = 10$

(c) $N = 51, N - K = 15$

(d) $N = 51, N - K = 20$

(e) $N = 51, N - K = 25$

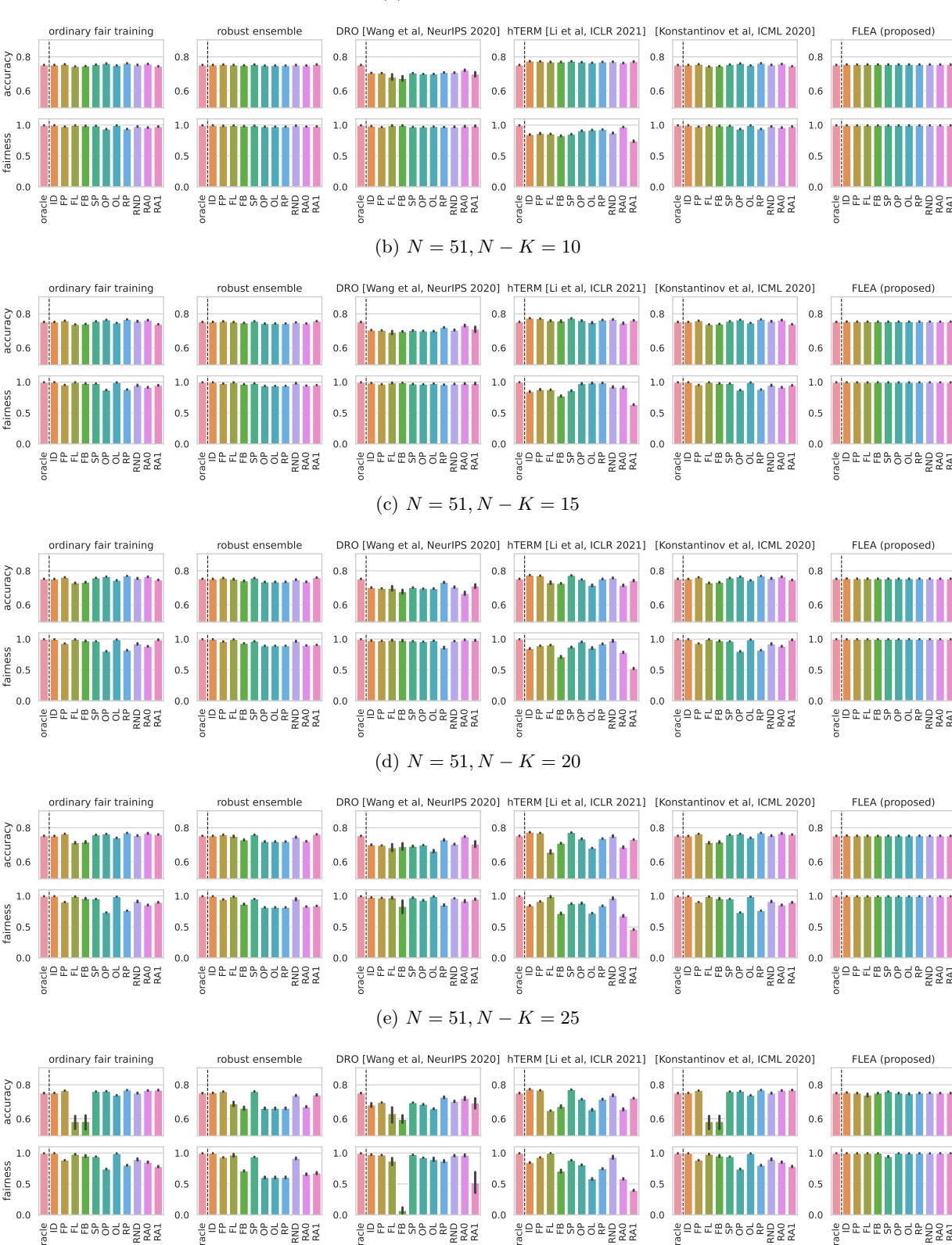

Figure 7: `folktables` dataset, preprocessing-based fairness

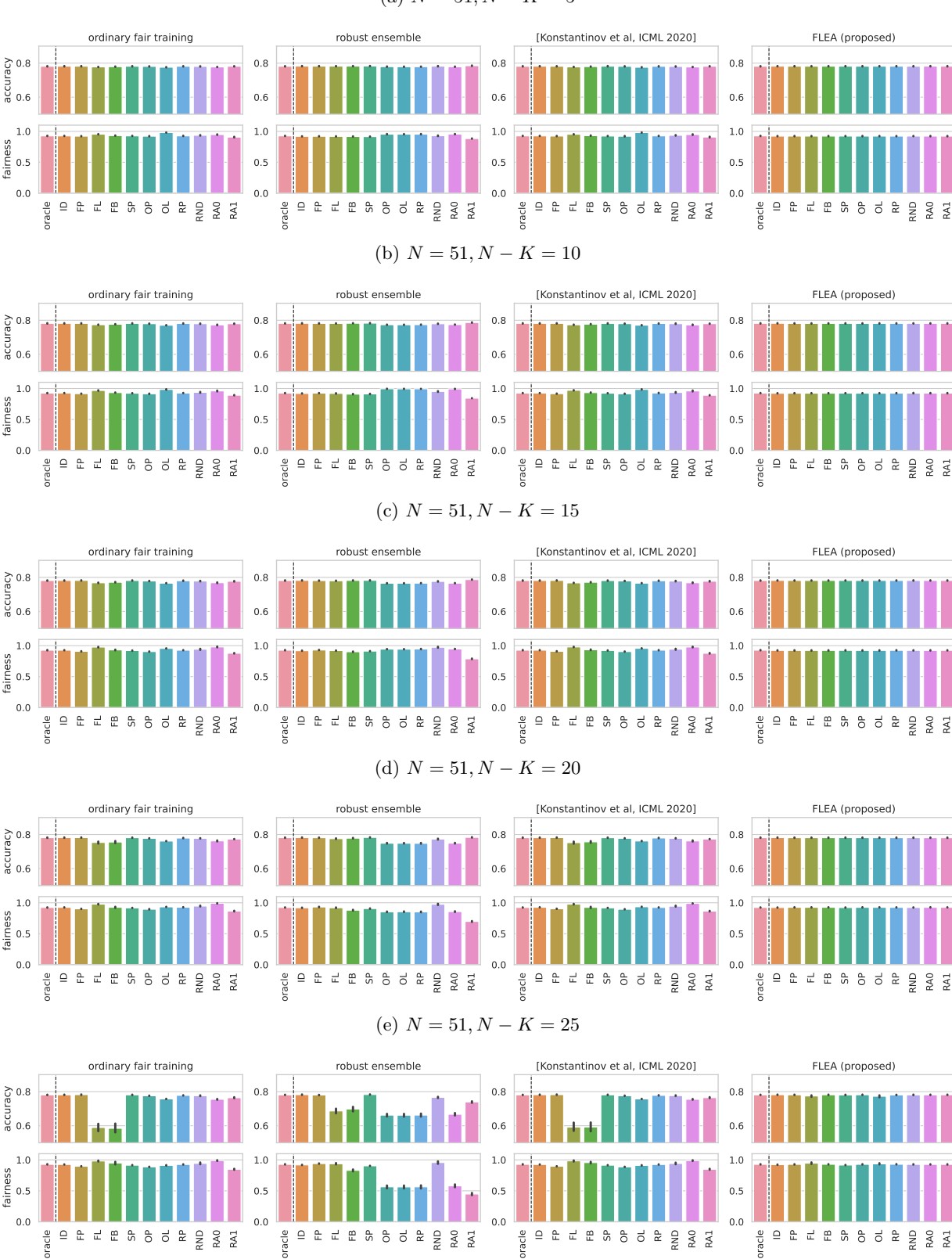

Figure 8: `folktables` dataset, postprocessing-based fairness

(a) $N = 51, N - K = 5$

(b) $N = 51, N - K = 10$

(c) $N = 51, N - K = 15$

(d) $N = 51, N - K = 20$

(e) $N = 51, N - K = 25$

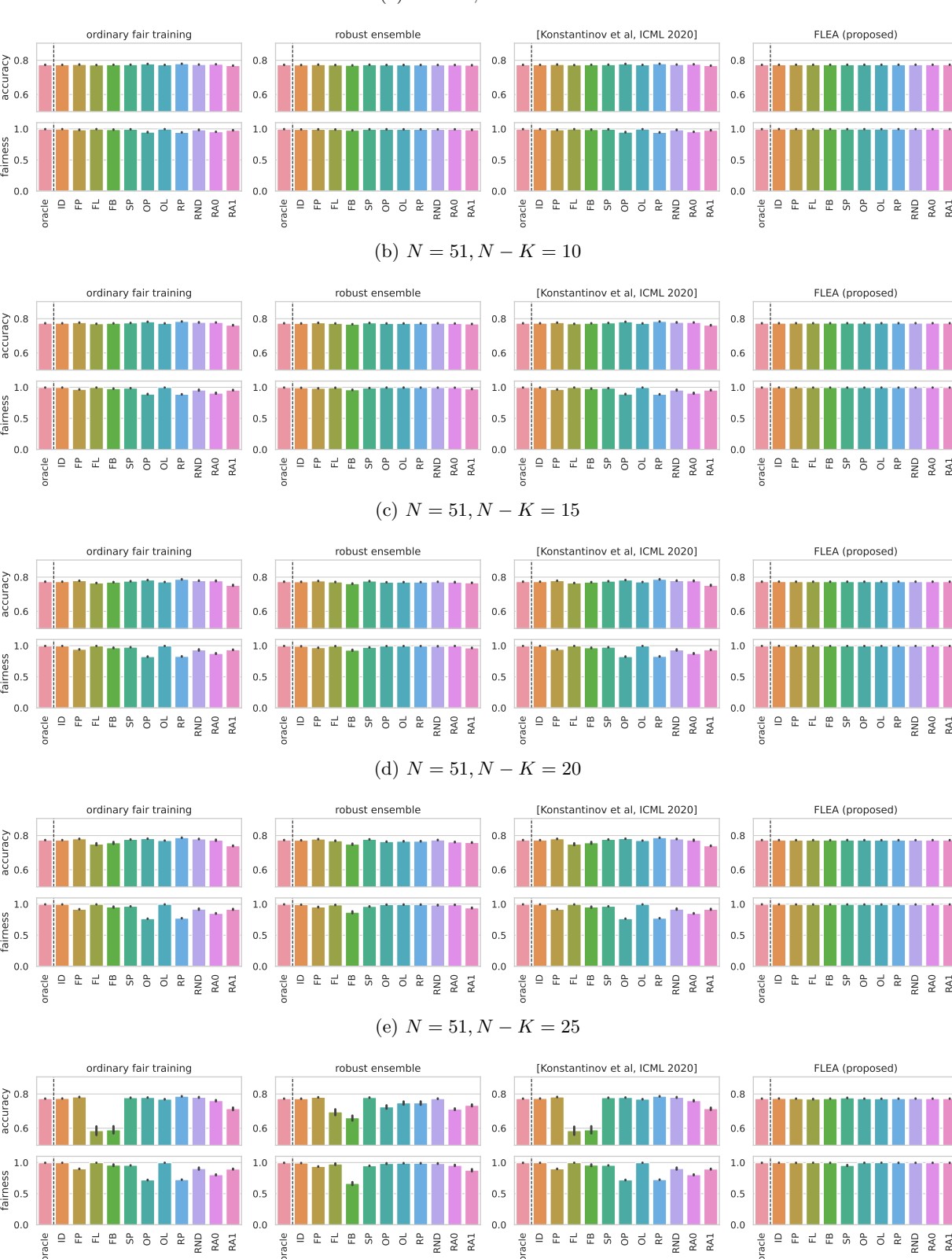

Figure 9: `folktables` dataset, adversarial fairness

(a) $N = 51, N - K = 5$

(b) $N = 51, N - K = 10$

(c) $N = 51, N - K = 15$

(d) $N = 51, N - K = 20$

(e) $N = 51, N - K = 25$

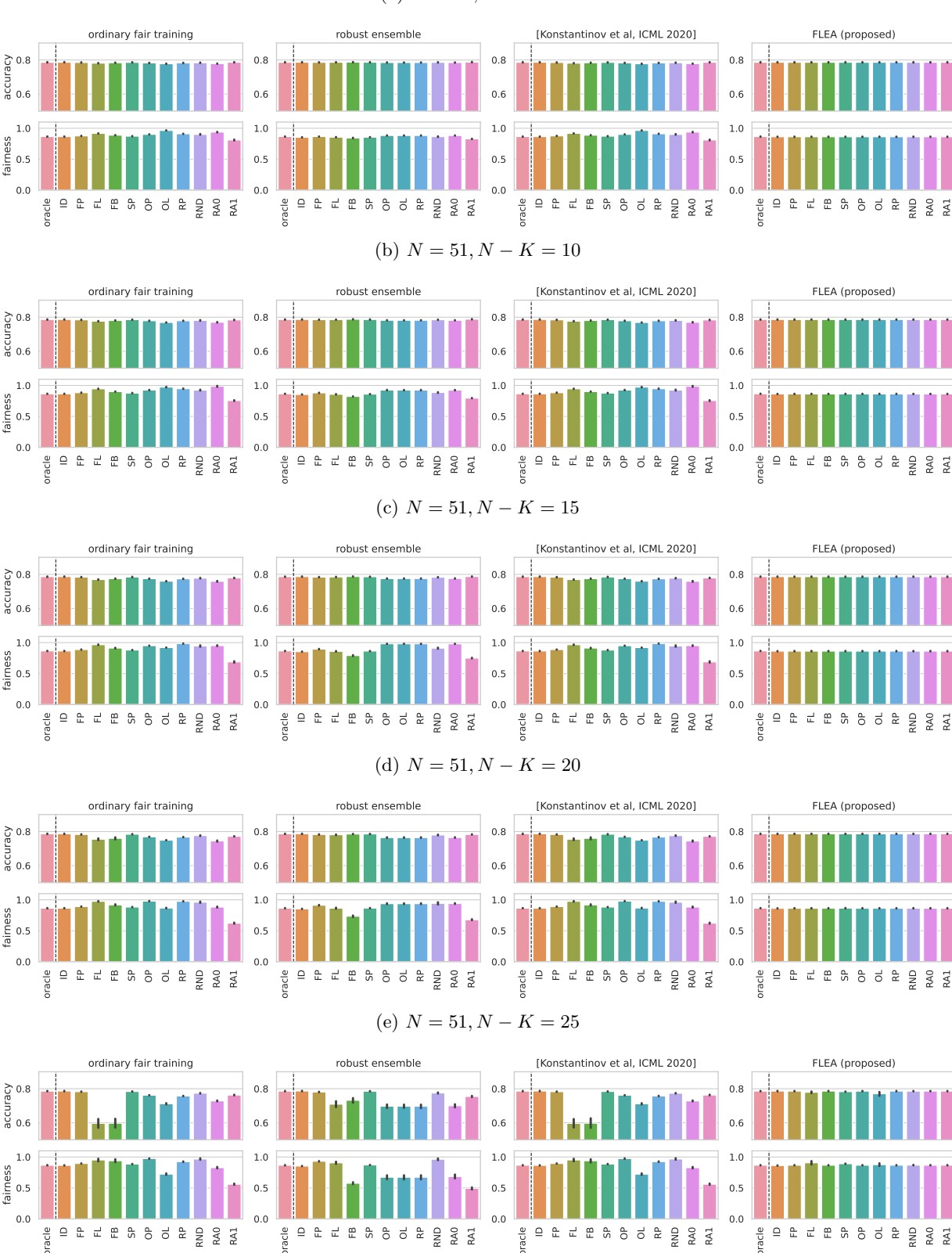

Figure 10: `folktables` dataset, fairness-unaware

(a) $N = 51, N - K = 5$

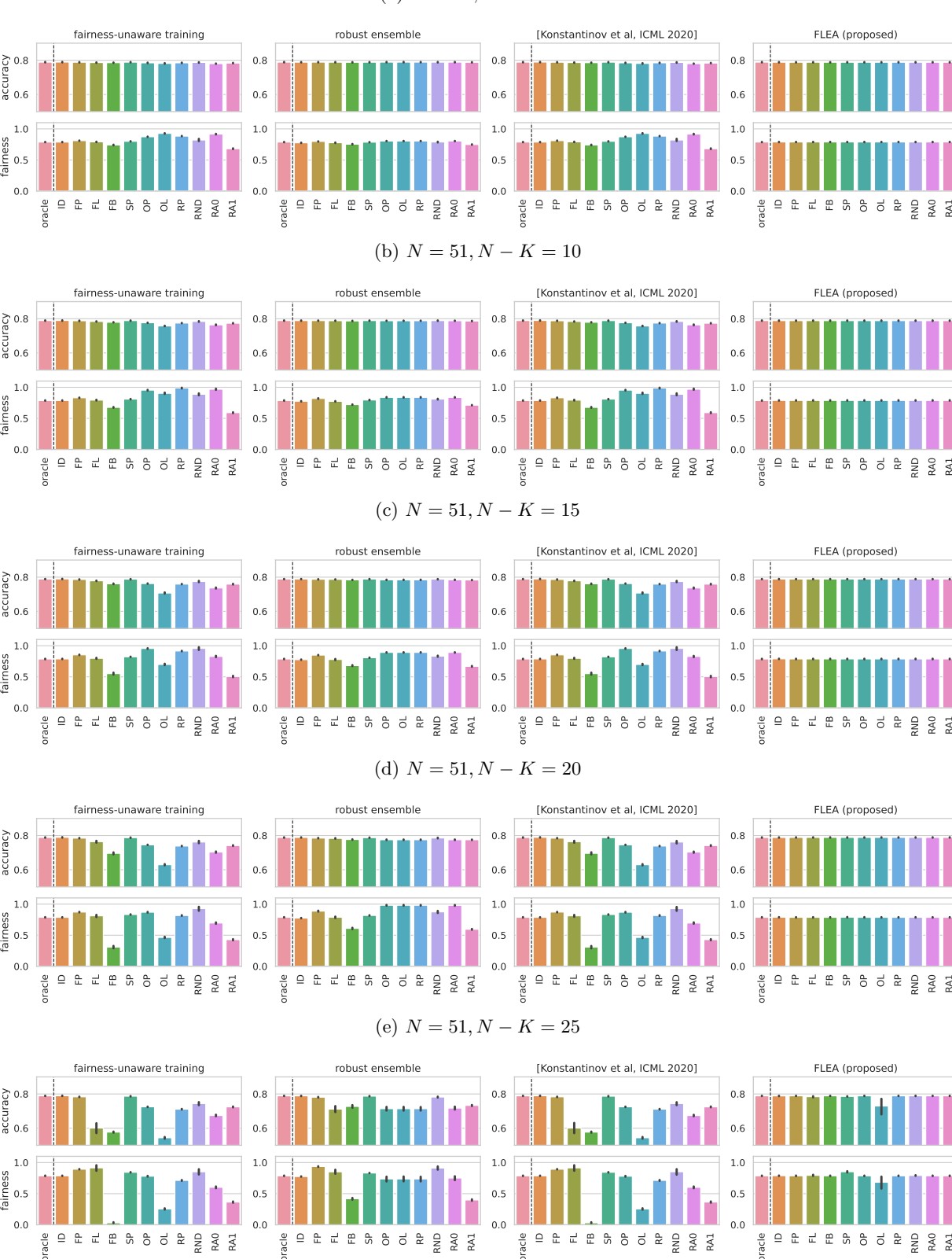

(b) $N = 51, N - K = 10$

(c) $N = 51, N - K = 15$

(d) $N = 51, N - K = 20$

(e) $N = 51, N - K = 25$

Table 4: Results of FLEA and baselines for robust fairness-aware multisource learning using preprocessing fairness and a non-linear (gradient boosted decision trees) classifer. See Table 1 for an explanation of setting and entries.

(a) `adult`, `COMPAS`, `drugs` and `germancredit` datasets with 5 sources of which 2 are unreliable.

| method | adult | | COMPAS | | drugs | | germancredit | |
|---|---|---|---|---|---|---|---|---|
| | accuracy | fairness | accuracy | fairness | accuracy | fairness | accuracy | fairness |
| naive | $66.2_{\pm 0.7}$ | $86.9_{\pm 1.8}$ | $55.6_{\pm 2.6}$ | $83.3_{\pm 4.7}$ | $60.0_{\pm 2.5}$ | $77.8_{\pm 3.7}$ | $53.9_{\pm 2.6}$ | $76.2_{\pm 3.8}$ |
| robust ensemble | $74.4_{\pm 0.7}$ | $71.8_{\pm 2.3}$ | $56.3_{\pm 2.5}$ | $41.2_{\pm 5.7}$ | $54.9_{\pm 2.3}$ | $36.5_{\pm 5.2}$ | $58.8_{\pm 1.8}$ | $48.0_{\pm 4.1}$ |
| (Konstantinov et al., 2020) | $77.5_{\pm 0.4}$ | $90.9_{\pm 1.7}$ | $56.2_{\pm 3.1}$ | $81.8_{\pm 6.3}$ | $52.7_{\pm 1.6}$ | $79.4_{\pm 2.2}$ | $53.3_{\pm 3.0}$ | $79.0_{\pm 4.1}$ |
| FLEA (proposed) | $77.9_{\pm 0.4}$ | $92.1_{\pm 1.1}$ | $62.3_{\pm 0.9}$ | $87.8_{\pm 5.4}$ | $62.8_{\pm 1.6}$ | $80.6_{\pm 3.3}$ | $64.5_{\pm 6.6}$ | $84.4_{\pm 3.7}$ |
| oracle | $78.5_{\pm 0.4}$ | $93.3_{\pm 1.5}$ | $64.8_{\pm 1.4}$ | $93.0_{\pm 5.9}$ | $64.1_{\pm 2.1}$ | $88.1_{\pm 4.6}$ | $69.0_{\pm 3.8}$ | $92.8_{\pm 4.7}$ |

# D  Additional Results

While our experimental evaluation in this work focussed on the setting of linear classification, FLEA is also applicable in combination with nonlinear classifiers. However, this leads to increased computational cost, and also the size of the training sources would have to be larger to effectively estimate the disc and disp measures. A compromise is to perform FLEA's filtering step with respect to linear classifiers but afterwards train a non-linear classifier on the resulting combined training set. We observed this setup to work well in practice, even though the theoretical guarantees do not hold.

As exemplary setting, we perform experiments in a subset of the situations using *gradient boosted decision trees* Friedman (2001) from the `xgboost` package[8] as nonlinear classifiers. We use 5-fold crossvalidation to select hyperparameters `n_estimators` $\in \{100, 200, 300, 400, 500\}$ and `max_depth` $\in \{3, 5, 7, 9\}$. To encourage fairness of the resulting classifiers we use the *preprocessing* approach, as that requires no changes to the actual classifier training routines. As baselines, we compare to the robust ensemble and the filtering approach of Konstantinov et al. (2020). The other baselines of Section 4.2 are not applicable, as they require modifications of the training process itself.

Table 4 reports the results in tabular form. Figures 11 and Figure 12 visualize the results for each adversary.

Overall, one can see the same trend as in the linear setting: naively merging the data sources leads to strong decreases in accuracy and fairness. The different robust methods overcome this to varying degrees, with FLEA always achieving the best results, i.e. closest to the hypothetical *oracle*.

A comparison of Table 4 to Table 1 shows that the use of a nonlinear instead of linear classifier generally does not lead to more accurate nor more fair classifiers in the tested setting. Presumably, this is because in the chosen categorical representation, the bottleneck for prediction quality is not a lack of expressibility of the hypothesis class, but rather the datasets' intrinsic noise. This view is supported by the fact that accuracy and fairness are not much increased even for the *oracle* approach, which is not affected by the data manipulations.

# E  Discussion of the role of disb, disc and disp and ablation study

FLEA relies on the combination of three dissimilarity measures: disc, disp and disb. In these section we discuss the importance of each of them and report on an ablation study to verify their practical significance.

Of the three measures, disc is indispensable to ensure classifier accuracy, as it is the only measure that depends on the label values. At the same time, disc is blind to changes in the protected attributes whenever those are not part of the feature set.

Even in the case when the protected attribute is among the features, the disc measure may not detect changes in the data that may harm fairness. For example, if one of the protected groups is much more rare than the other, changing even a small number of data points (e.g. the points from that group) can cause a large change in the conditional distributions of the data given the value of the protected attribute. At the same time, the discrepancy will remain largely unaffected, since only a few points have been changed in total.

---

[8]`https://xgboost.readthedocs.io/en/stable/python/index.html`

Figure 11: Nonlinear (gradient-boosted decision trees) classifier, preprocessing-based fairness

(a) `adult` ($N = 5, N - K = 2$)

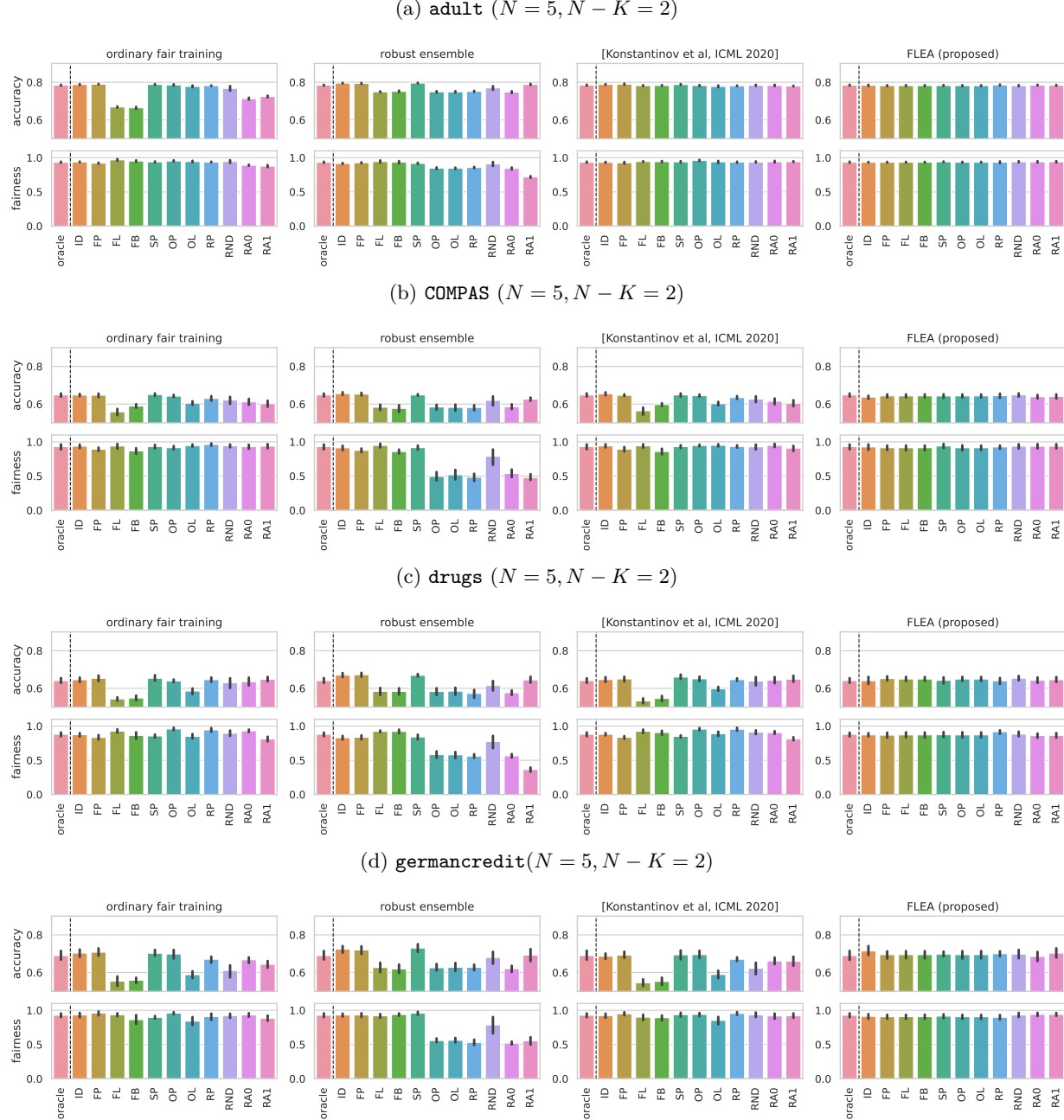

(b) `COMPAS` ($N = 5, N - K = 2$)

(c) `drugs` ($N = 5, N - K = 2$)

(d) `germancredit`($N = 5, N - K = 2$)

Therefore, filtering only based on disc is insufficient - sources with a different conditional distribution may get in through the filtering step, potentially causing unfair classifiers (on clean data) to appear fair on the (corrupted) dataset.

FLEA avoids such issues by additionally adopting the disp measure, which can reliably detect changes in the conditional distributions of the data given the value of the protected attribute, thereby ensuring reliable fairness estimates based on the sources that are returned by the filtering procedure. However, disb is not sensitive to manipulations in the size of the protected attributes, for example, an adversary who selectively drops examples of one protected group. However, the disb measure would detect such a manipulation. Thereby, it ensures that the disparity of the union of multiple sources is close to their average individual

Figure 12: Nonlinear (gradient-boosted decision trees) classifier, fairness-unaware learning

(a) `adult` $(N = 5, N - K = 2)$

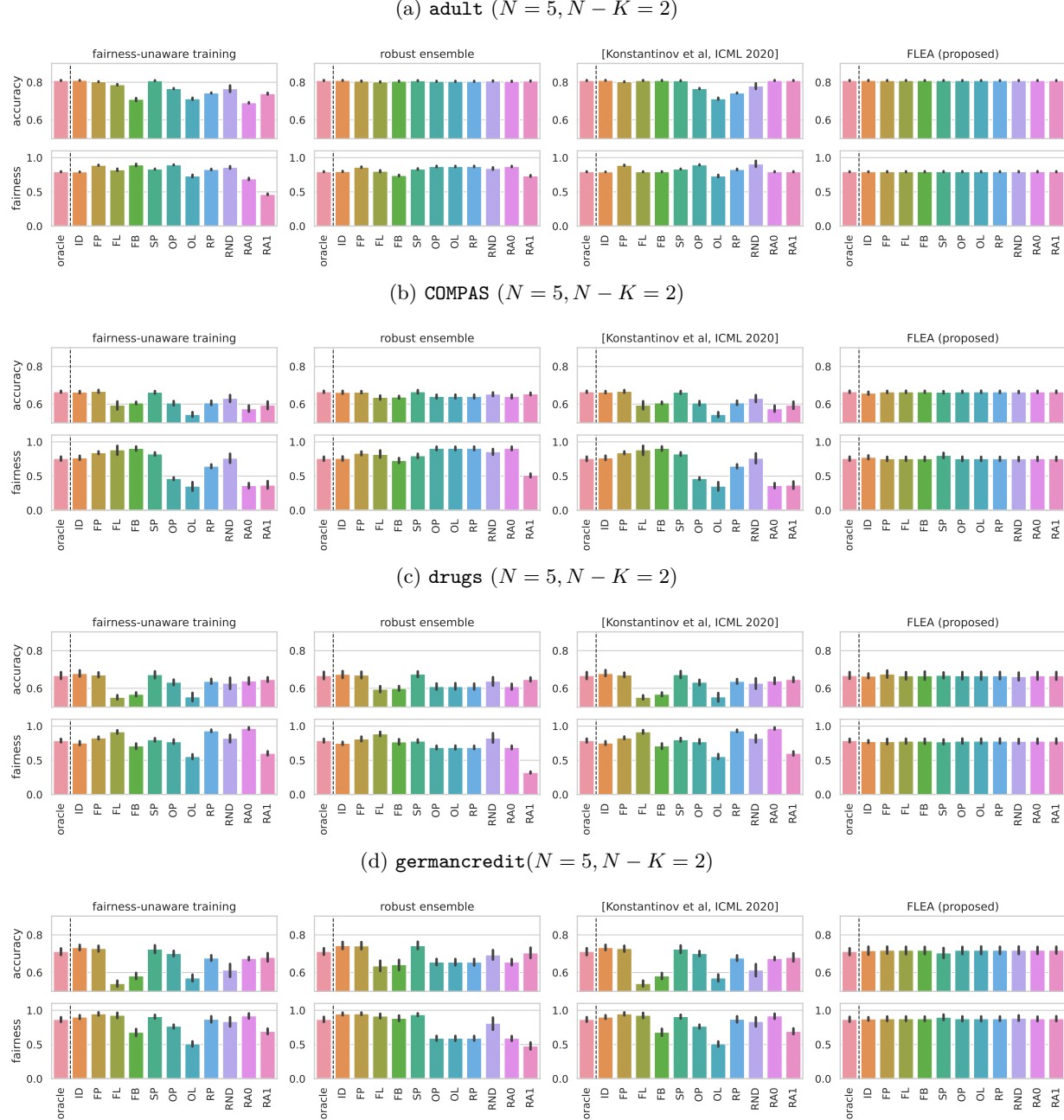

(b) `COMPAS` $(N = 5, N - K = 2)$

(c) `drugs` $(N = 5, N - K = 2)$

(d) `germancredit`$(N = 5, N - K = 2)$

disparity. This is important because a fairness-aware learner works on top of FilterSources by merging the data from the sources returned by the filtering algorithm. This aspect becomes apparent only in the proof of Theorem 1, see Section F.

### E.1 Ablation study

To understand the respective contributions of disb, disc, and disp on real-world data, we performed an ablation study that runs variants of FLEA in which any subsets of the three measures are used to compute

Table 5: Performance of FLEA with different combination of disb, disc, and disp activated or deactived (crossed-out). Reported results are in the same format at Table 1: minimal accuracy (A) and fairness (F) against any of the tested adversaries

(a) regularization-based fairness

|  |  | disb
disc
disp | disb
disc
~~disp~~ | disb
~~disc~~
disp | ~~disb~~
disc
disp | disb
~~disc~~
~~disp~~ | ~~disb~~
disc
~~disp~~ | ~~disb~~
~~disc~~
disp | ~~disb~~
~~disc~~
~~disp~~ |
|---|---|---|---|---|---|---|---|---|---|
| adult | A: | $70.3_{\pm0.4}$ | $70.3_{\pm0.4}$ | $63.7_{\pm9.9}$ | $70.3_{\pm0.4}$ | $53.3_{\pm16.6}$ | $70.3_{\pm0.4}$ | $64.3_{\pm10.1}$ | $57.4_{\pm16.5}$ |
| $N=5, N-K=2$ | F: | $98.2_{\pm1.1}$ | $98.2_{\pm1.1}$ | $91.2_{\pm18.4}$ | $98.2_{\pm1.1}$ | $64.0_{\pm22.5}$ | $89.2_{\pm5.5}$ | $91.8_{\pm18.6}$ | $68.8_{\pm22.4}$ |
| compas | A: | $66.2_{\pm1.1}$ | $66.1_{\pm1.2}$ | $61.9_{\pm8.9}$ | $66.2_{\pm1.1}$ | $43.8_{\pm13.0}$ | $66.0_{\pm1.1}$ | $61.7_{\pm9.4}$ | $56.0_{\pm13.0}$ |
| $N=5, N-K=2$ | F: | $95.4_{\pm2.6}$ | $93.1_{\pm1.8}$ | $94.3_{\pm2.6}$ | $95.3_{\pm2.5}$ | $86.2_{\pm7.5}$ | $90.2_{\pm1.9}$ | $94.3_{\pm2.4}$ | $68.7_{\pm26.2}$ |
| drugs | A: | $64.4_{\pm1.5}$ | $64.3_{\pm1.5}$ | $52.1_{\pm11.8}$ | $64.3_{\pm1.4}$ | $55.4_{\pm9.2}$ | $64.3_{\pm1.5}$ | $58.1_{\pm9.8}$ | $52.9_{\pm11.1}$ |
| $N=5, N-K=2$ | F: | $93.0_{\pm4.5}$ | $88.9_{\pm5.2}$ | $72.6_{\pm24.7}$ | $92.5_{\pm4.9}$ | $69.6_{\pm12.3}$ | $86.4_{\pm6.2}$ | $82.9_{\pm18.1}$ | $70.5_{\pm21.0}$ |
| germancredit | A: | $66.5_{\pm2.8}$ | $66.0_{\pm2.7}$ | $60.2_{\pm5.1}$ | $66.8_{\pm2.8}$ | $59.7_{\pm6.9}$ | $66.2_{\pm2.5}$ | $55.9_{\pm9.8}$ | $56.6_{\pm11.5}$ |
| $N=5, N-K=2$ | F: | $93.8_{\pm3.8}$ | $92.6_{\pm3.8}$ | $86.2_{\pm13.1}$ | $93.9_{\pm3.9}$ | $87.1_{\pm5.1}$ | $92.5_{\pm4.1}$ | $81.0_{\pm12.6}$ | $75.4_{\pm18.9}$ |
| folktables | A: | $75.5_{\pm0.2}$ | $75.5_{\pm0.2}$ | $74.6_{\pm0.2}$ | $75.4_{\pm0.2}$ | $74.4_{\pm0.5}$ | $75.4_{\pm0.2}$ | $74.6_{\pm0.2}$ | $74.2_{\pm0.5}$ |
| $N=51, N-K=5$ | F: | $99.5_{\pm0.3}$ | $99.0_{\pm0.5}$ | $99.3_{\pm0.3}$ | $99.5_{\pm0.2}$ | $93.0_{\pm2.6}$ | $99.1_{\pm0.6}$ | $99.4_{\pm0.2}$ | $92.7_{\pm4.0}$ |
| folktables | A: | $75.4_{\pm0.2}$ | $75.4_{\pm0.2}$ | $73.9_{\pm0.4}$ | $75.4_{\pm0.2}$ | $73.7_{\pm0.3}$ | $75.3_{\pm0.2}$ | $73.9_{\pm0.5}$ | $73.0_{\pm0.9}$ |
| $N=51, N-K=10$ | F: | $99.5_{\pm0.3}$ | $98.1_{\pm0.7}$ | $99.1_{\pm0.2}$ | $99.5_{\pm0.3}$ | $88.3_{\pm3.0}$ | $98.2_{\pm0.5}$ | $99.1_{\pm0.4}$ | $86.7_{\pm3.7}$ |
| folktables | A: | $75.4_{\pm0.2}$ | $75.4_{\pm0.3}$ | $72.7_{\pm0.6}$ | $75.4_{\pm0.2}$ | $72.3_{\pm0.9}$ | $75.3_{\pm0.3}$ | $72.7_{\pm0.8}$ | $68.8_{\pm8.1}$ |
| $N=51, N-K=15$ | F: | $99.7_{\pm0.2}$ | $97.0_{\pm0.6}$ | $98.9_{\pm0.4}$ | $99.6_{\pm0.3}$ | $82.4_{\pm2.4}$ | $97.1_{\pm1.0}$ | $98.9_{\pm0.5}$ | $78.9_{\pm5.4}$ |
| folktables | A: | $75.3_{\pm0.2}$ | $75.3_{\pm0.2}$ | $69.9_{\pm3.3}$ | $75.3_{\pm0.2}$ | $70.5_{\pm2.7}$ | $75.3_{\pm0.2}$ | $68.4_{\pm4.5}$ | $64.8_{\pm10.9}$ |
| $N=51, N-K=20$ | F: | $99.5_{\pm0.2}$ | $95.6_{\pm1.1}$ | $98.4_{\pm0.6}$ | $99.5_{\pm0.2}$ | $79.1_{\pm3.1}$ | $93.9_{\pm1.5}$ | $98.3_{\pm0.7}$ | $74.9_{\pm3.3}$ |
| folktables | A: | $74.0_{\pm1.4}$ | $75.0_{\pm0.3}$ | $53.9_{\pm14.4}$ | $74.1_{\pm1.3}$ | $41.6_{\pm14.1}$ | $75.1_{\pm0.3}$ | $59.3_{\pm11.5}$ | $56.5_{\pm13.3}$ |
| $N=51, N-K=25$ | F: | $94.2_{\pm1.5}$ | $89.9_{\pm1.6}$ | $95.2_{\pm4.7}$ | $94.5_{\pm1.8}$ | $79.4_{\pm6.7}$ | $89.3_{\pm1.7}$ | $96.9_{\pm1.7}$ | $72.6_{\pm2.7}$ |

the $D$-scores. The variant with all measures active is identical to FLEA. The variant with all measures inactive randomly chooses subsets to train on.

The results are presented in Table 5a. One can see that for all datasets, not using disc (column 3) for the filtering step has the most noticeable effect. This makes sense, because several of the adversaries make large changes to the labels and features, and disc is well suited to identify these.

Not using disp (column 2) has usually less of an effect, but in some situations it does lead to a noticeable drop in fairness, see e.g. COMPAS, and drugs, as well as folktables with $N - K \in \{5, 10, 15, 20\}$. This is also consistent with our expectations, as the measure is specifically able to detect even subtle manipulation that would negatively affect fairness. However, for folktables with $N - K = 25$, where the amount of manipulated data is very close to half, not using disp would actually be beneficial for the system. We attribute this to the fact that as a difference of ratios, disp is harder to estimate from small sample sets than the other two measures. A noisy estimate, however, can lead to clean sources to be suppressed, and manipulated ones to be selected. This explanation is also supported by the fact that for small datasets the variability of results is bigger when disp is included than when it is not.

The effect of not using disb (column 4) is small on real data. It never exceeds the standard deviation of the estimates. This, again, is expected, as the other two measures are typically able to ensure accuracy and fairness, whereas the role of disb is mainly to handle corner cases that are unlikely to occur in real data.

Dropping two measures from the filtering step only makes sense, if the remaining measure is disc (column 6). Even then, a decrease in accuracy and/or fairness is quite common, or at least an increase in variability.

Another interesting ablation study would be to determine the success rate of the FLEA's filtering step, i.e. how what fraction of the malignant sources it successfully suppresses. This, however, we cannot estimate, because we lack ground truth information which sources are malignant and which are not. From the experimental setup, only the information is available which sources have been manipulated and in what way. However, whether a manipulation is benign or malignant depends not only on the adversary's strategy, but also on

the actual data and the later learning strategy. The proxy measure of determining what fraction of all manipulated sources were detected would not be very meaningful, as adversaries can easily create manipulated data sources that are indistinguishable from clean ones, e.g. just shuffling the data point or not making any changes at all, as realized in our experiments by the *ID* adversary.

## F   Complete formulation and proof of Theorem 1

In this section we present the full proof of Theorem 1. We begin by reminding the reader of our notation and formal assumption from in F.1. Next in Section F.2 we state a few standard concentration results that are used in our main proof. In Section F.3 we define the population counterparts of the empirical discrepancy, disparity and disbalance measures, as understanding how well these measures are estimated from finite data is crucial for the proof of our results.

Finally, we present the full proof of Theorem 1 in Section F.4 and the proof of the concentration lemmas from Section F.2 in Section F.5.

### F.1   Assumptions and formal adversary model

For convenience of the reader, we repeat the formal notation and assumptions stated in 3.3.1. Initially, there are $N$ datasets $\tilde{S}_1, \ldots, \tilde{S}_N$, with the $i$-th set of samples being drawn i.i.d. from a distribution $p_i(x, y, a)$. We assume that all these distributions are clean, in the sense that they are close to the true target distribution $p$. Formally, we assume that each of the following conditions hold:

$$\mathrm{TV}(p_i(x, y, a), p(x, y, a)) \le \eta, \quad \text{and} \quad \max_{z \in \mathcal{A}} \big\{ \mathrm{TV}(p_i(x, y | a = z), p(x, y | a = z)) \big\} \le \eta, \tag{16}$$

where $\mathrm{TV}(p, q) = \sup_{B \in \mathcal{B}(\mathcal{X} \times \mathcal{Y} \times \mathcal{A})} |p(B) - q(B)|$ with $\mathcal{B}(X)$ denoting the Borel $\sigma$-algebra on a topological space $X$.

Once the clean datasets $\tilde{S}_1, \ldots, \tilde{S}_N$ are sampled, an *adversary* operates on them. This results in new datasets, $S_1, \ldots, S_N$, which the learning algorithm receives as input. The adversary is an arbitrary (deterministic or randomized) function $\mathcal{F} : \prod_{i=1}^N (\mathcal{X} \times \mathcal{Y} \times \mathcal{A})^n \to \prod_{i=1}^N (\mathcal{X} \times \mathcal{Y} \times \mathcal{A})^n$, with the only restriction that for a fixed subset of indices, $G \subset \{1, \ldots, N\}$, the data remains unchanged. That is, $S_i = \tilde{S}_i$ for all $i \in G$, and $S_i$ is arbitrary for $i \notin G$.

Note that the learner only observes the datasets $S_i$ and outputs a hypothesis based on them. Therefore, in the proof we will only work with the datasets $S_i$ and not with $\tilde{S}_i$, using that $S_i = \tilde{S}_i$ whenever $i \in G$, so that $S_i$ is i.i.d. from $p_i$. For simplicity, we refer to a dataset $S_i$ or a source $i \in [N]$ as clean if $i \in G$.

We assume without loss of generality that $\tau = \mathbb{P}_{(X,Y,A) \sim p}(A = 0) \in (0, \frac{1}{2}]$. For technical reasons, we also assume that $18\eta < \tau = \mathbb{P}_{(X,Y,A) \sim p}(A = 0)$.

### F.2   Concentration tools and notation

We first present the two lemmas which demonstrate uniform convergence of the empirical risk and the empirical fairness deviation measure respectively, for any hypothesis set $\mathcal{H}$ with finite VC dimension. The first is just the classic VC generalization bound, as given in Chapter 28.1 of Shalev-Shwartz & Ben-David (2014). The proof of the second lemma closely follows the proofs of similar results from Woodworth et al. (2017); Agarwal et al. (2018); Konstantinov & Lampert (2022) and is presented in Section F.5 for completeness.

**Lemma 1** (Uniform Convergence for Binary Loss). *Let $d$ be the VC-dimension of $\mathcal{H}$. Then for any dataset $S$ of size $n$ sampled i.i.d. from a distribution $p$, for all $\delta \in (0, 1)$,*

$$\mathbb{P}\left( \sup_{h \in \mathcal{H}} |\mathcal{R}_S(h) - \mathcal{R}_p(h)| > 2\sqrt{\frac{8d \log\left(\frac{en}{d}\right) + 2\log\left(\frac{4}{\delta}\right)}{n}} \right) \le \delta.$$

**Lemma 2** (Uniform Convergence for demographic parity). *Let $p$ be a distribution on $\mathcal{X} \times \mathcal{A} \times \mathcal{Y}$. Let $d = \mathrm{VC}(\mathcal{H}) \ge 1$ and let $\tau = \min_{a \in \{0,1\}} \mathbb{P}_{(X,Y,A) \sim p}(A = a)$ for some constant $\tau \in (0, 0.5]$. Then for any*

*dataset $S$ of size $n \geq \max\left\{\frac{8\log\left(\frac{8}{\delta}\right)}{\tau}, \frac{d}{2}\right\}$ sampled i.i.d. from $p$, for all $\delta \in (0, 1/2)$:*

$$\mathbb{P}_S\left(\sup_{h \in \mathcal{H}} |\Gamma_S(h) - \Gamma_p(h)| \geq 16\sqrt{2\frac{d\log\left(\frac{2en}{d}\right) + \log\left(\frac{24}{\delta}\right)}{n\tau}}\right) \leq \delta \tag{17}$$

For the dataset $S_i$, denote by

$$c_i := \sum_{(x,y,a) \in S_i} \mathbb{1}\{a = 0\} = |S_1^{a=0}|. \tag{18}$$

Denote $\tau_i = \mathbb{P}_{(X,Y,A) \sim p_i}(A = 0)$. Then for a clean data source we have that $c_i \sim \text{Bin}(n, \tau_i)$. Therefore, by the Hoeffding bound, for any $\delta > 0$:

$$\mathbb{P}\left(|c_i - n\tau_i| \geq n\sqrt{\frac{\log\left(\frac{2}{\delta}\right)}{2n}}\right) \leq 2\exp\left(-\frac{2\left(\sqrt{\frac{n}{2}\log\left(\frac{2}{\delta}\right)}\right)^2}{n}\right) = \delta. \tag{19}$$

Because, by assumption, $\tau = \mathbb{P}_{(X,Y,A) \sim p_i}(A = 0) = \min_{a \in \{0,1\}} \mathbb{P}_{(X,Y,A) \sim p}(A = a)$ and $TV(p_i, p) \leq \eta$, for any clean dataset $S_i$, it holds that

$$\tau_i = \mathbb{P}_{(X,Y,A) \sim p_i}(A = 0) \geq \mathbb{P}_{(X,Y,A) \sim p}(A = 0) - \eta = \tau - \eta.$$

In addition,

$$1 - \tau_i = \mathbb{P}_{(X,Y,A) \sim p_i}(A = 1) \geq \mathbb{P}_{(X,Y,A) \sim p}(A = 1) - \eta \geq \mathbb{P}_{(X,Y,A) \sim p}(A = 0) - \eta = \tau - \eta.$$

Recall also that $\tau - \eta \geq \tau - 18\eta > 0$ by assumption. Denote by:

$$\Delta(\delta) = \max\left\{2\sqrt{\frac{8d\log\left(\frac{en}{d}\right) + 2\log\left(\frac{4}{\delta}\right)}{n}}, 16\sqrt{2\frac{d\log\left(\frac{2en}{d}\right) + \log\left(\frac{24}{\delta}\right)}{n(\tau - \eta)}}, \sqrt{\frac{\log\left(\frac{2}{\delta}\right)}{2n}}\right\} \tag{20}$$

$$= 16\sqrt{2\frac{d\log\left(\frac{2en}{d}\right) + \log\left(\frac{24}{\delta}\right)}{n(\tau - \eta)}}. \tag{21}$$

The lemmas above, as well as the observation that $\min\{\tau_i, 1 - \tau_i\} \geq \tau - \eta$ for any clean source $i$, readily imply that:

$$\mathbb{P}\left(\sup_{h \in \mathcal{H}} |\mathcal{R}_{S_i}(h) - \mathcal{R}_{p_i}(h)| \geq \Delta(\delta)\right) \leq \delta, \tag{22}$$

$$\mathbb{P}_S\left(\sup_{h \in \mathcal{H}} |\Gamma_{S_i}(h) - \Gamma_{p_i}(h)| \geq \Delta(\delta)\right) \leq \delta \tag{23}$$

and

$$\mathbb{P}_S\left(|c_i - n\tau_i| \geq n\Delta(\delta)\right) \leq \delta, \tag{24}$$

for any clean $i$.

### F.3 Discrepancy, disparity and disbalance between distributions

In our proof we will consider the population counterparts of the between-dataset distances that we defined in the main body of the text. In particular, the discrepancy distance between two distributions $p$ and $q$ is

$$\text{disc}(p, q) = \sup_{h \in \mathcal{H}} |\mathcal{R}_p(h) - \mathcal{R}_q(h)|. \tag{25}$$

Similarly, the disparity is

$$\text{disp}(p,q) = \sup_{h \in \mathcal{H}} |\Gamma_p(h) - \Gamma_q(h)|, \tag{26}$$

where as an unfairness measure $\Gamma_p$ we will use:

$$\Gamma_p(h) = \mathbb{P}_{(X,Y,A) \sim p}(h(X) = 1|A = 0) - \mathbb{P}_{(X,Y,A) \sim p}(h(X) = 1|A = 1).$$

Finally, the disbalance is simply

$$\text{disb}(p,q) = |\mathbb{P}_p(A = 0) - \mathbb{P}_q(A = 0)|. \tag{27}$$

Next we study these distances, between a distribution $p_i$ of a clean source and the true target distribution $p$. Recall our assumptions about the bounded TV distances from Section F.1. Clearly, we have that $\text{disb}(p_i, p) \leq TV(p_i, p) \leq \eta$. Note also that:

$$\text{disc}(p_i, p) = \sup_{h \in \mathcal{H}} |\mathcal{R}_{p_i}(h) - \mathcal{R}_p(h)| = \sup_{h \in \mathcal{H}} \left| \mathbb{P}_{(X,Y,A) \sim p_i}(h(X) \neq Y) - \mathbb{P}_{(X,Y,A) \sim p}(h(X) \neq Y) \right| \leq \eta,$$

because any (measurable) classifier $h : \mathcal{X} \to \mathcal{Y}$ can be associated with a (Borel) set $S_h = \{(x, y, a) \in (\mathcal{X} \times \mathcal{Y} \times \mathcal{A}) : h(x) \neq y\}$. Finally, we bound the disparity in terms of $\eta$. Note that:

$$
\begin{aligned}
\text{disp}(p_i, p) &= \sup_{h \in \mathcal{H}} \Big| \mathbb{P}_{(X,Y,A) \sim p_i}(h(X) = 1|A = 0) - \mathbb{P}_{(X,Y,A) \sim p_i}(h(X) = 1|A = 1) \\
&\quad - \mathbb{P}_{(X,Y,A) \sim p}(h(X) = 1|A = 0) + \mathbb{P}_{(X,Y,A) \sim p}(h(X) = 1|A = 1) \Big| \\
&\leq \sup_{h \in \mathcal{H}} \Big( \left| \mathbb{P}_{(X,Y,A) \sim p_i}(h(X) = 1|A = 0) - \mathbb{P}_{(X,Y,A) \sim p}(h(X) = 1|A = 0) \right| \\
&\quad + \left| \mathbb{P}_{(X,Y,A) \sim p_i}(h(X) = 1|A = 1) - \mathbb{P}_{(X,Y,A) \sim p}(h(X) = 1|A = 1) \right| \Big) \\
&\leq \sup_{h \in \mathcal{H}} \left| \mathbb{P}_{(X,Y,A) \sim p_i}(h(X) = 1|A = 0) - \mathbb{P}_{(X,Y,A) \sim p}(h(X) = 1|A = 0) \right| \\
&\quad + \sup_{h \in \mathcal{H}} \left| \mathbb{P}_{(X,Y,A) \sim p_i}(h(X) = 1|A = 1) - \mathbb{P}_{(X,Y,A) \sim p}(h(X) = 1|A = 1) \right| \\
&\leq 2\eta.
\end{aligned}
$$

### F.4 Proof

**Theorem 1.** *Assume that $\mathcal{H}$ has a finite VC-dimension $d \geq 1$. Let $p$ be an arbitrary target data distribution and without loss of generality let $\tau = p(a = 0) \in (0, 0.5]$. Let $S_1, \ldots, S_N$ be $N$ datasets, each consisting of $n$ samples, out of which $K > \frac{N}{2}$ are sampled i.i.d. from a data distribution $p_i$ that is $\eta$-close the distribution $p$ in the sense of Section F.1. Assume that $18\eta < \tau$. For $\frac{1}{2} < \beta \leq \frac{K}{N}$ and $I = \text{FILTERSOURCES}(S_1, \ldots, S_N; \beta)$ set $S = \bigcup_{i \in I} S_i$. Let $\delta > 0$. Then there exists a constant $C = C(\delta, \tau, d, N, \eta)$, such that for any $n \geq C$, the following inequalities hold with probability at least $1 - \delta$ uniformly over all $f \in \mathcal{H}$ and against any adversary:*

$$|\Gamma_S(f) - \Gamma_p(f)| \leq \mathcal{O}(\eta) + \widetilde{\mathcal{O}}\left(\sqrt{\frac{1}{n}}\right), \qquad |\mathcal{R}_S(f) - \mathcal{R}_p(f)| \leq \mathcal{O}(\eta) + \widetilde{\mathcal{O}}\left(\sqrt{\frac{1}{n}}\right). \tag{28}$$

*Proof.* First, we characterize a set of values into which the empirical risks and empirical deviation measures of the clean data sources falls with probability at least $1 - \delta$. Then we show that because the clean datasets cluster in such a way, any individual dataset that is accepted by the FILTERSOURCES algorithm provides good empirical estimates of the true risk and the unfairness measure. Finally, we show that the same holds for the union of these sets, $S$, which implies the inequalities (28). For the risk, the last step is a straightforward consequence of the second. For the fairness, however, a careful derivation is needed that crucially uses the disbalance measure as well.

**Step 1** Let $G \subset [N]$ be the set of indexes $i$, such that $S_i$ was not modified by the adversary. By definition, $|G| = K$. Now consider the following events that, as we will show, describe the likely values of the studied quantities on the clean datasets.

In particular, for all $i \in G$, let $\mathcal{E}_i^{\mathcal{R}}$ be the event that:

$$\sup_{h \in \mathcal{H}} \left| \mathcal{R}_{S_i}(h) - \mathcal{R}_{p_i}(h) \right| \leq \Delta\left(\frac{\delta}{6N}\right), \tag{29}$$

let $\mathcal{E}_i^{\Gamma}$ be the event that

$$\sup_{h \in \mathcal{H}} \left| \Gamma_{S_i}(h) - \Gamma_{p_i}(h) \right| \leq \Delta\left(\frac{\delta}{6N}\right), \tag{30}$$

let $\mathcal{E}_i^{bin}$ be the event that

$$|c_i - n\tau_i| \leq n\Delta\left(\frac{\delta}{6N}\right) \tag{31}$$

and finally, let $\mathcal{E}_i^{count}$ be the event that

$$0 < c_i < n. \tag{32}$$

Denote by $(\mathcal{E}_i^{\mathcal{R}})^c, (\mathcal{E}_i^{\Gamma})^c$ and $(\mathcal{E}_i^{bin})^c, (\mathcal{E}_i^{count})^c$ the respective complements of these events. Then, by equations (22) and (23), (24), we have:

$$\mathbb{P}((\mathcal{E}_i^{\mathcal{R}})^c) \leq \frac{\delta}{6N}, \quad \mathbb{P}((\mathcal{E}_i^{\Gamma})^c) \leq \frac{\delta}{6N}, \quad \mathbb{P}((\mathcal{E}_i^{bin})^c) \leq \frac{\delta}{6N}, \quad \forall i \in G.$$

To bound the probability of the complement of $\mathcal{E}_i^{count}$, note that for any $i \in G$

$$1 - \tau_i = \mathbb{P}_{(X,Y,A) \sim p_i}(A = 1) \leq \mathbb{P}_{(X,Y,A) \sim p}(A = 1) + \eta = 1 - \tau + \eta$$

and that $1 - \tau + \eta < 1$ because of the assumption that $\eta < \tau$. Similarly,

$$\tau_i = \mathbb{P}_{(X,Y,A) \sim p_i}(A = 0) \leq \mathbb{P}_{(X,Y,A) \sim p}(A = 0) + \eta = \tau + \eta \leq 1 - \tau + \eta.$$

Now, for any $i \in G$, whenever $n \geq C_1(\delta, \tau, d, N) = \frac{\log\left(\frac{4N}{\delta}\right)}{\log\left(\frac{1}{1-\tau+\eta}\right)} \geq \max\left\{\frac{\log\left(\frac{4N}{\delta}\right)}{\log\left(\frac{1}{1-\tau_i}\right)}, \frac{\log\left(\frac{4N}{\delta}\right)}{\log\left(\frac{1}{\tau_i}\right)}\right\}$, we have that

$$\mathbb{P}\left(\left(\mathcal{E}_i^{count}\right)^c\right) = (1 - \tau_i)^n + \tau_i^n$$

$$\leq \exp\left(-n\log\left(\frac{1}{1-\tau_i}\right)\right) + \exp\left(-n\log\left(\frac{1}{\tau_i}\right)\right)$$

$$\leq \frac{\delta}{4N} + \frac{\delta}{4N} = \frac{\delta}{2N}.$$

Therefore, setting $\mathcal{E} := (\wedge_{i \in G} \mathcal{E}_i^{\mathcal{R}}) \wedge (\wedge_{i \in G} \mathcal{E}_i^{\Gamma}) \wedge (\wedge_{i \in G} \mathcal{E}_i^{bin}) \wedge (\wedge_{i \in G} \mathcal{E}_i^{count})$ then by the union bound the probability of $\mathbb{P}(\mathcal{E}^c) \leq K\frac{\delta}{6N} + K\frac{\delta}{6N} + K\frac{\delta}{6N} + K\frac{\delta}{2N} \leq 3\frac{\delta}{6} + \frac{\delta}{2} = \delta$.

Hence the probability of the event $\mathcal{E}$ that all of (29), (30), (31), and (32) hold is at least $1 - \delta$.

**Step 2** Now we show that under the event $\mathcal{E}$, the inequalities in (28) are fulfilled. Indeed, assume that $\mathcal{E}$ holds. Fix any adversary $\mathcal{A}$ and any $h \in \mathcal{H}$.

For any pair of clean sources $i, j \in [N]$ the triangle law and the derivations in Section F.3 give:

$$\text{disc}(S_i, S_j) = \sup_{h \in \mathcal{H}} |\mathcal{R}_{S_i}(h) - \mathcal{R}_{S_j}(h)|$$

$$\leq \sup_{h \in \mathcal{H}} |\mathcal{R}_{S_i}(h) - \mathcal{R}_{p_i}(h)| + \sup_{h \in \mathcal{H}} |\mathcal{R}_{p_i}(h) - \mathcal{R}_p(h)| + \sup_{h \in \mathcal{H}} |\mathcal{R}_p(h) - \mathcal{R}_{p_j}(h)| + \sup_{h \in \mathcal{H}} |\mathcal{R}_{p_j}(h) - \mathcal{R}_{S_j}(h)|$$

$$\leq 2\eta + 2\Delta\left(\frac{\delta}{6N}\right).$$

Similarly,

$$\mathrm{disp}(S_i, S_j) = \sup_{h\in\mathcal{H}} |\Gamma_{S_i}(h) - \Gamma_{S_j}(h)| \leq 4\eta + 2\Delta\left(\frac{\delta}{6N}\right)$$

and

$$\mathrm{disb}(S_i, S_j) = \left|\frac{c_i}{n} - \frac{c_j}{n}\right| \leq 2\eta + 2\Delta\left(\frac{\delta}{6N}\right).$$

Therefore, for any pair of clean sources $i, j \in [N]$:

$$\mathrm{disc}(S_i, S_j) + \mathrm{disp}(S_i, S_j) + \mathrm{disb}(S_i, S_j) \leq 8\eta + 6\Delta\left(\frac{\delta}{6N}\right). \tag{33}$$

It follows that, under $\mathcal{E}$, we have that $q_i \leq 8\eta + 6\Delta\left(\frac{\delta}{6N}\right)$ for any clean $i \in [N]$. Since the fraction of clean sources is $\frac{K}{N} \geq \beta$, it follows that also $q \leq 8\eta + 6\Delta\left(\frac{\delta}{6N}\right)$, where $q$ is the $\beta$-th quantile of the $q_i$'s.

Denote by $I = \textsc{FilterSources}(S_1, \ldots, S_N; \beta)$ the result of the filtering algorithm. Now for any $i \in I$, we have that $q_i \leq q \leq 8\eta + 6\Delta\left(\frac{\delta}{6N}\right)$. In addition, by the definition of $q_i$, $\mathrm{disc}(S_i, S_j) \leq \mathrm{disc}(S_i, S_j) + \mathrm{disp}(S_i, S_j) + \mathrm{disb}(S_i, S_j) \leq q_i$ for at least $|I| = \beta N > \frac{N}{2}$ values of $j \in [N]$. Since $K > \frac{N}{2}$, this means that $\mathrm{disc}(S_i, S_j) \leq q_i \leq 8\eta + 6\Delta\left(\frac{\delta}{6N}\right)$ for at least 1 value $j \in G$. Therefore, we have:

$$\sup_{h\in\mathcal{H}} |\mathcal{R}_{S_i}(h) - \mathcal{R}_p(h)| \leq \sup_{h\in\mathcal{H}} |\mathcal{R}_{S_i}(h) - \mathcal{R}_{S_j}(h)| + \sup_{h\in\mathcal{H}} |\mathcal{R}_{S_j}(h) - \mathcal{R}_{p_j}(h)| + \sup_{h\in\mathcal{H}} |\mathcal{R}_{p_j}(h) - \mathcal{R}_p(h)| \tag{34}$$

$$\leq 8\eta + 6\Delta\left(\frac{\delta}{6N}\right) + \Delta\left(\frac{\delta}{6N}\right) + \eta \tag{35}$$

$$= 9\eta + 7\Delta\left(\frac{\delta}{6N}\right) \tag{36}$$

because $\mathcal{E}$ holds. Similarly,

$$\sup_{h\in\mathcal{H}} |\Gamma_{S_i}(h) - \Gamma_p(h)| \leq 10\eta + 7\Delta\left(\frac{\delta}{6N}\right) \tag{37}$$

and

$$|c_i - n\tau| \leq 9\eta n + 7n\Delta\left(\frac{\delta}{6N}\right). \tag{38}$$

**Step 3** Finally, we study the risk and disparity measures based on all filtered data $S = \cup_{i\in I} S_i$.

Denote by $\mathcal{R}_S(h)$ the empirical risk across the entire trusted dataset $I$:

$$\mathcal{R}_S(h) := \frac{1}{|I|} \sum_{i\in I} \mathcal{R}_{S_i}(h). \tag{39}$$

Then the triangle law gives:

$$|\mathcal{R}_S(h) - \mathcal{R}_p(h)| = \left|\frac{1}{|I|}\left(\sum_{i\in I} \mathcal{R}_{S_i}(h) - \mathcal{R}_p(h)\right)\right| \leq \frac{1}{|I|} \sum_{i\in I} |\mathcal{R}_{S_i}(h) - \mathcal{R}_p(h)| = 9\eta + 7\Delta\left(\frac{\delta}{6N}\right)$$

Since

$$3\Delta\left(\frac{\delta}{6N}\right) = 112\sqrt{2\frac{d\log\left(\frac{2en}{d}\right) + \log\left(\frac{144N}{\delta}\right)}{(\tau - \eta)n}} = \widetilde{\mathcal{O}}\left(\sqrt{\frac{d}{(\tau - \eta)n}}\right), \tag{40}$$

the bound on the risk follows.

Denote by $\Gamma_S(h)$ the empirical estimate of demographic parity across the entire trusted dataset $I$:

$$\Gamma_S(h) := \frac{\sum_{j\in I}\sum_{i=1}^{n}\mathbb{1}\{h(x_i^{(j)})=1, a_i^{(j)}=0\}}{\sum_{j\in I}\sum_{i=1}^{n}\mathbb{1}\{a_i^{(j)}=0\}} - \frac{\sum_{j\in I}\sum_{i=1}^{n}\mathbb{1}\{h(x_i^{(j)})=1, a_i^{(j)}=1\}}{\sum_{j\in I}\sum_{i=1}^{n}\mathbb{1}\{a_i^{(j)}=1\}}. \tag{41}$$

For convenience, denote $v_j = v_j(h) = \sum_{i=1}^{n}\mathbb{1}\{h(x_i^{(j)})=1, a_i^{(j)}=0\}$ and $w_j = w_j(h) = \mathbb{1}\{h(x_i^{(j)})=1, a_i^{(j)}=1\}$, so that:

$$\Gamma_S(h) = \frac{\sum_{j\in I}v_j}{\sum_{j\in I}c_j} - \frac{\sum_{j\in I}w_j}{\sum_{j\in I}(n-c_j)}.$$

Our goal is to bound the difference $\left|\Gamma_S - \frac{1}{|I|}\sum_{i\in I}\Gamma_{S_i}\right|$, and the difference $\left|\frac{1}{|I|}\sum_{i\in I}\Gamma_{S_i} - \Gamma_p\right|$, and use these two bounds to bound $|\Gamma_S - \Gamma_p|$. The second bound follows directly from (37):

$$\left|\Gamma_p - \frac{1}{|I|}\sum_{i\in I}\Gamma_{S_i}\right| \le 10\eta + 7\Delta\left(\frac{\delta}{6N}\right) \tag{42}$$

To compute the first bound, we first build on (38) to note

$$\left|\frac{c_i}{n\tau} - 1\right| = \frac{|c_i - n\tau|}{n\tau} \le \frac{9\eta n + 7n\Delta\left(\frac{\delta}{6N}\right)}{n\tau} \le \frac{9\eta + 7\Delta\left(\frac{\delta}{6N}\right)}{\tau - \left(9\eta + 7\Delta\left(\frac{\delta}{6N}\right)\right)}$$

$$\left|\frac{n\tau}{c_i} - 1\right| = \frac{|n\tau - c_i|}{c_i} \le \frac{9\eta n + 7n\Delta\left(\frac{\delta}{6N}\right)}{c_i} \le \frac{9\eta + 7\Delta\left(\frac{\delta}{6N}\right)}{\tau - \left(9\eta + 7\Delta\left(\frac{\delta}{6N}\right)\right)}$$

and therefore,

$$1 - \frac{9\eta + 7\Delta\left(\frac{\delta}{6N}\right)}{\tau - \left(9\eta + 7\Delta\left(\frac{\delta}{6N}\right)\right)} \le \frac{c_i}{n\tau} \le 1 + \frac{9\eta + 7\Delta\left(\frac{\delta}{6N}\right)}{\tau - \left(9\eta + 7\Delta\left(\frac{\delta}{6N}\right)\right)} \tag{43}$$

$$1 - \frac{9\eta + 7\Delta\left(\frac{\delta}{6N}\right)}{\tau - \left(9\eta + 7\Delta\left(\frac{\delta}{6N}\right)\right)} \le \frac{n\tau}{c_i} \le 1 + \frac{9\eta + 7\Delta\left(\frac{\delta}{6N}\right)}{\tau - \left(9\eta + 7\Delta\left(\frac{\delta}{6N}\right)\right)} \tag{44}$$

Applying the same logic to $n - c_i$:

$$1 - \frac{9\eta + 7\Delta\left(\frac{\delta}{6N}\right)}{1 - \tau - \left(9\eta + 7\Delta\left(\frac{\delta}{6N}\right)\right)} \le \frac{n - c_i}{n - n\tau} \le 1 + \frac{9\eta + 7\Delta\left(\frac{\delta}{6N}\right)}{1 - \tau - \left(9\eta + 7\Delta\left(\frac{\delta}{6N}\right)\right)} \tag{45}$$

$$1 - \frac{9\eta + 7\Delta\left(\frac{\delta}{6N}\right)}{1 - \tau - \left(9\eta + 7\Delta\left(\frac{\delta}{6N}\right)\right)} \le \frac{n - n\tau}{n - c_i} \le 1 + \frac{9\eta + 7\Delta\left(\frac{\delta}{6N}\right)}{1 - \tau - \left(9\eta + 7\Delta\left(\frac{\delta}{6N}\right)\right)} \tag{46}$$

Now consider,

$$\frac{1}{|I|}\sum_{j\in I}\Gamma_{S_j} = \frac{1}{|I|}\sum_{j\in I}\frac{v_j}{c_j} - \frac{1}{|I|}\sum_{j\in I}\frac{w_j}{(n-c_j)}$$

$$\le \frac{1}{|I|}\sum_{j\in I}\frac{v_j}{c}\left(1 + \frac{9\eta + 7\Delta\left(\frac{\delta}{6N}\right)}{\tau - \left(9\eta + 7\Delta\left(\frac{\delta}{6N}\right)\right)}\right) - \frac{1}{|I|}\sum_{j\in I}\frac{w_j}{(n-c)}\left(1 - \frac{9\eta + 7\Delta\left(\frac{\delta}{6N}\right)}{1 - \tau - \left(9\eta + 7\Delta\left(\frac{\delta}{6N}\right)\right)}\right)$$

$$= \frac{\sum_{j\in I}v_j}{\sum_{j\in I}c}\left(1 + \frac{9\eta + 7\Delta\left(\frac{\delta}{6N}\right)}{\tau - \left(9\eta + 7\Delta\left(\frac{\delta}{6N}\right)\right)}\right) - \frac{\sum_{j\in I}w_j}{\sum_{j\in I}(n-c)}\left(1 - \frac{9\eta + 7\Delta\left(\frac{\delta}{6N}\right)}{1 - \tau - \left(9\eta + 7\Delta\left(\frac{\delta}{6N}\right)\right)}\right)$$

$$\le \frac{\sum_{j\in I}v_j}{\sum_{j\in I}c_j\left(1 - \frac{9\eta + 7\Delta\left(\frac{\delta}{6N}\right)}{\tau - \left(9\eta + 7\Delta\left(\frac{\delta}{6N}\right)\right)}\right)}\left(1 + \frac{9\eta + 7\Delta\left(\frac{\delta}{6N}\right)}{\tau - \left(9\eta + 7\Delta\left(\frac{\delta}{6N}\right)\right)}\right)$$

$$- \frac{\sum_{j \in I} w_j}{\sum_{j \in I}(n - c_j)\left(1 + \frac{9\eta + 7\Delta\left(\frac{\delta}{6N}\right)}{1 - \tau - \left(9\eta + 7\Delta\left(\frac{\delta}{6N}\right)\right)}\right)}\left(1 - \frac{9\eta + 7\Delta\left(\frac{\delta}{6N}\right)}{1 - \tau - \left(9\eta + 7\Delta\left(\frac{\delta}{6N}\right)\right)}\right)$$

$$= \left(\frac{\sum_{j \in I} v_j}{\sum_{j \in I} c_j}\right)\frac{1 + \frac{9\eta + 7\Delta\left(\frac{\delta}{6N}\right)}{\tau - \left(9\eta + 7\Delta\left(\frac{\delta}{6N}\right)\right)}}{1 - \frac{9\eta + 7\Delta\left(\frac{\delta}{6N}\right)}{\tau - \left(9\eta + 7\Delta\left(\frac{\delta}{6N}\right)\right)}} - \left(\frac{\sum_{j \in I} w_j}{\sum_{j \in I}(n - c_j)}\right)\frac{1 - \frac{9\eta + 7\Delta\left(\frac{\delta}{6N}\right)}{1 - \tau - \left(9\eta + 7\Delta\left(\frac{\delta}{6N}\right)\right)}}{1 + \frac{9\eta + 7\Delta\left(\frac{\delta}{6N}\right)}{1 - \tau - \left(9\eta + 7\Delta\left(\frac{\delta}{6N}\right)\right)}}$$

$$= \frac{\sum_{j \in I} v_j}{\sum_{j \in I} c_j} - \frac{\sum_{j \in I} w_j}{\sum_{j \in I}(n - c_j)} + 2\left(\frac{\sum_{j \in I} v_j}{\sum_{j \in I} c_j}\right)\frac{\frac{9\eta + 7\Delta\left(\frac{\delta}{6N}\right)}{\tau - \left(9\eta + 7\Delta\left(\frac{\delta}{6N}\right)\right)}}{1 - \frac{9\eta + 7\Delta\left(\frac{\delta}{6N}\right)}{\tau - \left(9\eta + 7\Delta\left(\frac{\delta}{6N}\right)\right)}} + 2\left(\frac{\sum_{j \in I} w_j}{\sum_{j \in I}(n - c_j)}\right)\frac{\frac{9\eta + 7\Delta\left(\frac{\delta}{6N}\right)}{1 - \tau - \left(9\eta + 7\Delta\left(\frac{\delta}{6N}\right)\right)}}{1 + \frac{9\eta + 7\Delta\left(\frac{\delta}{6N}\right)}{1 - \tau - \left(9\eta + 7\Delta\left(\frac{\delta}{6N}\right)\right)}},$$

where we have used that $\frac{1+t}{1-t} = 1 + \frac{2t}{1-t}$ and $\frac{1-t}{1+t} = 1 - \frac{2t}{1+t}$. Now, using $v_j \leq c_j$ and $w_j \leq n - c_j$ and simplifying the fractions further, we obtain

$$\leq \frac{\sum_{j \in I} v_j}{\sum_{j \in I} c_j} - \frac{\sum_{j \in I} w_j}{\sum_{j \in I}(n - c_j)} + \frac{2\left(9\eta + 7\Delta\left(\frac{\delta}{6N}\right)\right)}{\tau - 2\left(9\eta + 7\Delta\left(\frac{\delta}{6N}\right)\right)} + \frac{2\left(9\eta + 7\Delta\left(\frac{\delta}{6N}\right)\right)}{1 - \tau}$$

$$= \Gamma_S + \frac{2\left(9\eta + 7\Delta\left(\frac{\delta}{6N}\right)\right)}{\tau - 2\left(9\eta + 7\Delta\left(\frac{\delta}{6N}\right)\right)} + \frac{2\left(9\eta + 7\Delta\left(\frac{\delta}{6N}\right)\right)}{1 - \tau}$$

Using analogue steps, we can show that

$$-\frac{1}{|I|}\sum_{i \in I}\Gamma_{S_i} \leq -\Gamma_S + \frac{2\left(9\eta + 7\Delta\left(\frac{\delta}{6N}\right)\right)}{\tau} + \frac{2\left(9\eta + 7\Delta\left(\frac{\delta}{6N}\right)\right)}{1 - \tau - 2\left(9\eta + 7\Delta\left(\frac{\delta}{6N}\right)\right)}$$

Combining these two bounds:

$$\left|\frac{1}{|I|}\sum_{i \in I}\Gamma_{S_i} - \Gamma_S\right| \leq \frac{2\left(9\eta + 7\Delta\left(\frac{\delta}{6N}\right)\right)}{\tau - 2\left(9\eta + 7\Delta\left(\frac{\delta}{6N}\right)\right)} + \frac{2\left(9\eta + 7\Delta\left(\frac{\delta}{6N}\right)\right)}{1 - \tau - 2\left(9\eta + 7\Delta\left(\frac{\delta}{6N}\right)\right)} \tag{47}$$

Now, combining (47) with (42), and using the triangle inequality as before,

$$|\Gamma_p - \Gamma_S| \leq 10\eta + 7\Delta\left(\frac{\delta}{6N}\right) + \frac{2\left(9\eta + 7\Delta\left(\frac{\delta}{6N}\right)\right)}{\tau - 2\left(9\eta + 7\Delta\left(\frac{\delta}{6N}\right)\right)} + \frac{2\left(9\eta + 7\Delta\left(\frac{\delta}{6N}\right)\right)}{1 - \tau - 2\left(9\eta + 7\Delta\left(\frac{\delta}{6N}\right)\right)} \tag{48}$$

Recalling from (20) that $\Delta = 16\sqrt{2\frac{d\log\left(\frac{2en}{d}\right) + \log\left(\frac{24}{\delta}\right)}{n(\tau - \eta)}}$, we obtain that

$$|\Gamma_p - \Gamma_S| \leq \mathcal{O}(\eta) + \widetilde{\mathcal{O}}\left(\frac{1}{\sqrt{n}}\right). \tag{49}$$

$\square$

## F.5  Proof of Lemma 2

Let $S = \{(x_i, y_i, a_i)\}_{i=1}^n$. For $a \in \{0, 1\}$, denote:

$$\gamma_S^a(h) = \frac{\sum_{i=1}^n \mathbb{1}\{h(x_i) = 1, a_i = a\}}{\sum_{i=1}^n \mathbb{1}\{a_i = a\}} \tag{50}$$

and

$$\gamma_p^a(h) = \mathbb{P}(h(X) = 1 | A = a), \tag{51}$$

so that $\Gamma_S(h) = \gamma_S^0(h) - \gamma_S^1(h)$ and $\Gamma_p(h) = \gamma_p^0(h) - \gamma_p^1(h)$.

First we use a technique of Woodworth et al. (2017); Agarwal et al. (2018) for proving concentration results about conditional probability estimates to bound the probability of a large deviation of $\Gamma_S(h)$ from $\Gamma_p(h)$, for a fixed hypothesis $h \in \mathcal{H}$. Our result is similar to the one in Woodworth et al. (2017), but for demographic parity, instead of equal odds.

**Lemma 3.** *Let $h \in \mathcal{H}$ be a fixed hypothesis and $p \in \mathcal{P}(\mathcal{X} \times A \times \mathcal{Y})$ be a fixed distribution. Let $\tau = \min_{a \in \{0,1\}} \mathbb{P}_{(X,Y,A) \sim p}(A = a) \in (0, 0.5]$. Then for any dataset $S$, drawn i.i.d. from $p$, of size $n$ and for any $\delta \in (0,1)$ and any $t > 0$:*

$$\mathbb{P}\left(|\Gamma_S(h) - \Gamma_p(h)| > 2t\right) \le 6 \exp\left(-\frac{t^2 \tau n}{8}\right). \tag{52}$$

*Proof.* Denote by $S_a = \{i \in [n] : a_i = a\}$ the set of indexes of the points in $S$ for which the protected group is $a$. Let $c_a := |S_a|$ and $P_a = \mathbb{P}_{(X,Y,A) \sim p}(A = a)$, so that $\tau = \min_a P_a$. For both $a \in \{0,1\}$, we have:

$$\mathbb{P}\left(|\gamma_S^a - \gamma_p^a| > t\right) = \sum_{S_a} \mathbb{P}\left(|\gamma_S^a - \gamma_a| > t|S_a\right) \mathbb{P}(S_a)$$

$$\le \mathbb{P}\left(c_a \le \frac{1}{2} P_a n\right) + \sum_{S_a : c_a > \frac{1}{2} P_a n} \mathbb{P}\left(|\gamma_S^a - \gamma_a| > t|S_a\right) \mathbb{P}(S_a)$$

$$\le \exp\left(-\frac{P_a n}{8}\right) + \sum_{S_a : c_a > \frac{1}{2} P_a n} 2 \exp\left(-2t^2 c_a\right) \mathbb{P}(S_a)$$

$$\le \exp\left(-\frac{P_a n}{8}\right) + 2 \exp\left(-t^2 P_a n\right)$$

$$\le 3 \exp\left(-\frac{t^2 \tau n}{8}\right).$$

The triangle law gives:

$$|(\gamma_S^0 - \gamma_S^1) - (\gamma_p^0 - \gamma_p^1)| = |\gamma_S^0 - \gamma_S^1 - \gamma_p^0 + \gamma_p^1| \le |\gamma_S^0 - \gamma_p^0| + |\gamma_S^1 - \gamma_p^1|.$$

Combining the previous two results:

$$\mathbb{P}(|(\gamma_S^0 - \gamma_S^1) - (\gamma_p^0 - \gamma_p^1)| > 2t) \le \mathbb{P}\left(|\gamma_S^0 - \gamma_p^0| + |\gamma_S^1 - \gamma_p^1| > 2t\right)$$

$$\le \mathbb{P}\left((|\gamma_S^0 - \gamma_p^0| > t) \vee (|\gamma_S^1 - \gamma_p^1| > t)\right)$$

$$\le \mathbb{P}\left(|\gamma_S^0 - \gamma_p^0| > t\right) + \mathbb{P}\left(|\gamma_S^1 - \gamma_p^1| > t\right)$$

$$\le 6 \exp\left(-\frac{t^2 \tau n}{8}\right).$$

$\square$

Finally, we prove Lemma 2 by extending the previous result to hold uniformly over the whole hypothesis space, for any hypothesis space $\mathcal{H}$ with a finite VC-dimension $d := \mathrm{VC}(\mathcal{H})$. The extension is essentially identical to Konstantinov & Lampert (2022) and is included here for completeness.

**Lemma 2** (Uniform convergence for demographic parity)**.** *Let $d = \mathrm{VC}(\mathcal{H}) \ge 1$ and let $\tau = \min_{a \in \{0,1\}} \mathbb{P}_{(X,Y,A) \sim p}(A = a)$ for some constant $\tau \in (0, 0.5]$. Then for any dataset $S$ of size $n \ge \max\left\{\frac{8 \log\left(\frac{8}{\delta}\right)}{\tau}, \frac{d}{2}\right\}$ sampled i.i.d. from $p$, for all $\delta \in (0, 1/2)$:*

$$\mathbb{P}_S\left(\sup_{h \in \mathcal{H}} |\Gamma_S(h) - \Gamma_p(h)| \ge 16\sqrt{2\frac{d \log\left(\frac{2en}{d}\right) + \log\left(\frac{24}{\delta}\right)}{n\tau}}\right) \le \delta \tag{53}$$

*Proof.* To extend Lemma 3 to hold uniformly over $\mathcal{H}$, we first prove a version of the classic symmetrization lemma (Vapnik, 2013) for $\Gamma$ and then proceed via a standard growth function argument.

1) Consider a ghost sample $S' = \{(x_i', a_i', y_i')\}_{i=1}^n$ also sampled i.i.d. from $p$. For any $h \in \mathcal{H}$, let $\Gamma_{S'}(h)$ be the empirical estimate of $\Gamma_p(h)$ based on $S'$.

We show the following symmetrization inequality for the $\Gamma$ measure:

$$\mathbb{P}_S\left(\sup_{h \in \mathcal{H}} |\Gamma_S(h) - \Gamma_p(h)| \geq t\right) \leq 2\mathbb{P}_{S,S'}\left(\sup_{h \in \mathcal{H}} |\Gamma_{S'}(h) - \Gamma_S(h)| \geq t/2\right), \tag{54}$$

for any constant $t \geq 8\sqrt{\frac{2\log(12)}{n\tau}}$.

Indeed, let $h^*$ be the hypothesis achieving the supremum on the left-hand side.[9] Then:

$$\mathbb{1}(|\Gamma_S(h^*) - \Gamma_p(h^*)| \geq t)\mathbb{1}(|\Gamma_{S'}(h^*) - \Gamma_p(h^*)| \leq t/2) \leq \mathbb{1}(|\Gamma_{S'}(h^*) - \Gamma_S(h^*)| \geq t/2).$$

Taking expectation with respect to $S'$:

$$\mathbb{1}(|\Gamma_S(h^*) - \Gamma_p(h^*)| \geq t)\mathbb{P}_{S'}(|\Gamma_{S'}(h^*, S') - \Gamma_p(h^*)| \leq t/2) \leq \mathbb{P}_{S'}(|\Gamma_{S'}(h^*) - \Gamma_S(h^*)| \geq t/2).$$

Now using Lemma 3:

$$\mathbb{P}_{S'}\left(|\Gamma_{S'}(h^*) - \Gamma_p(h^*)| \leq t/2\right) \geq 1 - 6\exp\left(-\frac{t^2\tau n}{128}\right) \geq 1 - \frac{1}{2} = \frac{1}{2},$$

where the second inequality follows from the condition $t \geq 8\sqrt{\frac{2\log(12)}{n\tau}}$. Therefore,

$$\frac{1}{2}\mathbb{1}(|\Gamma_S(h^*) - \Gamma_p(h^*)| \geq t) \leq \mathbb{P}_{S'}(|\Gamma_{S'}(h^*) - \Gamma_S(h^*)| \geq t/2).$$

Taking expectation with respect to $S$:

$$\begin{aligned}\mathbb{P}_S(|\Gamma_S(h^*) - \Gamma_p(h^*)| \geq t) &\leq 2\mathbb{P}_{S,S'}(|\Gamma_{S'}(h^*) - \Gamma_S(h^*)| \geq t/2)\\ &\leq 2\mathbb{P}_{S,S'}(\sup_{h \in \mathcal{H}} |\Gamma_{S'}(h) - \Gamma_S(h)| \geq t/2).\end{aligned}$$

2) Next we use the symmetrization inequality (54) to bound the large deviation of $\Gamma_S(h)$ uniformly over $\mathcal{H}$. Specifically, given $n$ points $x_1, \ldots, x_n \in \mathcal{X}$, denote

$$\mathcal{H}_{x_1,\ldots,x_n}\{(h(x_1), \ldots, h(x_n)) : h \in \mathcal{H}\}.$$

Then define the growth function of $\mathcal{H}$ as:

$$G_{\mathcal{H}}(n) = \sup_{x_1,\ldots,x_n} |\mathcal{H}_{x_1,\ldots,x_n}|. \tag{55}$$

We will use that well-known Sauer's lemma (Vapnik, 2013), which states that whenever $n \geq d$, $G_{\mathcal{H}}(n) \leq \left(\frac{en}{d}\right)^d$

Notice that given the two datasets $S, S'$, the values of $\Gamma_S$ and $\Gamma_{S'}$ depend only on the values of $h$ on $S$ and $S'$ respectively. Therefore, for any $t \geq 8\sqrt{\frac{2\log(12)}{\tau n}}$,

$$\mathbb{P}_S\left(\sup_{h \in \mathcal{H}} |\Gamma_S - \Gamma_p(h)| \geq t\right) \leq 2\mathbb{P}_{S,S'}\left(\sup_{h \in \mathcal{H}} |\Gamma_{S'}(h) - \Gamma_S(h)| \geq \frac{t}{2}\right) \tag{56}$$

---

[9]If the supremum is not attained, the argument can be repeated for each element of a sequence of classifiers approaching the supremum

$$\leq 2G_{\mathcal{H}}(2n)\mathbb{P}_{S,S'}\left(|\Gamma_{S'}(h) - \Gamma_S(h)| \geq \frac{t}{2}\right) \tag{57}$$

$$\leq 2G_{\mathcal{H}}(2n)\mathbb{P}_{S,S'}\left(\left(|\Gamma_S(h) - \Gamma_p(h)| \geq \frac{t}{4}\right) \vee \left(|\Gamma_{S'}(h) - \Gamma_p(h)| \geq \frac{t}{4}\right)\right) \tag{58}$$

$$\leq 4G_{\mathcal{H}}(2n)\mathbb{P}_S\left(|\Gamma_S(h) - \Gamma_p(h)| \geq \frac{t}{4}\right) \tag{59}$$

$$\leq 24G_{\mathcal{H}}(2n)\exp\left(-\frac{t^2\tau n}{516}\right) \tag{60}$$

$$\leq 24\left(\frac{2en}{d}\right)^d \exp\left(-\frac{t^2\tau n}{516}\right). \tag{61}$$

Here the second-to-last inequality is due to the same bound on the difference between $\Gamma_S$ and $\Gamma_p$ that was used in the previous lemma, and the last one follows from Sauer's lemma. Now if we use the threshold $t = 16\sqrt{2\frac{d\log\left(\frac{2eN}{d}\right)+\log\left(\frac{24}{\delta}\right)}{\tau n}} > 8\sqrt{\frac{2\log(12)}{\tau n}}$, we get:

$$\mathbb{P}_S\left(\sup_{h\in\mathcal{H}}|\Gamma_S(h) - \Gamma_p(h)| \geq 16\sqrt{2\frac{d\log\left(\frac{2en}{d}\right) + \log\left(\frac{24}{\delta}\right)}{\tau n}}\right) < \delta. \tag{62}$$

$\square$

