# OpenReview forum: "FLEA: Provably Robust Fair Multisource Learning from Unreliable Training Data"
_TMLR — Accepted by TMLR_

### Review · Reviewer_mx1h · 2022-06-28

**Summary Of Contributions:**

This work considers the problem of fair learning in a context where the training data is composed of multiple sources, some of which are not representative of the true data distribution. A filtering approach (FLEA) is proposed to remove data instances that could negatively affect fairness and accuracy.

Overall, the proposed approach relies on three metrics (discrepancy, disparity, and disbalance) to ensure that the learned classifier is respectively accurate, fair, and remains fair in presence of data corruptions. More precisely, the pairwise disparity, discrepancy and disbalance between all pairs of data sources are computed and aggregated into a score used to filter out undesirable sources. Moreover, the authors show that FLEA can guarantee the learning of a fair classifier under a 50% data corruption rate. Several experiments are performed and the results show that the proposed approach can significantly improve the performance of fairness-aware learning approaches in the presence of unreliable data.

**Requested Changes:**

Elaborate more on the requirements for dataset size in high corruption setting
- Explain more why the three metrics are aggregated. Why not other alternatives like using the minimum of the three metrics or solving the underlying multiobjective optimization problem?
-  Report false negative rates of the filtering approach

**Strengths And Weaknesses:**

**Strengths**::
The proposed approach is generic enough to cover a wide range of multisource learning scenarios (from both data-quality and adversarial perspectives). In particular, it considers the problem of multisource learning in the presence of an adversary that can arbitrarily manipulate a fixed subset of the sources, which is unknown to the learning algorithm. The authors elaborate on how existing SOTA fairness-aware and multisource learning approaches only partially address the problem. The proposed solution achieves better performance compared to existing approaches. The experiments are quite exhaustive, covering a wide range of scenarios, and demonstrating the generic nature of the solution.

**Weakness**::
The paper did not discuss what "enough data" means in high data corruption settings. Also, some design choices are not well substantiated. For example, why D-Scores are computed as a sum. What about using the minimum of the three metrics? Finally, the performance of the filtering especially false-negative rates could also give more credit to the proposed solution

---

> ### Author Response · Authors · 2022-07-04
> **Author response to reviewer mx1h**
>
> Thank you for the encouraging review. We would like to clarify a few points.
>
> **1) what does "enough data" mean in high data corruption settings?**
>
> We use "enough" to informally quantify that FLEA's guarantees improve with growing sample size both theoretically and empirically. The theoretical result is Theorem 1, which establishes generalization guarantees with $\tilde O(\sqrt{1/n})$ for sample size $n$. More precise expressions appear in the proof (Section D.4), but they are complex and we did not find them to provide a lot of intuition. As is the case for most generalization bounds, we would not recommend using their numerical values for practical decisions.
>
> Experimentally, we observe close to perfect results until the number of samples dropped below approximately 150 per source (`drugs` with N=11, `germancredit` with N>5), but this value will depend also on other factors, such as the data dimension.
>
> **2) why are the three metrics aggregated?**
>
> We need to aggregate the information of the different scores because FLEA has to make a hard decision whether to include a source or not, and each individual score is not sufficient to do so. For the guarantees to hold, the composite should be small only when all three metrics are small. The sum has this property. Deciding based on the *minimum* of the three metrics would overlook some malignant manipulations. For example, the "flip labels" adversary does not  impact disparity at all (that is computed without label information), yet it can catastrophically impact the learned classifier. Theoretical guarantees similar to the ones we state hold indeed if we used the *maximum* of the three scores. We did not find this a more intuitive choice, though, and empirically it was somewhat less robust for small sample sizes.
>
> We are not sure what you mean by "solving the underlying multiobjective optimization problem?", as the three metrics are not used inside of an optimization problem. Could you please clarify this point?
>
> **3) false negative rates of the filtering approach**
>
> We agree that this would be an interesting measure and we had considered running an ablation study for it. However, ultimately we decided against that because we have no proper way of evaluation. Note that the goal of FLEA's filtering step is not to filter out all manipulated sources, but only the *malignant* ones, i.e. the ones that would negatively affect the learning step. Identifying *all* manipulated sources would be impossible, as the adversary can choose to change the data as little as it wants, for example just shuffle the data or even do nothing ("ID" in our experiments).
>
> In our experiments we have the ground truth information about which sources are manipulated, but in some cases the manipulations are *benign* (they do not harm learning), and letting such sources pass should not be considered a "false negative". We do not have ground truth information whether a manipulation is benign or malignant, as this depends not only on the adversary's strategy, but also on the actual data and the later learning strategy.
> If requested we could add a study on how often FLEA’s filtering step filters out the manipulated sources, regardless if these are malignant or benign. However, this would be a (potentially confusing) overestimation of the false negative rate.
>
> In contrast, we consider the accuracy and fairness values we report a better reflection on how well the filtering step succeeds, as it directly reflects if sources that made it through the filtering step had an adverse effect on learning.

---

> > ### Comment · Reviewer_mx1h · 2022-07-08
> > **Thanks for the clarification**
> >
> > Thank you for clarifying 1), 2), and 3).
> >
> > For 2), regarding multi-objective analysis, I was thinking about a scenario where the metrics are optimized. So please ignore this comment.
> >
> > Overall, while additional experiments are not necessary given the answers you provide, I believe emphasizing these points in a discussion section of the paper could help.

---

### Review · Reviewer_RTB6 · 2022-07-03

**Summary Of Contributions:**

This paper considers a filter-based approach towards robust learning with multiple data sources. The setup is motivated by the situations where i.i.d. assumption is not satisfied across data sources. The paper propose to consider dissimilarity (D-score) which consist of three parts, namely, disparity, discrepancy, and disbalance, to capture the existence of "unclean" data sources. An intuitive procedure (Algorithm 1), as well as large sample performance guarantees, is also provided.

**Broader Impact Concerns:**

I do not see any concern in terms of ethical implications of the paper.

**Requested Changes:**

For the usage of disbalance, i.e., disb() term, it would be very helpful if authors can provide some clarifications to avoid misunderstandings. For theoretical analysis (Section 3.3), I think readers can benefit more from the implication of the provided theorem(s) after presenting the technical results for homogeneous and heterogeneous settings.

**Strengths And Weaknesses:**

The strength of the paper lies in the detailed decomposition of the D-score -- discrepancy, disparity, and disbalance -- for the purpose of enforcing robust fair learning, and also the exposition of the approach and experimental results. Here, "fair" is in the sense of demographic parity. In terms of technical details, I think the presentation is overall clear and I don't see obvious technical invalidity.

The weakness of the paper comes from a worry about the usage of disbalance in the D-score. If I understand it correctly, in the literature, disbalance (or representation imbalance) is often referred to as a direct measure of (potential) violation of demographic parity in terms of the disparity in base rates across groups (not sources). Here, disbalance seems to mean the source-level base rates difference. I am not sure how the disparity measurement across sources can help mitigate demographic parity, which is on group level (each data source contains multiple groups). If we only consider fairness violation on prediction, then disp() term serves the purpose. But if we were to consider fairness violation on data itself, I am not sure how disb() term can help.

---

> ### Author Response · Authors · 2022-07-04
> **Author response to reviewer RTB6**
>
> Thank you for the encouraging review.
>
> **1) usage of “disbalance” in the D-score**
>
> This might be a case of overlapping nomenclature that we'll be happy to sort out. A term with a similar definition as our *disbalance* indeed appears in the fairness literature (e.g. Sapiezynski et al, 2017) to quantify potential fairness issues of a dataset. Our measure is different in exact definition and use. We’ll be happy to rename our quantity in case that its name would cause confusion otherwise.
> What we call *disbalance* in the manuscript is not actually a fairness-related measure, but it quantifies if the relative size of protected groups differs between sources. This information is needed to prove the formal guarantees. To see its relevance, imagine an adversary who manipulates sources by dropping a large subset of the examples of one protected group. Demographic parity (2) might not be affected, because that measures differences of per-protected-group *averages* (i.e. absolute group size does not matter). Consequently, the observed *disparity* (4) could be small as well. However, the *disbalance* is sensitive to the action of this adversary. Thereby it allows FLEA to offer protection against group size manipulations, too.
> Note that FLEA’s filtering step does not aim at identifying or fixing fairness violations in individual data sources. That would be part of the fairness-aware learning algorithm, for which we don’t make any assumptions on its working mechanism. FLEA’s filtering step rather identifies data sources that have been manipulated in a potentially malignant way compared to the clean (=unmanipulated) data distribution (regardless if that is fair or unfair).
>
> **2) readers can benefit more from the implication of the provided theorem(s) after presenting the technical results for homogeneous and heterogeneous settings**
>
> We’ll be happy to edit the manuscript such that the readers can understand our contributions better. However, we are not sure what edits exactly you are suggesting. Would you like us to move material from elsewhere in the manuscript to after the technical results? Or do you mean to add new material at that place?

---

> > ### Comment · Reviewer_RTB6 · 2022-07-07
> > **Thank authors for the response**
> >
> > Thank you for your responses.
> >
> > Re: Response 1)
> >
> > Thank you for the clarification. In particular, FLEA "[does] not make any assumptions on working mechanism [of the fair predictor]", and that "FLEA's filtering [...] identifies data sources that have been manipulated ... compared to clean (=unmanipulated), regardless if that is fair or unfair". I am still having some difficulty connecting FLEA's use case to fair robust learning (e.g., as indicated in the title). If FLEA does not have any assumption on fairness definition, how can FLEA guarantee a fair learning process? If I understand correctly, the disp() term is dedicated to fairness violation quantification, right?
> >
> > Re: Response 2)
> >
> > I think readers would naturally wonder how to parse these technical results. Additional information in any form (either rearrangement of material, or some new paragraphs) would be helpful, and I personally do not have preference.

---

> > > ### Author Response · Authors · 2022-07-08
> > > **FLEA guarantees robustness in the context of fair learning**
> > >
> > > **I am still having some difficulty connecting FLEA's use case to fair robust learning (e.g., as indicated in the title). If FLEA does not have any assumption on fairness definition, how can FLEA guarantee a fair learning process?**
> > >
> > > The correct formulation would be that FLEA guarantees robustness in the context of fair learning. We will try to clarify by giving a somewhat bigger picture:
> > >
> > > a) There is a true data distribution $p$ at prediction time. That distribution could typically be biased (otherwise, fairness of classifiers would not be a major issue), but it is the "true distribution" that we cannot influence. The goal is to learn a classifier that will be fair (and accurate) under these prediction-time conditions, according to some fixed definition of fairness (for us: demographic parity).
> > >
> > > b) If we are given i.i.d. data from $p$, it is a well-understood and (arguably) mostly solved problem to learn a classifier that fulfills the chosen fairness definition. Many such *fairness-aware learning methods* exist, based on preprocessing, postprocessing, regularization, adversarial training and more. Our experiments in Table 2-26 confirm this: without data manipulations (bars "oracle" or "ID"), the learned classifiers learned by "ordinary fair training" are accurate and fair.
> > >
> > > c) The situation changes drastically if the training data is not i.i.d. but (partially) noisy or manipulated. The classifiers learned by fairness-aware learning algorithms are often *not* fair and accurate anymore (bars for other adversaries in the table).
> > >
> > > d) If all training data comes in a big chunk, it is provably impossible to overcome the problem of noisy/manipulated data (see other replies).
> > >
> > > e) What we show in our work is: if the data comes in multiple chunks, some of which are i.i.d. from $p$ and some might not be, then the problem can be overcome both in term of theoretical guarantee as well as practical performance. FLEA is our proposed method for this.
> > >
> > > f) FLEA's role is to protect a fairness-aware learning method by suppressing noisy/manipulated data sources. Once these are gone, the remaining data is i.i.d. and the fairness-aware learner can do its job. As a meta-method, FLEA can work with any specific fairness-aware learning algorithm as a subroutine. That's what we mean by "no assumptions on the working mechanism".
> > >
> > > g) To suppress sources, ideally one would want to identify those whose data distribution is different from that if the majority. However, it is impossible to identify *all possible* differences in distribution when given only finite data (see page 4, paragraph 3). Instead, FLEA measures how dissimilar classifiers could become if trained on one data source or another. This is quantified by *disc* (for accuracy) and *disp* and *disb* (for fairness). Finding a large enough group of sources for which these pairwise values are small, one has a guarantee that either the sources are not manipulated, or only in a way that doesn't hurt learning. To get these guarantees, *disp* and *disb* must reflect the designed fairness definition (=demographic parity).
> > >
> > > **If I understand correctly, the disp() term is dedicated to fairness violation quantification, right?**
> > >
> > > Almost. *disp* measures *maximal differences in fairness violation* between two sources. If there exists a classifier that is fair on data source but very unfair on the other, then these two sources are unlikely to come from the same data distribution. If all possible classifiers are (almost) equally fair or unfair on the two datasets ($disp$ is small) then -for the purpose of fairness- we can safely merge the sources.

---

### Review · Reviewer_77G8 · 2022-07-04

**Summary Of Contributions:**

The paper proposes a filtering-based algorithm to identify the unreliable data sources that have a negative impact on fairness or accuracy in a multisource learning setting. The major contribution is to establish that as long as the unreliable data sources constitute less than half of all sources, it is still able to identify and suppress their negative impact by utilizing three dissimilarity measures between the data sources (disparity, discrepancy, and disbalance).

**Requested Changes:**

1. The paper should make the implementation of discrepancy and disparity estimates detailed and clear.

2. The paper should discuss how well the filtering algorithm identifies unreliable data sources (theoretically and empirically) and how it improves the generalization bounds.

2. The paper should discuss the computation complexities and the running time of the filtering process.

4. The paper should include some experiments on more complex models.



**Strengths And Weaknesses:**


Overall, it is a solid paper with theoretical analyses and promising empirical performance. The strengths and weakness are listed as follows:

Strengths:

1. The paper proposes a sound solution to improve fair multisource learning from unreliable training data, which is an important application scenario. The idea of the proposed filtering algorithm is well motivated.

2. The related works are well summarized and the contributions are clearly demonstrated.

3. Theoretical analyses on the accuracy and fairness achieved by the proposed algorithm are well established by the generalization bounds under both homogeneous and heterogeneous data distribution settings.

4. Comprehensive experimental results are presented to demonstrate the effectiveness of FLEA compared with multiple baseline methods on multiple datasets. The empirical performance is promising.

Weakness:

1. The implementation details of the proposed filtering algorithm are not clear. For instance, in section 3.2, the computation of disc and disp measures is not demonstrated sufficiently.

2. The generalization bounds imply that the generalization errors of accuracy and fairness vanish with the rate $1/\sqrt{n}$. However, there is a lack of discussion on how the filtering algorithm improves these generalization bounds compared with other algorithms. For instance, is it possible that an algorithm without filtering can also achieve vanishing generalization error with infinite samples?
Moreover, the theorems do not quantify how well the filtering algorithm identifies the unreliable data sources.

3. Although the paper presents a significant amount of experimental results, all of them are based on simple linear models. It is unclear whether the observations and conclusions can generalize to more complicated models.

5. There is no discussion on computation complexities and the running time of the filtering process.

6. The paper claims to defend against adversarial attacks, but all the data attacks are very simple random attacks that are not really adversarial. It is unclear how the proposed filtering method performs under stronger adversarial attacks.

7. In the experiments in the appendix, the proposed FLEA sometimes even outperforms the oracle performance where unreliable data sources are excluded beforehand. Clarifications of this will be helpful.

---

> ### Author Response · Authors · 2022-07-06
> **Author response to reviewer 77G8**
>
> Thank you for your detailed review. In the following we hope to address your concerns:
>
> **1) The implementation details of the proposed filtering algorithm are not clear. For instance, in section 3.2, the computation of disc and disp measures is not demonstrated sufficiently.**
>
> Computing *disc* and *disp* consists of training classifiers on specific subsets of the data. We can expand the description and add pseudocode. We will also release our source code on github.
>
> **2) There is a lack of discussion on how the filtering algorithm improves these generalization bounds compared with other algorithms. For instance, is it possible that an algorithm without filtering can also achieve vanishing generalization error with infinite samples?**
>
> None of the other algorithms provides generalization bounds for fairness.
>
> A bound for accuracy only in the homogeneous setting is given in (Konstantinov et al, 2020). It has the same $\sqrt{1/n}$-dependence as ours, which was in fact proven to be optimal in the same paper and (Jain and Orlitsky, 2020).
>
> For algorithms without filtering, i.e. merging data sources and then running any possible procedure on that, it is provably impossible to guarantee a vanishing generalization error under our adversarial model, even if the number of samples and/or the number of samples goes to infinity. For *accuracy* this was proven in (Kearns&Li, 1993), for *fairness* in (Konstantinov&Lampert, 2022). We mention this as the ‘single dataset’ case but will be happy to clarify it further.
>
> **The theorems do not quantify how well the filtering algorithm identifies the unreliable data sources.**
>
> This is an important aspect that we will clarify in our manuscript. The theorems cannot give guarantees that all manipulated sources are identified and suppressed. Such guarantees are in fact mathematically impossible, see our examples below. The theorems do provide guarantees that learning will not be negatively affected, which we believe is what ultimately matters.
>
> The point of empirically measuring filtering success was raised also by Reviewer mx1h. Here is a copy of our answer:
>
> *We agree that this would be an interesting measure and we had considered running an ablation study for it. However, ultimately we decided against that because we have no proper way of evaluation. Note that the goal of FLEA's filtering step is not to filter out all manipulated sources, but only the malignant ones, i.e. the ones that would negatively affect the learning step. Identifying all manipulated sources would be impossible, as the adversary can choose to change the data as little as it wants, for example just shuffle the data or even do nothing ("ID" in our experiments).*
>
> *In our experiments we have the ground truth information about which sources are manipulated, but in some cases the manipulations are benign (they do not harm learning), and letting such sources pass should not be considered a "false negative". We do not have ground truth information whether a manipulation is benign or malignant, as this depends not only on the adversary's strategy, but also on the actual data and the later learning strategy. If requested we could add a study on how often FLEA’s filtering step filters out the manipulated sources, regardless if these are malignant or benign. However, this would be a (potentially confusing) overestimation of the false negative rate.*
>
> *In contrast, we consider the accuracy and fairness values we report a better reflection on how well the filtering step succeeds, as it directly reflects if sources that made it through the filtering step had an adverse effect on learning.*

---

> ### Author Response · Authors · 2022-07-06
> **Author response to reviewer 77G8 (continued)**
>
> **3) Although the paper presents a significant amount of experimental results, all of them are based on simple linear models. It is unclear whether the observations and conclusions can generalize to more complicated models.**
>
> Indeed, it is our choice of linear classifiers that allows us to test many scenarios (datasets, data splits, adversaries, methods to enforce fairness, baselines and random seeds). In total, we reported results from over 70.000 experiments, many of which required training multiple classifiers as internal steps. We’ll be happy to also add experiments for non-linear classifiers, but given the much higher computational cost this could necessarily only be for a subset of the settings. Please let us know which would be the most interesting ones to you.
>
> Note that our theory holds for arbitrary classifier families, but the number of necessary samples per source grows with the complexity of the hypothesis set. A practical option (without formal guarantees) is to run the filtering step of FLEA with linear classifiers but then use more complex fairness-aware classifiers afterwards. We don’t expect surprises from such experiments, though: whenever the filtering step succeeds, which is in most tested cases, the resulting data is simply a clean dataset, so we can train *any* classifier afterwards without any concerns.
>
> A different aspect is that for more complex classifiers, the adversaries might have more subtle options to manipulate the data. In contrast to the linear setting, these would probably have to be created numerically. Please see also our reply to 5).
>
> **4) There is no discussion on computation complexities and the running time of the filtering process.**
>
> We’ll be happy to expand our discussion on this and move Appendix A.4 to the main manuscript.
>
> For FLEA’s filtering step, the main cost is training classifiers on pairs of sources. Afterwards, a final classifier is computed on $Kn$ data points. Overall, the computational complexity is $O(M^2 F(2n)) + F(Kn)$, where $F$ is the complexity of training a single classifier. Note that $M/2<K\leq M$, so if $F$ is linear, the complexity of the filtering term dominates, if $F$ is quadratic, both terms are of comparable order. If $F$ is of higher order, the final classifier training dominates the computational load.
>
> The practical runtime depends on many other factors, of course. As we detail in Appendix A.4, for us each experiment (filtering + classifier training) took a few minutes to compute, varying depending on the data, learning methods, etc.
>
> **5) The paper claims to defend against adversarial attacks, but all the data attacks are very simple random attacks that are not really adversarial. It is unclear how the proposed filtering method performs under stronger adversarial attacks.**
>
> Thank you for raising this issue. We discuss this in Section 4.1, but we will clarify it in the manuscript.
>
> The attacks are indeed adversarial in the sense that they are designed to maximally affect fairness and/or accuracy while (for some of them) staying hard to detect. The design was done manually, because for the linear classifiers used one can read off the effects of manipulations directly without the need for gradient-based techniques. We would be happy to try adversaries based on numeric optimization as well, though we are currently not aware of techniques that can attack discrete/categorical data as are present in most fairness datasets.
>
> **6) In the experiments in the appendix, the proposed FLEA sometimes even outperforms the oracle performance where unreliable data sources are excluded beforehand. Clarifications of this will be helpful.**
>
> We’ll be happy to look into this. Are you referring to any specific table or figure? In general, when FLEA filters out exactly the manipulated sources, the resulting classifier will be identical to the oracle one. When FLEA lets some manipulated sources (benign or malignant) pass, the results could -by random chance- be better than for the oracle. This should not be a statistically significant effect, though. In case you are referring to cases with higher fairness but lower accuracy, this would not be so surprising, as achieving high fairness by itself can happen quite easily, e.g. by random or constant classifiers.

---

> > ### Comment · Reviewer_77G8 · 2022-08-03
> > **Thanks for the detailed reponses**
> >
> > Dear authors,
> >
> > Thank you for the detailed responses that resolve my concerns. The revision also clearly clarifies the contributions and limitations of this work.

---

### Review · Reviewer_68xw · 2022-07-04

**Summary Of Contributions:**

The authors propose FLEA, a filtering approach for fair federated learning in presence of unreliable training data from malicious devices. The authors do a decent job putting their work in context, provide an algorithm, provide theoretical guarantees followed by experiments. From a conceptual point of view, the **federated learning from multiple malicious sources problem** and **fair federated learning problem** are treated in a disjoint manner in this paper and the contribution mostly lies on the former front. As detailed in the summary of strengths and weaknesses, unfortunately the treatment of the paper in its current form is lacking in several major ways, which requires a major revision.

**Broader Impact Concerns:**

Broader impact is sufficiently addressed.

**Requested Changes:**

* Is there a converse bound for Thm 1 that can establish the linear dependence on $\eta$ is optimal? Can you please elaborate?

* At the end of Section 3, you want to connect with personalized FL literature as they overcome the dependence on $\eta$ via personalization.

* In practice, one would imagine that the filtering and federated model learning need to be performed jointly. Can the authors comment on that?

* Please compare with robust aggregation methods such as multi krum.

* Please compare FilterSource with (Khetan et al 2018) and (Li et al 2021) as they also perform filtering of unreliable sources.

* Please chime in on partial device participation and how the filtering can be done in a federated manner itself.

* Please include the implementation details of the filtering algorithm, especially as it intersects with federated learning. Currently, it is not clear what each device needs to do at each round.

* Please empirically report settings where number of devices scale, and showcase the scaling behavior of the proposed algorithm with number of devices.

* Please compare your end-to-end algorithm with hierarchical TERM (Li et al 2021), where negative tilting is done at device level and positive tilting at sensitive attribute level.

**Strengths And Weaknesses:**


Strengths:
* The paper is generally well-written, and related work is discussed in a decent manner.

* The authors derive theoretical guarantees for robust federated learning in heterogenous settings.

Weaknesses:
* As authors have correctly mentioned, learning from multiple unreliable sources has been studied in the literature before. Here I point out to key recent baselines that are missing from the mix, where they study the same problem in the context of noisy annotators. Without comparisons with these works, it is not easy to compare the empirical performance of the proposed FilterSource method with those existing in the literature.

Khetan, A., Lipton, Z.C. and Anandkumar, A., 2018. Learning from noisy singly-labeled data. ICLR.

Li, T., Beirami, A., Sanjabi, M. and Smith, V., 2021. Tilted empirical risk minimization. ICLR.

* The efficacy of the proposed filtering approach in a federated setting is unclear. While this is implicit, given lack of details on the algorithm, my understanding is that the authors are assuming **full device participation** at each round which is unrealistic for real-world federated setups.

* It is unclear where and how the computations required for FilterSource are performed. In particular, these computations scale quadratically with the number of devices, something that is prohibitive when we have millions of devices. Without the details (which are mostly missing), I am not able to assess the applicability of the proposed algorithm in real world settings.

* The experimental setup mostly encompasses linear models. It is not clear whether the proposed idea extends beyond that.

* A strong baseline for the proposed algorithm would be hierarchical TERM (Li et al 2021), where negative tilting is done at device level and positive tilting at sensitive attribute level. This baseline is missing.

* A discussion around key baselines on fair federated learning is missing:

Mohri, M., Sivek, G. and Suresh, A.T., 2019, May. Agnostic federated learning. In International Conference on Machine Learning (pp. 4615-4625). ICML.

Li, T., Sanjabi, M., Beirami, A. and Smith, V., 2020. Fair resource allocation in federated learning. ICLR.

---

> ### Author Response · Authors · 2022-07-06
> **Request for clarification**
>
> Dear Reviewer 68xw,
>
> Thank you for the detailed review. Unfortunately, we believe there has been a misunderstanding that we'd like to sort out before formulating a more detailed response.
>
> *Our work is not about federated learning. We study learning with data that comes from multiple sources, but in a centralized setting, not distributed. Data privacy is of no particular concern, and clients do not drop in and out (in fact, there are no 'clients').*
>
> Of course, many of your comments are relevant and interesting nevertheless, and we'll be happy to adapt our manuscript accordingly. However, in light of the misunderstanding and the time and space constraints, would it be possible that you give us an indication which aspects we should concentrate on in our reply and which might not be necessary?

---

> > ### Comment · Reviewer_68xw · 2022-07-07
> > **Thanks for clarification**
> >
> > Thanks for your quick note!
> >
> > I apologize for misunderstanding your work, but also find this an opportunity for further clarification. Statements like what I have quoted below give the impression that you are looking at federated learning problems. At the very least a clear setup of the problem and usecases throughout the paper should help better situate the work. You may also add a note that federated learning is a non-goal. As you also mentioned, many of the other comments apply regardless of whether the setup is federated.
> >
> > *To our awareness, the only prior work that considers achieving fairness in a multisource learning setting and in the presence of data corruption is the one of Li et al. (2021). However, this paper focuses on personalized federated learning and on a fairness objective which postulates that models’ performances should be relatively similar across edge devices. In contrast, we study a setup
> > where a single global model is trained and aim to ensure that this model does not act discriminatory against members of protected subgroups.*
> >
> > Given that the setup is a centralized one, with an additional feature identifying the source ID of the data, I think it is crucial to put the work in the right context through a revision. In this case, there is no distinction between the FilterSource and learning from noisy annotators; so I think those would be crucial baselines for this work.
> >
> > In addition, there is work that is specifically looking into adversarial fairness attacks, which goes one extra step in looking at an environment where adversaries are specifically trying to fool the model to become unfair. See for example (Mehrabi et al 2021). I think making a connection with this literature is also very important because as it currently stands the fairness improvement is a byproduct of the filtering, and not the goal of your filtering. In other words, you are not actively trying to defend against fairness attacks. This should also be clearly discussed.
> >
> > Mehrabi, N., Naveed, M., Morstatter, F. and Galstyan, A., 2021, May. Exacerbating algorithmic bias through fairness attacks. In Proceedings of the AAAI Conference on Artificial Intelligence (Vol. 35, No. 10, pp. 8930-8938).
> >
> > Finally, once the revisions to situate the work are done, you can perhaps safely ignore my comments on connections with fair federated learning literature. Further, all my comments of partial device participation can be safely ignored after such revision. However, I believe the rest of the comments/questions (including questions about scaling with the number of source, comparison with {robust aggregation methods, learning from noisy annotators, robustness against fairness attacks}) are still applicable.
> >
> > Hope that helps!\
> > Reviewer 68xw

---

> > > ### Author Response · Authors · 2022-07-08
> > > **Short follow-up**
> > >
> > > Thanks for your reply. As requested, we will prepare revision including clarifications and adapted experiments. However, to avoid another misunderstanding:
> > >
> > > **[...] as it currently stands the fairness improvement is a byproduct of the filtering, and not the goal of your filtering. In other words, you are not actively trying to defend against fairness attacks.**
> > >
> > > This is not correct. FLEA's filtering step is specifically designed to suppress sources that could affect fairness, regardless of how their data was manipulated (in particular, if a fairness attack algorithm was used, regardless of which one). That's the dedicated role of *disp/disb*, see our "bigger picture" reply to Reviewer RTB6. Our Theorems guarantee that after filtering the fairness of any classifier on the training set is close to its fairness at prediction time, thereby enabling fairness-aware learning with the need for any additional "defences". This is by design, not by accident.
> > >
> > > Our experiments illustrate this, admittedly not against the method of (Mehrabi, 2021), but against many others, even stronger ones. Note that (Mehrabi, 2021) is a single dataset method that tries to maximally hurt fairness while changing only very few points of the original dataset. Our adversaries have no limiting constraints of making only small modifications. They can change all data of their source in an arbitrary way. That makes it much easier to construct strong adversaries. E.g. one of the tested adversaries overwrites the target value of all examples by their protected attribute. That simple rule already would cause an absolute maximum of unfairness, if one simply learned on the resulting data.
> > >
> > > **[...] there is no distinction between the FilterSource and learning from noisy annotators**
> > >
> > > We'll expand our discussion, but the settings do differ. In the noisy annotator setting typically only the labels are affected by noise or manipulations, whereas FLEA's adversaries, all data can be changed. Consequently, FLEA's guarantees also hold for noisy annotators, but also for a broader class of problems (sensor errors, mistakes made during data entering, data collection bias...).

---

> > > > ### Comment · Reviewer_68xw · 2022-07-09
> > > > **RE: Short follow-up**
> > > >
> > > > Dear authors:
> > > >
> > > > I apologize again for misunderstanding your work. However, absent the details of FilterSource algorithm, I was not able to understand/assess how fairness attacks were being tackled by your framework. In this case, it would perhaps make the most sense to first revise the paper and include the details of the algorithmic contribution so that we can get the same page.
> > > >
> > > > Thanks,\
> > > > Reviewer 68xw

---

> ### Author Response · Authors · 2022-07-13
> **Clarification about requested TERM experiments**
>
> Dear Reviewer 68xw,
>
> We're in the process of revising the manuscript and running additional experiments as requested, including non-linear classifiers and the attack from (Mehrabi et al 2021). You also asked about a comparison **to hierarchical TERM (Li et al 2021), where negative tilting is done at device level and positive tilting at sensitive attribute level**.
>
> Are we correct that by this you mean to have the sum of two group-TERM loss terms, one grouped by protected attribute and one grouped by source, but without another level of tilting inside of the losses' exponents? We're asking because in the *hierarchical* TERM one tilted loss would appear within the exponent of the other tilted loss, but we're not sure how this would reflect our setting, in which there are two orthogonal ways to split the data.

---

> > ### Comment · Reviewer_68xw · 2022-07-13
> > **Re: clarification on TERM**
> >
> > Thanks for the update!
> >
> > I think what I had in mind is two hierarchies as follows:
> >
> > For each source, compute TERM with positive tilting with respect to the sensitive attribute, i.e., the average of the loss given each sensitive attribute and then aggregated via positive tilting.
> >
> > Then, aggregate the per source (tilted) losses using a negative tilt.
> >
> > Hope that clarifies! If it is still unclear, please let me know and I’ll type up what I mean mathematically.

---

> > > ### Author Response · Authors · 2022-07-13
> > > **Re: clarification on TERM**
> > >
> > > Okay, thanks for the quick clarification. We'll try that then.

---

### Decision · Action_Editors · 2022-08-05

**Recommendation:** Accept as is

**Comment:**

The paper studies an important problem where one tries to learn a fair classifier with multiple training datasets, some of which are adversarially corrupted. The theoretical contribution is solid though the novelty might be limited. (It's a relatively straightforward extension of robust multi-source learning.). Empirical evaluation is quite extensive and shows fairly promising results.

The reviewers unanimously agreed to accept this paper as it is.  One of the reviewers gave a harsh review, but it was based on their own misunderstanding that the paper was considering federated learning scenarios.  The authors clarified this point during the discussion phase, and the reviewer appropriately updated their review.  There were some constructive (and critical) comments that the reviewers suggested, and the authors successfully addressed them, e.g., they clarified how the newly proposed metrics are estimated by adding pseudocodes in the appendix, and they added a clear discussion on computational complexity.  The revision reads much better.

No certification applicable.

In sum, I recommend to accept the paper as it is.  Let me know if you have any questions.